# Characteristics of building fragility curves for seismic and non-seismic tsunamis: case studies of the 2018 Sunda Strait, 2018 Sulawesi-Palu and 2004 Indian Ocean tsunamis.

Elisa Lahcene[1], Ioanna Ioannou[2], Anawat Suppasri[3], Kwanchai Pakoksung[3], Ryan Paulik[4], Syamsidik Syamsidik[5], Frederic Bouchette[1] and Fumihiko Imamura[3]

[1]GEOSCIENCES-Montpellier, Montpellier University II, France
[2]EPICentre, Department of Civil, Environmental and Geomatic Engineering, University College London, UK
[3]International Research Institute of Disaster Science, Tohoku University, Japan
[4]National Institute of Water and Atmospheric Research (NIWA), Wellington, New Zealand
[5]Tsunami and Disaster Mitigation Research Center (TDMRC), Universitas Syiah Kuala, Banda Aceh, Indonesia

*Correspondence to:* Elisa Lahcene (elisa.lahcene54@gmail.com), Ioanna Ioannou (ioanna.ioannou@ucl.ac.uk)

**Abstract.** Indonesia has experienced several tsunamis triggered by seismic and non-seismic (i.e., landslides) sources. These events damaged or destroyed coastal buildings and infrastructure, and caused considerable loss of life. Based on the Global Earthquake Model (GEM) guidelines, this study assesses the empirical tsunami fragility to the buildings inventory of the 2018 Sunda Strait, 2018 Sulawesi-Palu and 2004 Indian Ocean (Khao Lak/Phuket, Thailand) tsunamis. Fragility curves represent the impact of tsunami characteristics on structural components and
express the likelihood of a structure reaching or exceeding a damage state in response to a tsunami intensity measure. The Sunda Strait and Sulawesi-Palu tsunamis are uncommon events still poorly understood compared to the Indian Ocean tsunami (IOT) and their post-tsunami databases include only flow depth values. Using TUNAMI two-layer model, we thus reproduce the flow depth, the flow velocity and the hydrodynamic force of these two tsunamis for the first time. The flow depth is found to be the best descriptor of tsunami damage for both events.
Accordingly, the building fragility curves for complete damage reveal that: (i) in Khao Lak/Phuket, the buildings affected by the IOT sustained more damage than the Sunda Strait tsunami, characterized by shorter wave periods, (ii) the buildings performed better in Khao Lak/Phuket than in Banda Aceh (Indonesia). Although the IOT affected both locations, ground motions were recorded in the city of Banda Aceh and buildings could have been seismically damaged prior to the tsunami arrival, and (iii) the buildings of Palu-City exposed to the Sulawesi-Palu tsunami
were more susceptible to complete damage than the ones affected by the IOT, in Banda Aceh, between 0 and 2 m-flow depth. Similar to the Banda Aceh case, the Sulawesi-Palu tsunami load may not be the only cause of structural destruction. The buildings susceptibility to tsunami damage in the waterfront of Palu-City could have been enhanced by liquefaction events triggered by the 2018 Sulawesi earthquake.

## 1.  Introduction

Indonesia is regularly facing natural disasters such as earthquakes, volcanic eruptions and tsunamis because of its geographic location in a subduction zone of three tectonic plates (Eurasian, India-Australian and Pacific plates) (Marfai et al., 2008; Sutikno, 2016). The Sunda Arc extends for 6 000 kilometers, from the North of Sumatra to Sumbawa Island (Lauterjung et al., 2010) (Fig. 1a). Megathrust earthquakes regularly occur in this region, causing
horizontal and vertical movement of the ocean floor, which tends to be tsunamigenic (McCloskey et al., 2008; Nalbant et al., 2005; Rastogi, 2007). These tsunamis are likely to cause greater destruction as they can follow prior damaging earthquake ground shaking and/or liquefaction (Sumer et al., 2007; Sutikno, 2016). Earthquake-

generated tsunamis also tend to have longer wave periods attacking the coast than non-seismic ones (Day, 2015; Grezio et al., 2017). On Dec. 26 2004, the Sumatra-Andaman earthquake ($M_w$ = 9.0-9.3) hit the north of Sumatra, Indonesia (Fig. 1b). The rupture of the seafloor is estimated at 1200 km length and around 200 km width (Ammon et al., 2005; Krüger and Ohrnberger, 2005; Lay et al., 2005). In the city of Banda Aceh, a strong ground shaking was recorded (Lavigne et al., 2009). This megathrust earthquake was the second largest ever recorded (Løvholt et al., 2006) and caused the deadliest tsunami in the world. Overall, a dozen of Asian and African countries have been devastated, with around 280 000 casualties (Asian Disaster Preparedness Center, 2007; Suppasri et al., 2011). Although earthquakes represent the main cause of tsunamis, non-seismic events such as landslides can also initiate tsunami waves (Grezio et al., 2017; Ward, 2001). After a few months of volcanic activity in the Sunda Strait, Indonesia, the Anak Krakatau Volcano erupted on Dec. 22 2018, leading to its southwestern flank failure (Fig. 1c). It triggered a relatively short wave period tsunami (~7 min) (Muhari et al., 2019), which devastated the western coast of Banten and the southern coast of Lampung with a death toll of 437 (Heidarzadeh et al., 2020; Muhari et al., 2019; National Agency for Disaster Management (BNPB), 2018; Syamsidik et al., 2020). The tsunami generation process is unclear. The subaerial/submarine landslide volume is still investigated and ranges between 0.10 and 0.30 km$^3$ according to recent studies (Dogan et al., 2021; Grilli et al., 2019; Omira and Ramalho, 2020; Paris et al., 2020; Williams et al., 2019). Almost two months before this event, an unexpected tsunami struck Palu-Bay, on Sulawesi Island, claiming 2 000 deaths and considerable loss to property (Association of Southeast Asian Nations (ASEAN)-Coordinating Centre for Humanitarian Assistance on disaster, 2018; Omira et al., 2019). The Sulawesi earthquake ($M_w$ = 7.5) occurred near the Palu-Koro strike-slip fault, 50 km northwest of Palu-Bay (Fig. 1d) (Socquet et al., 2019). Ground shaking led to significant liquefaction along the coast (Paulik et al., 2019; Sassa and Takagawa, 2019). The fault mechanism did not suggest that the tsunami would be so destructive. The wave reached rapidly Palu (~8 min), implying that its source was inside or near the bay (Muhari et al., 2018; Omira et al., 2019). Its short wave period (~3.5 min) also indicates a non-seismic source (i.e., landslide). Some studies suggested that submarine landslides are responsible for the main tsunami. Moreover, a dozen of coastal landslides were reported during field surveys and likely contributed to amplify tsunami waves (Arikawa et al., 2018; Muhari et al., 2018; Omira et al., 2019; Pakoksung et al., 2019). However, according to Ulrich et al. (2019), those subaerial/submarine landslides may not be the only tsunami source as the Sulawesi earthquake rupture may have also induced a large portion of the tsunami waves.

The term "tsunami fragility" is a measure recently proposed to estimate structural damage and casualties caused by a tsunami, as mentioned by Koshimura et al. (2009b). Tsunami fragility curves are functions expressing the damage probability of structures (or death ratio) based on the hydrodynamic characteristics of the tsunami inundation flow (Koshimura et al., 2009b, 2009a). These functions have been widely developed after tsunami events such as the 2004 IOT (Koshimura et al., 2009a, 2009b; Murao and Nakazato, 2010; Suppasri et al., 2011), the 2006 Java tsunami (Reese et al., 2007), the 2010 Chilean tsunami (Mas et al., 2012) or the 2011 Great East Japan tsunami (Suppasri et al., 2012, 2013). Several methods aim to develop building fragility curves based on (i) a statistical analysis of on-site observations during field surveys of damage and flow depth data (empirical methods) (Suppasri et al., 2015, 2020), (ii) the interpretation of damage data from remote sensing coupled with tsunami inundation modelling (hybrid methods) (Koshimura et al., 2009a; Mas et al., 2020; Suppasri et al., 2011)

or (iii) structural modelling and response simulations (analytical methods) (Attary et al., 2017; Macabuag et al., 2014).

Here, we empirically developed building fragility curves for the 2018 Sunda Strait, 2018 Sulawesi-Palu and 2004 Indian Ocean (Khao Lak/Phuket, Thailand) tsunamis based on the GEM guidelines (Rossetto et al., 2014). From the field surveys conducted after the 2018 Sunda Strait (Syamsidik et al., 2019b), 2018 Sulawesi-Palu (Paulik et al., 2019) and 2004 Indian Ocean (Khao Lak/Phuket, Thailand) (Foytong and Ruangrassamee, 2007; Ruangrassamee et al., 2006) events, we identify three databases called DB_Sunda2018, DB_Palu2018 and

DB_Thailand2004, respectively. In the literature, tsunami inundation modelling has been performed many times to better understand the tsunami hydrodynamic, especially for earthquake-generated tsunamis (Charvet et al., 2014; Gokon et al., 2011; Koshimura et al., 2009a; Macabuag et al., 2016; De Risi et al., 2017; Suppasri et al., 2011). Compared to the 2004 IOT, the 2018 Indonesian tsunamis are uncommon events remaining less understood. Therefore, to improve our understanding of the structural damage caused by the Sunda Strait and Sulawesi-Palu

tsunamis and to discuss the impact of wave period, ground shaking and liquefaction events, we reproduce their tsunami intensity measures (i.e., flow depth, flow velocity and hydrodynamic force) based on two-layer modelling (TUNAMI two-layer). We then compared the fragility curves of the Sunda Strait, Sulawesi-Palu and Indian Ocean (Khao Lak/Phuket) tsunamis to those derived for the 2004 IOT in Banda Aceh (Indonesia), produced by Koshimura et al. (2009a). In this study, we explore the characteristics of building fragility curves for the 2018 Sunda Strait

event and 2004 IOT in Khao Lak/Phuket, as well as for complex events, such as the 2018 Sulawesi-Palu tsunami in Palu-City and the 2004 IOT in Banda Aceh, where the tsunamis may not be the only cause of structural destruction. Studying the impact of the wave period, ground shaking and liquefaction events on the structural performance of buildings aims to improve our knowledge on the relationship between local vulnerability and tsunami hazard in Indonesia.

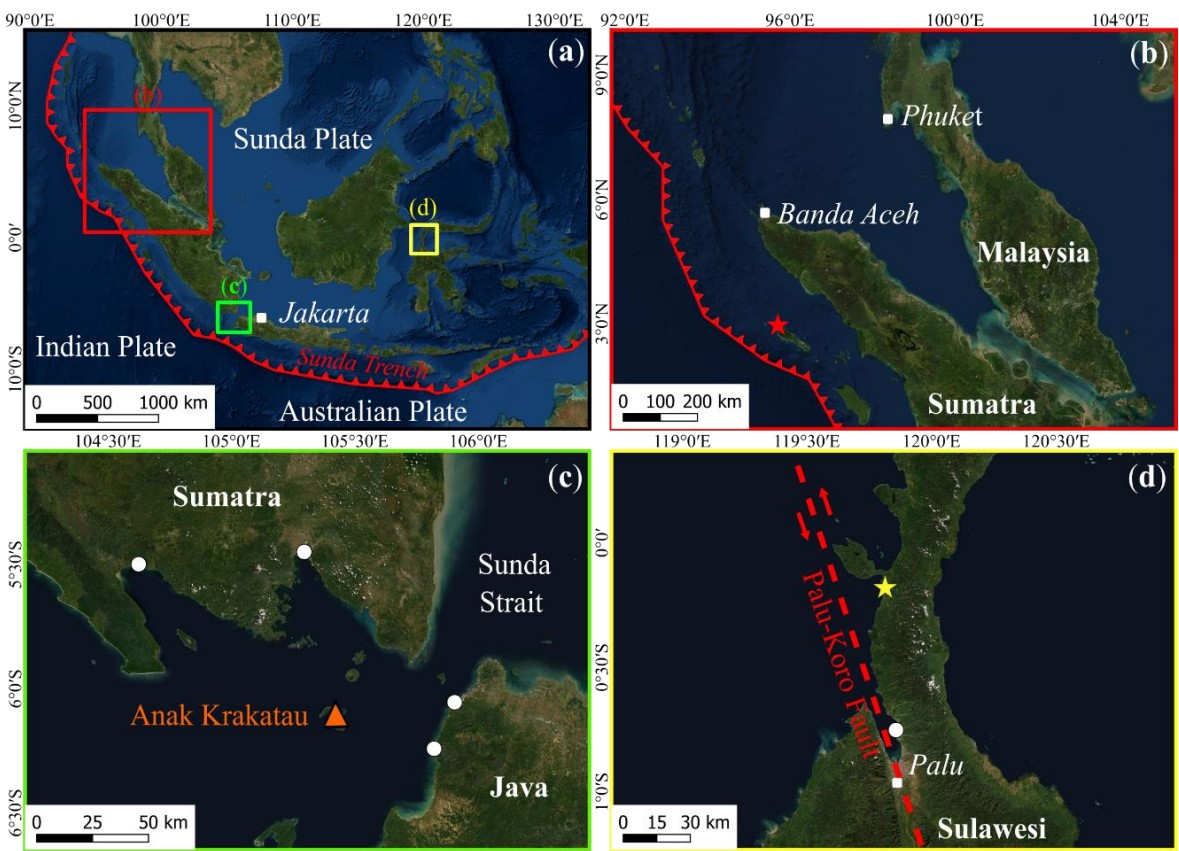

Epicentre of the 2004 Indian Ocean Earthquake
Epicentre of the 2018 Sulawesi-Palu Earthquake (Pakoksung et al., 2019)
Tidal gauge

**Figure 1. (a) Indonesia partially surrounded by the Sunda Trench, (b) epicenter location of the 2004 Indian Ocean earthquake, (c) location of the Sunda Strait and the Anak Krakatau volcano and (d) epicenter location of the 2018 Palu-Sulawesi earthquake (Pakoksung et al., 2019) and the Palu-Koro fault crossing Palu-Bay, on Sulawesi Island, Indonesia (background ESRI).**


## 2. Post-tsunami databases

A post-tsunami database has been established for Sunda Strait area by Syamsidik et al. (2019b), Palu-Bay by Paulik et al. (2019) and Khao Lak/Phuket by Ruangrassamee et al. (2006) and Foytong and Ruangrassamee (2007) in urban areas strongly affected by these events. These databases includes 98, 371 and 120 observed flow depth

traces at buildings, respectively. Here, the tsunami fragility analysis stands on subsets of the original databases of the 2018 Sunda Strait, 2018 Sulawesi-Palu and 2004 Indian Ocean (Khao Lak/Phuket) tsunamis, as explained in sections 3.2.2, 3.2.3 and 2.2, respectively. We define these subsets as "new" databases and we call them DB_Sunda2018, DB_Palu2008 and DB_Thailand2004, respectively. We note that the use of smaller databases for the fragility assessment is expected to increase the uncertainty in the exact shape of the fragility curves. Each

database gathers exclusive information regarding the degree of damage, the building characteristics and the flow depth traces (Tables 1,2). A brief analysis of the key variables (i.e., damage scale, building class, and tsunami intensity) are presented below.

### 2.1. Damage state

Each field survey adopted a different scale to record the degree of structural damage. In DB_Sunda2018, the five-

state damage scale, proposed by Macabuag et al. (2016) and Suppasri et al. (2020), is adopted, ranging from no

damage to complete damage/washed away. In DB_Palu2018, the observed damage was classified into four states: no damage, partial damage repairable, partial damage unrepairable and complete damage, as proposed by Paulik et al. (2019). Finally, in DB_Thailand2004, a four-state damage scale is defined by Ruangrassamee et al. (2006). To simplify the comparison between the fragility curves, a harmonization of damage scales is proposed (Table 1).

In this study, a four-state damage scale ranging from $ds_0$- $ds_3$ is used.

**Table 1. Harmonization between the different damage scales used in DB_Sunda2018, DB_Palu2018 and DB_Thailand2004.**

| Damage state | DB_Sunda2018 | DB_Palu2018 | DB_Thailand2004 |
|---|---|---|---|
| $ds_0$ | No damage | No damage | No damage |
| $ds_1$ | Minor damage<br>Moderate damage | Partial damage, repairable | Damage to secondary members |
| $ds_2$ | Major damage | Partial damage, unrepairable | Damage to primary members |
| $ds_3$ | Complete damage, washed away | Complete damage | Collapse |

### 2.2. Building characteristics

Each survey also recorded the building construction type, which influences the damage probability (Suppasri et al., 2013). In Table 2, among the 94 buildings included in DB_Sunda2018: 67 are confined masonry, 26 are timber and 1 is steel frame building. In DB_Palu2018, most of the buildings are confined masonry with unreinforced clay bricks (~95 %). The database also includes reinforced concrete and timber buildings. Finally, DB_Thailand2004 contains only reinforced concrete buildings. We note that after the 2004 IOT, 120 flow depth traces were recorded

at reinforced concrete structures (e.g., residence, hotel, school, shop, bridge…) in Khao Lak/Phuket area. As we are not considering the data regarding the surveyed bridges, DB_Thailand2004 includes only 117 reinforced concrete buildings.

### 2.3. Tsunami intensity

The tsunami intensity has been measured in terms of flow depth level. Table 2 also presents the number of flow depth traces at surveyed buildings and the range of flow depth levels for each database.

**Table 2. Observed flow depth traces at buildings, range of flow depth levels and building characteristics in DB_Sunda2018, DB_Palu2018 and DB_Thailand2004.**

| | DB_Sunda2018 | DB_Palu2018 | DB_Thailand2004 |
|---|---|---|---|
| **Observed flow depth traces at buildings** | 94 | 124 | 117 |
| **Range of observed flow depth levels at buildings (m)** | (0.20, 6.60) | (0.10, 3.65) | (0.15, 10.00) |
| **Number of buildings per construction type** | 67 confined masonry<br>26 timber<br>1 steel | 119 confined masonry<br>4 reinforced concrete<br>1 timber | 117 reinforced concrete |

### 3. Tsunami intensity simulations

### 3.1. Tsunami numerical modelling with a landslide source

### 3.1.1. Tsunami inundation model

The tsunami model TUNAMI two-layer used in Sunda Strait and Palu areas relies on a two-layer numerical model solving non-linear shallow water equations. It considers two-interfacing layers, appropriate kinematic and dynamic boundary conditions at the seafloor, interface, and water surface (Imamura and Imteaz, 1995; Pakoksung et al., 2019). To reproduce the landslide-generated tsunami, we model the interactions between tsunami generation and submarine landslides, as upper and lower layers. The mathematical model performed in the landslide-tsunami code is obtained from a stratified medium with two layers. The first layer, composed of a homogeneous inviscid fluid with constant density, $\rho_1$, represents the seawater, and the second layer is composed of a fluidized granular material with a density, $\rho_s$, and porosity, $\varphi$. As assumed by Macías et al. (2015), the mean density of the fluidized sliding mass is constant and equals $\rho_2 = (1 - \varphi)\rho_s + \varphi\rho_1$. We consider the two layers immiscible. The governing equations are written as follows:

Continuity equation of the seawater (first layer).

$$\frac{\partial Z_1}{\partial t} + \frac{\partial Q_{1x}}{\partial x} + \frac{\partial Q_{1y}}{\partial y} = 0 \tag{1}$$

Momentum equations of the seawater in the x and y directions.

$$\frac{\partial Q_{1x}}{\partial t} + \frac{\partial}{\partial x}\left(\frac{Q_{1x}^2}{D_1}\right) + \frac{\partial}{\partial y}\left(\frac{Q_{1x}Q_{1y}}{D_1}\right) + gD_1\frac{\partial Z_1}{\partial x} + gD_1\frac{\partial Z_2}{\partial x} + \tau_{1x} = 0 \tag{2}$$

$$\frac{\partial Q_{1y}}{\partial t} + \frac{\partial}{\partial x}\left(\frac{Q_{1x}Q_{1y}}{D_1}\right) + \frac{\partial}{\partial y}\left(\frac{Q_{1y}^2}{D_1}\right) + gD_1\frac{\partial Z_1}{\partial y} + gD_1\frac{\partial Z_2}{\partial y} + \tau_{1y} = 0 \tag{3}$$

Continuity equation of the landslide (second layer).

$$\frac{\partial Z_2}{\partial t} + \frac{\partial Q_{2x}}{\partial x} + \frac{\partial Q_{2y}}{\partial y} = 0 \tag{4}$$

Momentum equations of the landslide in the x and y directions.

$$\frac{\partial Q_{2x}}{\partial t} + \frac{\partial}{\partial x}\left(\frac{Q_{2x}^2}{D_2}\right) + \frac{\partial}{\partial y}\left(\frac{Q_{2x}Q_{2y}}{D_2}\right) + gD_2\frac{\partial Z_2}{\partial x} + gD_2\frac{\rho_1}{\rho_2}\frac{\partial Z_1}{\partial x} + \tau_{2x} = 0 \tag{5}$$

$$\frac{\partial Q_{2y}}{\partial t} + \frac{\partial}{\partial x}\left(\frac{Q_{2x}Q_{2y}}{D_2}\right) + \frac{\partial}{\partial y}\left(\frac{Q_{2y}^2}{D_2}\right) + gD_2\frac{\partial Z_2}{\partial y} + gD_2\frac{\rho_1}{\rho_2}\frac{\partial Z_1}{\partial y} + \tau_{2y} = 0 \tag{6}$$

Index 1 and 2 refer to the first and the second layers respectively. $\rho_1$ and $\rho_2$ are the densities of the seawater and the landslide. $Z_i$ $(x,y,t)$, $Q_i$ $(x,y,t)$ and $\tau_i$ $(x,y,t)$ represent the level of the layer based on the mean water level, the vertically integrated discharge and the bottom stress in each layer at each point $(x,y)$ over the time t, respectively (Fig. A1). $D_i$ denotes the thickness of each layer. The fifth term of the momentum equations (Eqs. 2, 3, 5, 6) represents the interaction between the two layers. The tsunami model provides the maximum water flow depth

and flow velocity along the coast during the tsunami inundation. The hydrodynamic force acting on buildings and infrastructure is defined as the drag force per unit width of the structure, as shown in Eq. (7) (Koshimura et al., 2009b).


$$F = \frac{1}{2} C_D \rho u^2 D \tag{7}$$

$C_D$ represents the drag coefficient ($C_D = 1.0$ for simplicity), $\rho$ is the seawater density ($\rho = 1000$ kg/m$^3$), $u$ stands for the current velocity (m/s), and $D$ is the inundation depth (m).

**3.1.2 Flow resistance within a tsunami inundation area**

BATNAS and DEMNAS, Indonesia, provided the bathymetric and topographic data with 180 and 8 m-resolutions, respectively. The data was established from SAR images (http://tides.big.go.id/DEMNAS/index.html). Both datasets were resampled to three computational domains with a grid size of 20-m resolution (Fig. 2a,b). In Palu-City, the bathymetric and topographic data with 1-m resolution were obtained through Lidar images and supplied

by the Agency for Geo-spatial Information (BIG), Indonesia (Fig. 2c,d).

For tsunami inundation modelling in a densely populated area, we apply a resistance law with the composite equivalent roughness coefficient depending on the land use and building conditions, as shown in Eq. (8) (Aburaya and Imamura, 2002; Koshimura et al., 2009a).

$$n = \sqrt{n_0^2 + \frac{C_D}{2gd} * \frac{\theta}{100 - \theta} * D^{4/3}} \tag{8}$$

$n_o$ corresponds to the Manning's roughness coefficient ($n_o = 0.025$ s.m$^{-1/3}$), $C_D$ represents the drag coefficient ($C_D = 1.5$ (Federal Emergency Management Agency (FEMA), 2003)) and the constant $d$ signifies the horizontal scale of buildings (~15 m). $\theta$ is the building occupation ratio in percent (0-100 %) for each computational cell of 20 x 20 m$^2$ and 1 x 1 m$^2$ resolutions in Sunda Strait and Palu areas, respectively. $\theta$ is obtained by computing the building area over each pixel using GIS data. The computational cell corresponding to buildings can be inundated by the $n$

Manning coefficient through the term $D$, which represents the simulated flow depth (m). In the urban areas of Sunda Strait and Palu, the average occupation ratios are 24 % and 84 % respectively (Fig. 2b,d). In non-residential area, we set the Manning's roughness coefficients inland and on the seafloor to 0.03 and 0.025 respectively, which are typical values for vegetated and shallow water areas (Kotani, 1998).

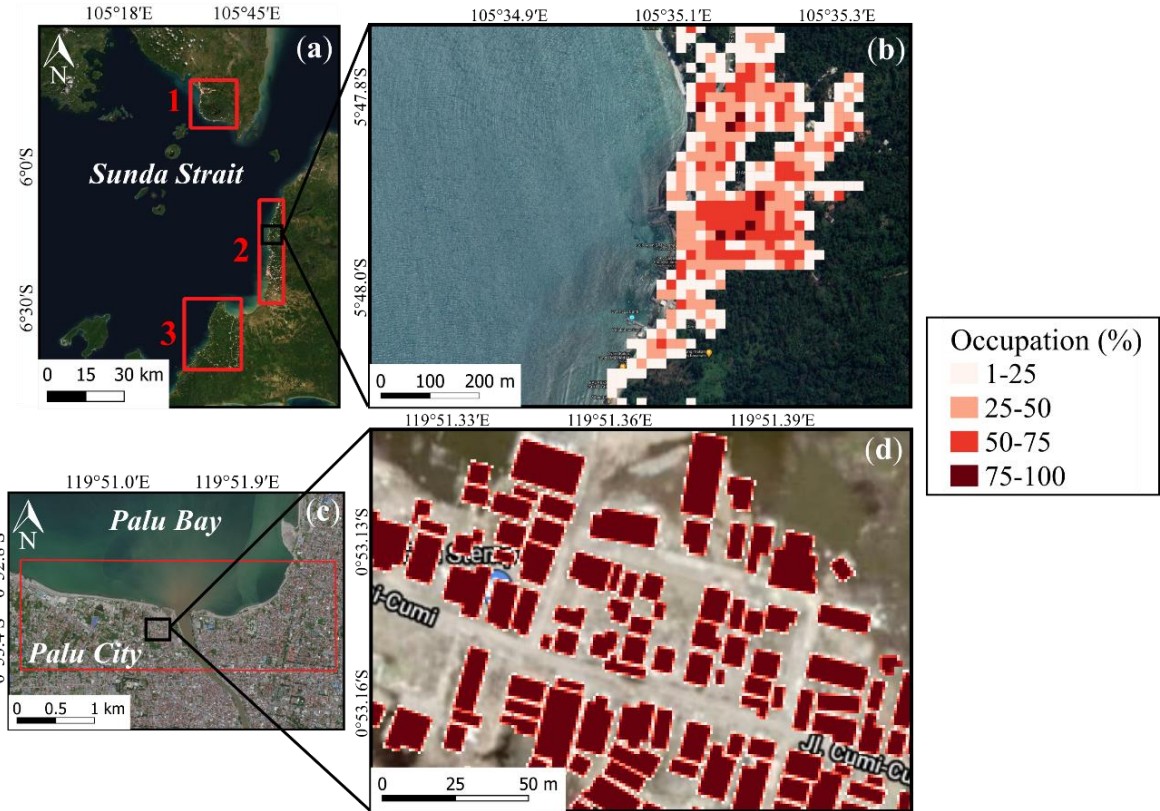

**Figure 2. (a,c) Computational areas in the Sunda Strait (1-3) and Palu-City, (b,d) magnified view of the building occupation ratio in the Sunda Strait (20-m resolution) and Palu-City (1-m resolution) (background ESRI and © Google Maps).**

## 3.2. Calibration and validation of the tsunami inundation model

### 3.2.1. Performance parameters

The tsunami inundation model is calibrated using two performances parameters: $K$ and $\kappa$ proposed by AIDA (1978), as defined below:

$$\log K = \frac{1}{n} \sum_{i=1}^{n} \log K_i \tag{9}$$

$$\log \kappa = \sqrt{\frac{1}{n} \sum_{i=1}^{n} (\log K_i)^2 - (\log K)^2} \tag{10}$$

$$K_i = \frac{x_i}{y_i} \tag{11}$$

$x_i$ and $y_i$ are the recorded and simulated tsunami flow depths at location $i$. $K$ is defined as the geometrical mean of $K_i$ and $\kappa$ is defined as deviation/variance from $K$. The Japan Society of Civil Engineers (JSCE) (2002) recommends $0.95 < K < 1.05$ and $\kappa < 1.45$ for the model results to achieve "good agreement" in the tsunami source model and propagation/inundation model evaluation (Otake et al., 2020; Pakoksung et al., 2018).

### 3.2.2. The 2018 Sunda Strait tsunami inundation model

To correct the Digital Surface Model (DSM), we removed the vegetation, buildings and infrastructures elevations based on the linear smoothing method and used the resulting Digital Elevation Model (1st DEM) as topography in the tsunami inundation model (Fig. 3). The vertical accuracy of the DSM/DEM is about 4 m. The 2018 Sunda Strait tsunami model depends on the density of the landslide ($\rho_2$), its stable slope ($\alpha$), its volume ($V_L$) and its sliding time ($t_S$). As proposed by Paris et al. (2020), the low sensitivity parameters are set as follows: $\rho_2 = 1500$ kg/m$^3$, $\alpha = 5°$ and $V_S = 0.15$ km$^3$. We reach the best fit between the simulated and observed flow depths at buildings for 10 min sliding time. Nevertheless, most of the simulated flow depths are underestimated compared to the observed ones, with a mean difference of 0.28 m ± 1 m. Using QGIS software, we smoothed the 1st DEM to remove these mean difference in elevation at buildings where the flow depth is underestimated. The resulting DEM (2nd DEM) provides a topography more reliable at buildings (Fig. 3). We realized three cross-sections along the Sunda Strait coasts to show the different corrections applied to the DSM (Fig. 4a-g). $K$ and $\kappa$ values for damaged buildings are 0.99 and 1.11, respectively, which means that we achieve "good agreement" for the Sunda Strait tsunami model, displayed in Fig. 5a-f. We note that the simulated inundation zone overlays 94 buildings out of 98. In section 4.1, the Sunda Strait tsunami fragility assessment is based on these 94 buildings (DB_Sunda2018). Simulation snapshots of the Sunda Strait tsunami propagation are shown in Fig. B1 10, 20, 60 and 120 s after the tsunami generation. In Figure B2, the simulated tsunami height based on the best-fitting parameters is also displayed. Figure B3 illustrates the maximum simulated flow velocity of the 2018 Sunda Strait tsunami inundation model.

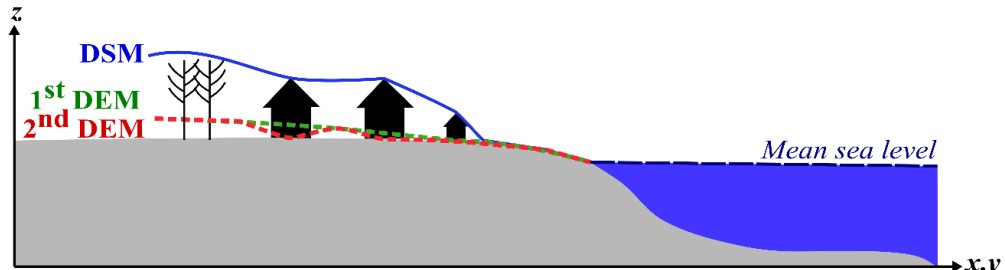

**Figure 3. Topographic corrections performed on the DSM and the 1st DEM. The 2nd DEM is used as new topography in TUNAMI two-layer model.**

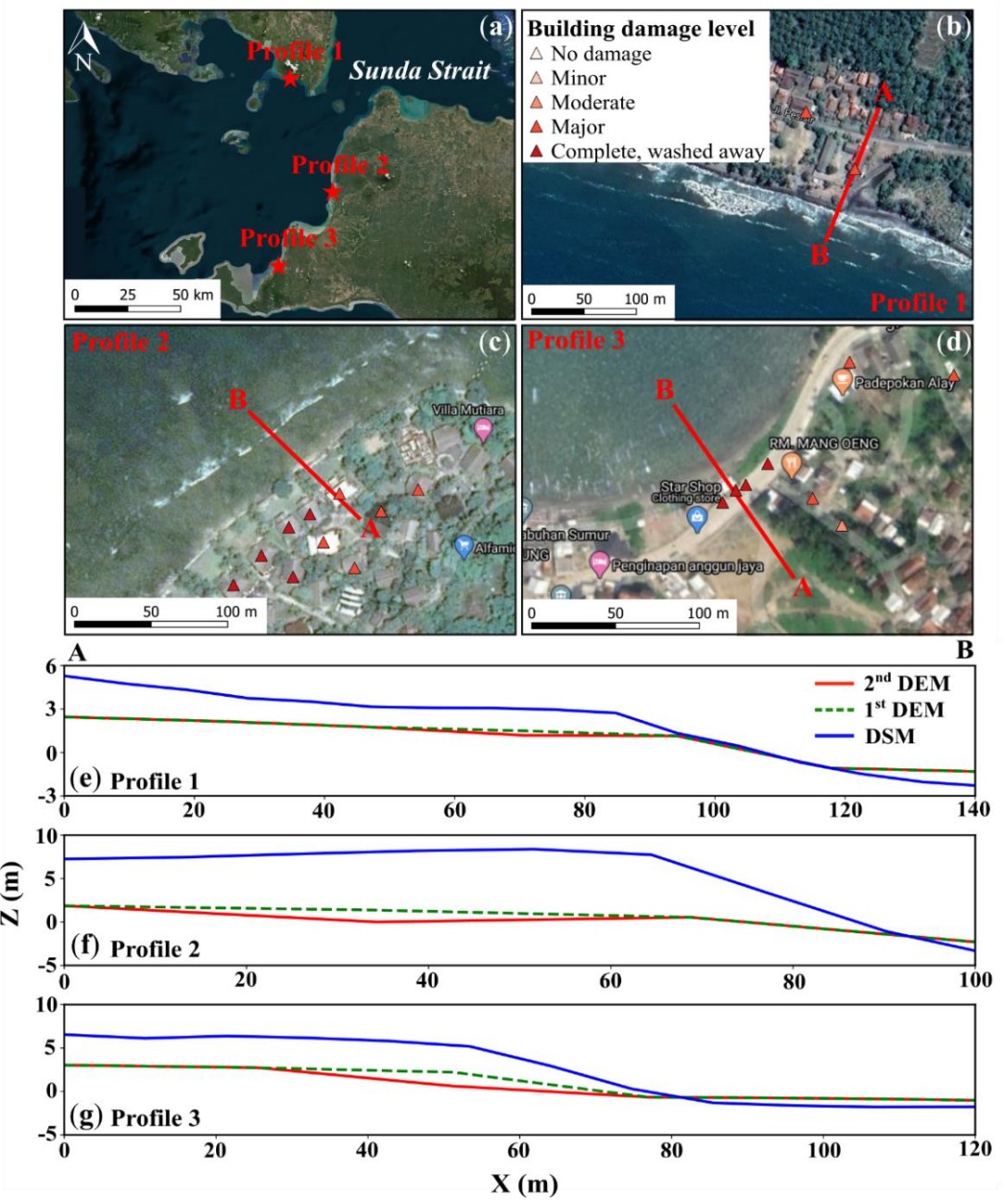

**Figure 4. (a) Cross-sections along Sunda Strait coasts. One cross-section is realized in the computational areas (b,e) 1, (c,f) 2 and (d,g) 3 to illustrate the topographic corrections applied to the DSM at buildings using QGIS (a triangle represents a building) (background ESRI and © Google Maps).**

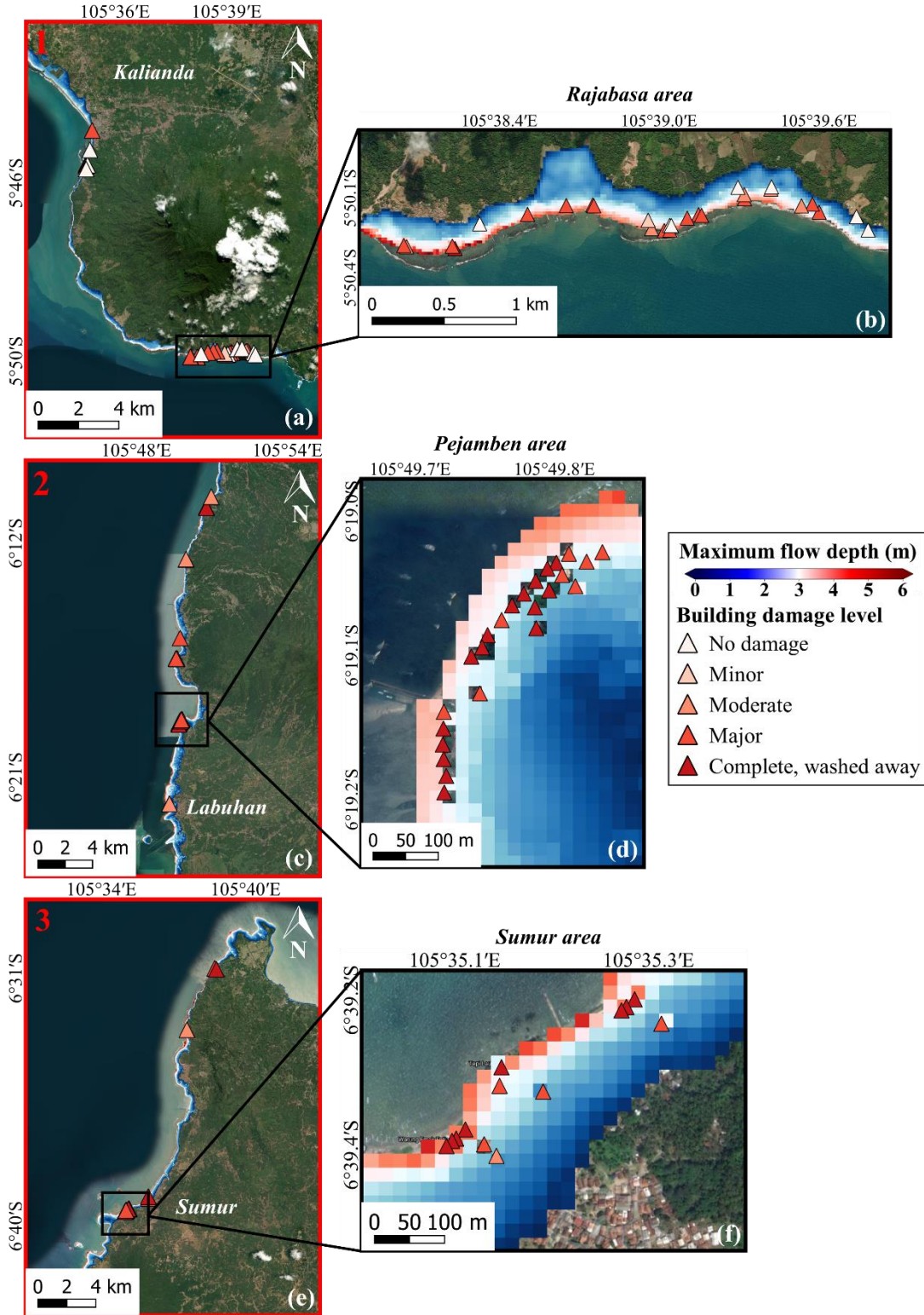

**Figure 5.** (a,c,e) Sunda Strait final tsunami inundation model with the maximum simulated flow depth overlaying on the damaged building data in the computational areas 1 to 3, (b,d,f) magnified views of the maximum simulated flow depth in Rajabasa, Pejamben and Sumur areas (background ESRI and © Google Maps).

### 3.2.3. The 2018 Sulawesi-Palu tsunami inundation model

We increased the mean sea level (MSL) by 2.3 m to reproduce the high tide during the 2018 Sulawesi-Palu tsunami. As shown by Pakoksung et al. (2019), the observed waveform at Pantoloan tidal gauge does not fit the simulated

one with the Finite Fault Model of TUNAMI-N2. Although recent studies show that seismic seafloor deformation may be the primary cause of the tsunami (Gusman et al., 2019; Ulrich et al., 2019), in this study, the main assumption is that the 2018 Sulawesi-Palu was triggered by subaerial/submarine landslides. According to Heidarzadeh et al. (2018), a large landslide to the north or the south of Pantoloan tidal gauge is responsible for the significant height wave recorded. Arikawa et al. (2018) also identified several sites of potential subsidence in the northern part of Palu-Bay. Based on these previous studies, we assume two large landslides: L1 and L2. Small landslides (S1-S12) also occurred in the bay; their location stands on observations from satellite imagery, field surveys and video footage (Arikawa et al., 2018; Carvajal et al., 2019) (Fig. 6). From the trial and error method and the topographic/bathymetric data provided by the Agency for Geo-spatial Information (BIG), we determined the soil property and achieved the volume of the landslides (Table 3). In Figure 7, the submarine landslides model reproduces well the tsunami observations at Pantoloan.

The calibration of the model depends on the landslide S8 because (i) as a small landslide, its volume is too small to distort the simulated wave height at the Pantoloan tidal gauge, (ii) it has the largest volume among the other small landslides and (iii) it is close and ideally oriented to Palu-City; the slide direction, captured by an aircraft pilot, is perpendicular to the bay (Carvajal et al., 2019). The density of the landslides ($\rho_2$), their stable slope ($\alpha$), their sliding time ($t_s$) are set as follows: $\rho_2 = 2000$ kg/m$^3$ (Palu bay receives a large amount of fine continental deposits such as clay-sized sediments (Frederik et al., 2019)), $\alpha = 14°$ (Chakrabarti, 2005) and $t_S = 10$ min. For a landslide ratio of 1.2 (i.e., S8 volume is multiplied by 1.2), the tsunami model shows a great similarity between observed and simulated flow depths (a = 1.027). The simulated tsunami inundation zone overlays 175 traces out of 371 because (i) 151 buildings with flow depth traces are not included in our computational area (Fig. 2c) and (ii) 45 buildings are outside the simulated tsunami envelope, which is shorter than the surveyed one (Fig. 8). The geometric mean is near the recommended values ($K = 0.93$) while the standard deviation as well as the Root Mean Square Error (RMSE) are high ($\kappa = 2.18$, RMSE = 0.92 m). Therefore, to develop accurate and reliable curves, we set a 1-m confidence interval including 124 flow depth traces at buildings out of 175 (Fig. 9). In section 4.2, the Sulawesi-Palu tsunami fragility assessment is based on these 124 buildings (DB_Palu2018). $K$ and $\kappa$ values for damaged buildings are 0.93 and 2.14 respectively, with a Root Mean Square Error of 0.26 m. The validity of the model is mainly based on the geometric mean $K$, close to 0.95, so we consider the tsunami inundation model accurate enough (Fig. 8). In Figure C1, the simulation snapshots of the Sulawesi-Palu tsunami propagation are shown 2, 10, 30, 60 s after the tsunami generation. The simulated tsunami height based on the best-fitting parameters is also displayed in Fig. C2. Figure C3 illustrates the maximum simulated flow velocity of the 2018 Sulawesi-Palu tsunami inundation model.

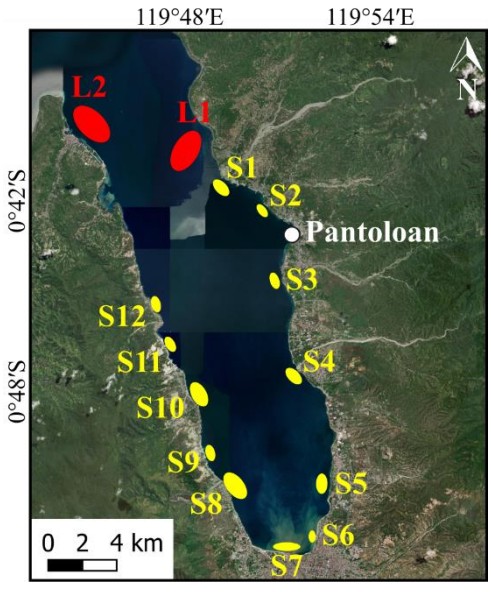

Figure 6. Location of the hypothesized landslides (S: small and L: large) in Palu-Bay (background ESRI).

Table 3. Hypothesized landslide parameters (location and volume) in Palu-Bay.

| No. | Location (latitude; longitude) | Volume ($10^6\,m^3$) |
|---|---|---|
| L1[a] | -0.655;119.749 | 37.54 |
| L2[a] | -0.670;119.801 | 31.93 |
| S1[b] | -0.680;119.821 | 0.60 |
| S2[b] | -0.703;119.842 | 0.18 |
| S3[b] | -0.737;119.851 | 0.25 |
| S4[b] | -0.789;119.862 | 0.75 |
| S5[b] | -0.852;119.878 | 0.22 |
| S6[b] | -0.879;119.871 | 0.60 |
| S7[b] | -0.885;119.858 | 2.44 |
| S8[b] | -0.846;119.822 | 4.45 |
| S9[b] | -0.832;119.813 | 0.83 |
| S10[b] | -0.804;119.808 | 2.17 |
| S11[b] | -0.774;119.792 | 0.55 |
| S12[b] | -0.754;119.788 | 0.83 |

a: based on our assumption from Arikawa et al. (2018) and Heidarzadeh et al. (2018)

b: based on observations from satellite imagery, field surveys and video footage (Arikawa et al., 2018; Carvajal et al., 2019).

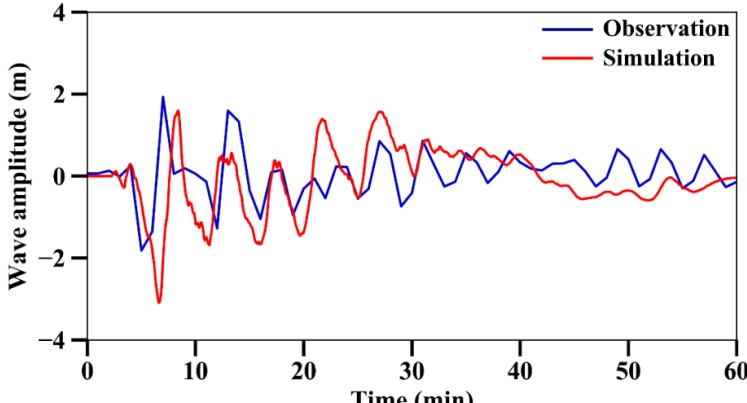

**Figure 7. Comparison between observed and simulated wave heights at Pantoloan tidal gauge, in Palu-Bay, Sulawesi, Indonesia.**

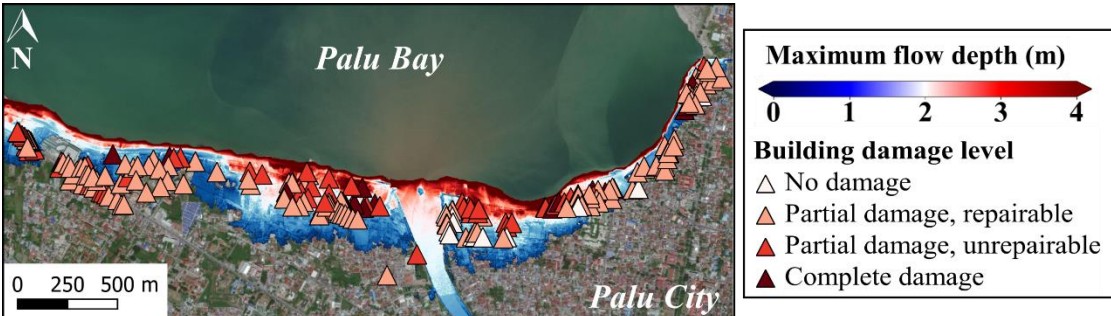

**Figure 8. Sulawesi-Palu final tsunami inundation model with the maximum simulated flow depth overlaying on the damaged building data (background ESRI).**

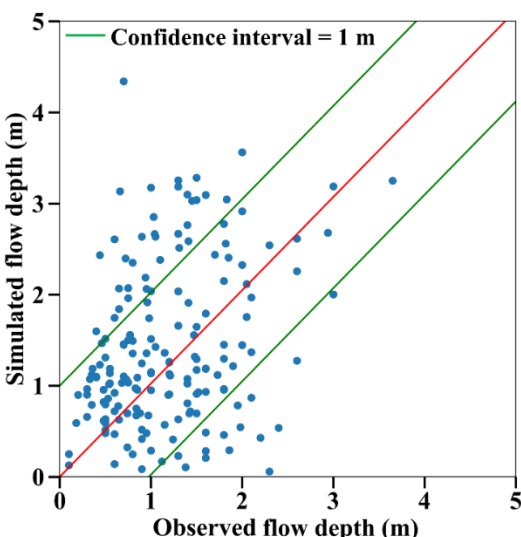

**Figure 9. Comparison between observed and simulated flow depths at damaged building for a S8 ratio of 1.2; a confidence interval is set at 1-m flow depth.**

## 4.  Tsunami fragility assessment

The proposed fragility assessment framework has two main steps. In the first step, an exploratory analysis aims to
(i) assess the trends that the available data follow and (ii) determine the main explanatory variables that need to be

included in the statistical model and their influence on the slope and intercept of the fragility curves. Then, we select a statistical model and examine its goodness-of-fit to the data based on the observations of the exploratory analysis. We note that the development of the computed fragility curves for the 2018 Sunda Strait and 2018 Sulawesi-Palu tsunamis is directly based on DB_Sunda2018 and DB_Palu2018, in which each building has both observed and simulated flow depth values (Table 4).

**Table 4. Number of buildings used for the tsunami fragility analysis of the 2018 Sunda Strait, 2018 Sulawesi-Palu and 2004 IOT (Khao Lak/Phuket) events.**

| Database | Tsunami intensity measure | | | |
| --- | --- | --- | --- | --- |
| | Observed flow depth | Simulated flow depth | Simulated flow velocity | Simulated hydrodynamic force |
| DB_Sunda2018[a] | 94 | 94 | 94 | 94 |
| DB_Palu2018[b] | 124 | 124 | 124 | 124 |
| DB_Thailand2004 | 117 | / | / | / |

[a]surveyed buildings included in the Sunda Strait simulated tsunami inundation zone.

[b]surveyed buildings included in the Palu simulated tsunami inundation zone and in the 1-m confidence interval.

To explore the relationship between the tsunami intensity and the probability of damage, we fit a Generalised Linear Model (GLM) to the data of each database, as proposed by the GEM guidelines (Rossetto et al., 2014). A GLM assumes that the response variable $y_{ij}$ is assigned 1 if the building $j$ sustained damage $DS \geq ds_i$ and 0 otherwise. The variable follows a Bernoulli distribution:

$$y_{ij} \sim Bernoulli\ (\pi_i(\tilde{x}_j)) \tag{12}$$

where $\pi_i(\tilde{x}_j)$ is the probability that a building $j$ will reach or exceed the 'true' damage state $ds_i$ given estimated tsunami intensity level $\tilde{x}_j$. The Bernoulli distribution is characterised by its mean:

$$\mu_{ij} = \pi_i(\tilde{x}_j) \tag{13}$$

which is expressed here in terms of a probit model, commonly used to express the mean in the empirical fragility assessment field (Rossetto et al., 2013), defined in terms of $\Phi[.]$, the cumulative distribution function of a standard normal distribution:

$$\Phi^{-1}\big[\pi_i\big(\tilde{x}_j\big)\big] = \eta_{ij} \tag{14}$$

where $\eta_{ij}$ is the linear predictor, which can be written in the form:

$$\eta_{ij} = \theta_{0i} + \theta_{1i}\ln(\tilde{x}_j) \tag{15}$$

where $\theta_{1i}$, $\theta_{0i}$ are the two regression coefficients, representing the slope and the intercept, respectively, of the fragility curve corresponding to damage state $ds_i$. For the exploratory analysis, the tsunami intensity is measured in terms of observed flow depth levels. We also fit the GLM models to subsets of data of each database to explore the importance of the construction type to the shape of the fragility curves. The confidence in the exact shape of the mean curves is estimated and presented in terms of the 90 % confidence intervals around the best-estimate

curves.

Based on the aforementioned observations, we construct parametric statistical models for the three databases to (i) identify the simulated tsunami measure type that fits the data best and (ii) to construct fragility curves for the tsunami intensity type that fits the data best.

Ideally, the response variable $y_{ij}$ of an appropriate statistical model is the damage state $i = \{0, 1, 2, 3\}$ sustained by a building $j$. The damage state follows a categorical distribution (i.e. also called a generalized Bernoulli distribution) which describes the possible levels of damage $i = \{0, 1, 2, 3\}$ sustained by a given building (Table 1). The random component of this model can be written as:

$$y_{ij} \sim Categorical\left(P(DS = ds_i|\tilde{x}_j)\right) \tag{16}$$

where $P(DS = ds_i|\tilde{x}_j)$ is the probability that a building $j$ will reach the 'true' damage state $ds_i$ given estimated tsunami intensity level $\tilde{x}_j$:

$$P(DS = ds_i|\tilde{x}_j) = \begin{cases} 1 - \pi_i(\tilde{x}_j), & i = 0 \\ \pi_i(\tilde{x}_j) - \pi_{i+1}(\tilde{x}_j), & 0 < i < i_{max} \\ \pi_i(\tilde{x}_j), & i = i_{max} \end{cases} \tag{17}$$

Multiple expressions of the systematic component are constructed to test their goodness of fit. With regard to the link function, apart from the commonly used probit function, two alternative expressions found in the GEM guidelines for empirical vulnerability assessment (Rossetto et al., 2013) namely the logit and complementary loglog (termed here 'cloglog') are considered in the form:

$$\eta_{ij} = \begin{cases} \Phi^{-1}\left[\pi_i(\tilde{x}_j)\right], & probit \\ ln\left(\dfrac{\pi_i(\tilde{x}_j)}{1 - \pi_i(\tilde{x}_j)}\right), & logit \\ ln\left(-ln\left(1 - \pi_i(\tilde{x}_j)\right)\right), & cloglog \end{cases} \tag{18}$$

The linear predictor is also expressed in various forms of increasing complexity, as depicted in Eq.(19):

(19.1)

(19.2)

(19.3)

$$
\eta_{ij} = \begin{cases}
\theta_0 + \theta_1 \tilde{x}_j & (19.4) \\
\theta_0 + \theta_{1i} \tilde{x}_j & \\
\theta_0 + \theta_1 \tilde{x}_j + \theta_2 class & \\
\theta_0 + \theta_{1i} \tilde{x}_j + \theta_2 class & (19.5) \\
\theta_0 + \theta_1 \tilde{x}_j + \theta_2 class + \theta_3 \tilde{x}_j class &
\end{cases}
$$

where *class* is a categorical unordered variable which expresses here the construction type. $\theta_{0\text{-}3}$ are the unknown
regression coefficients of the model. Eq.(19.1) and Eq.(19.2) assume that the fragility curves are only influenced by the tsunami intensity. Eq.(19.1) assumes that the slope of the fragility curves is the same for all damage states. By contrast, Eq.(19.2) allows the slope of each curve to vary for each damage state; the slope varies for each fragility curve. The following three equations account for the influence of the building class (i.e. the construction type) in the shape of the fragility curves. All three equations assume that the construction type affects the intercept of the fragility curves and only Eq.(19.5) assumes that the construction type affects both the intercept and the slope of the curves. Finally, Eq.(19.3) and Eq.(19.5) assume identical slopes for all fragility curves irrespective of the damage state. By contrast, Eq.(19.4) relaxes this assumption and considers that the slope changes for each damage state. The combinations of random and systematic components result in five distinct models (Table 5).

**Table 5. Statistical models examined for each database.**

| Model | Component | |
|:---:|:---:|:---:|
| | **Random** | **Systematic** |
| **M1** | | Eq.(19.1) |
| **M2** | | Eq.(19.2) |
| **M3** | Eq.(16) | Eq.(19.3) |
| **M4** | | Eq.(19.4) |
| **M5** | | Eq.(19.5) |

In what follows, we fit multiple models to each database based on the observations of the exploratory analysis. We examine the goodness of fit of these models for a given tsunami intensity measure and link function with two formal tests, as proposed in the GEM guidelines (Rossetto et al., 2014). Firstly, we compare the Akaike Information Criterion (AIC) values, which estimates the prediction error of the examined models (Akaike, 1974). The model with the lowest value fits the data best. The alternative models used in this study are nested, which means that the more complex model includes all the terms of the simpler ones plus an additional term. For this reason, we also perform a series of likelihood ratio tests to examine whether the fit provided by the model with the lowest AIC value is statistically significant over its alternative nested models, which relaxes its assumptions (Rossetto et al., 2014). We also use the AIC value to determine which of these simulated intensity measures fits the data best. Furthermore, the 90 % confidence intervals of the best-estimate fragility curves are constructed using bootstrap analysis. According to the latter analysis, 1000 samples of the database are obtained with replacement and the selected model is refitted to each sample.

## 4.1. DB_Sunda2018

We fit the GLM models to the data in DB_Sunda2018 (irrespective of their structural characteristics) and we plot the obtained probit functions against the natural logarithm of the observed flow depth to explore how the slope and the intercept of the models change for each damage state (Fig. 10a). The 90 % confidence intervals around the best-estimate curves are also included. All three curves have positive slopes, which indicates that the flow depth is an adequate descriptor of the damage caused by a tsunami, as the probability of a given damage state being reached or exceeded increases with the increase in the flow depth. The slope of each function is similar for $ds_2$ and $ds_3$ and different for $ds_1$. Nonetheless, the curve corresponding to $ds_1$ is also associated with substantial uncertainty. In Figure 10b, we fit probit models to subsets of the available data for the two main construction types. One of the drawbacks of the small database is that not all damage states were observed for each building class. Therefore, the comparison of probit models is limited for damage states $ds_2$ and $ds_3$. The curves for the two construction types appear to be substantially different. As expected, the timber buildings are more vulnerable than the confined masonry buildings. Their intercept is responsible for the difference, as the two curves are parallel. It indicates the need to develop a statistical model, which allows only the intercept to change with the construction type and the slope should be identical.

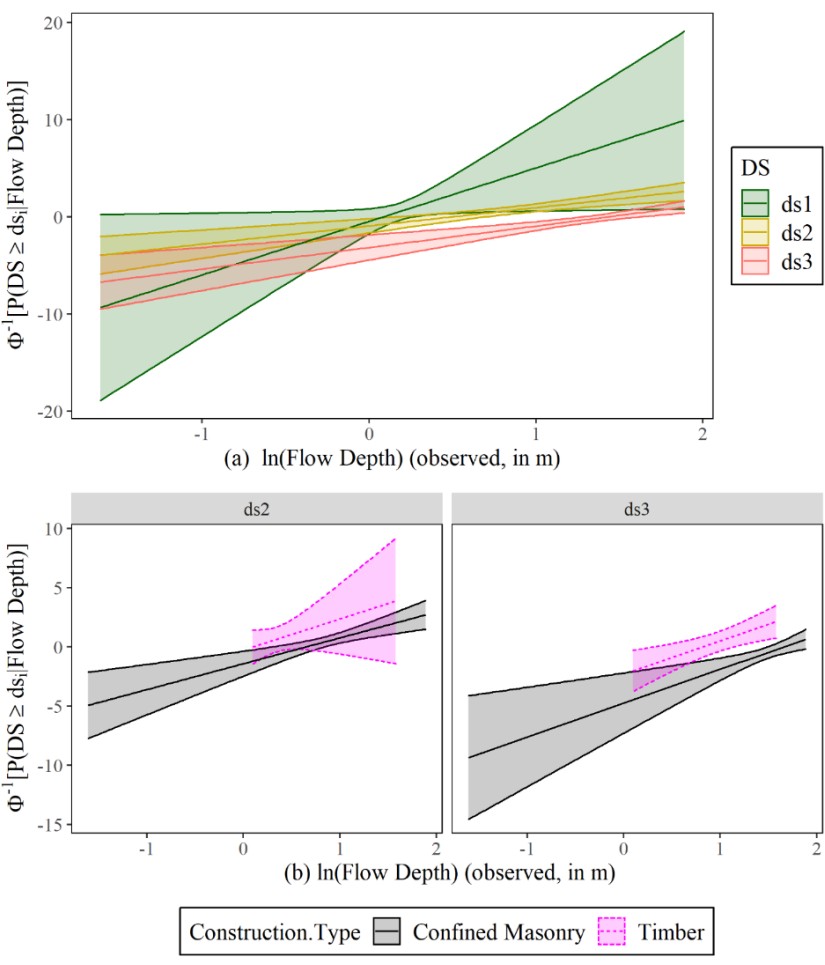

**Figure 10. Probit functions fitted for each individual damage state to (a) DB_Sunda2018 to assess whether the observed flow depth is an efficient descriptor of damage; (b) to assess whether the construction type affected the shape of fragility curves for $ds_2$ and $ds_3$. In both cases, the 90 % confidence interval is plotted.**

Following the main observations of the exploratory analysis, we consider that M3 is an acceptable model with two explanatory variables: the tsunami intensity and the construction type. To assess its goodness of fit, we consider each link function with three alternatives for the linear predictor (i.e., M4, M5 and M1), which relax some of its assumptions. In Table 6, we compare the AIC values of the three models to assess the fit of the different models for the observed flow depth levels assuming the probit link function. M3 has the smallest AIC value than its alternatives, which indicates that it fits the data better than the remaining three models. Nonetheless, some of these differences are rather small and it raises the question of whether the improvement in the fit provided by M3 is statistically significant over its alternatives. To address this, we perform likelihood ratio tests and the results are reported in Table 7. We note that the $p$-values vary for the three comparisons. The $p$-value is significantly above the 0.05 threshold when the identical slope for each fragility curve assumption (i.e. comparison of 'M3' and 'M4') is tested. This means that the M4 (which assumes varying slopes for each damage state) does not provide a statistically significant improvement than its alternative. Therefore, the fit of M3 is the best. Similarly, the $p$-value is well-above the threshold for M3 vs M5, highlighting that the construction type does not affect the slope of the fragility curves. By contrast, the $p$-value is well below the threshold for the comparison of M3 and M1, indicating that the construction type is an important variable and affects only the intercept. Having concluded that M3 based on the observed flow depth data fits the data better than its alternatives (i.e. M4, M5 and M1), we repeat the procedure to identify which simulated intensity type fits the data best. Table 6 also shows the comparison of the AIC values for the three simulated tsunami intensity types. For all simulated intensity types, M3 is identified as the model which fits the data better than its alternatives and this conclusion is further reinforced by the likelihood ratio tests presented in Table 7. By comparing the AIC values for M3 for all three simulated intensity types, we note that the simulated flow depth is the tsunami intensity that fits the data best. The aforementioned observations can also be made if instead of the probit link function, the two alternative functions (i.e. logit and cloglog) are considered, as depicted in Table D1. The comparison of the AIC values of M3 for the three link functions identifies the probit link function as the one that fits the data best.

The regression coefficients of the 2018 Sunda Strait fragility curves based on the best fitted M3 model with a probit link function are listed in Table E1. An advantage of constructing a complex model that accounts for the ordinal nature of the damage and for the two main construction types in the systematic component is that fragility curves for timber buildings can be obtained even for the states for which there is available data. A timber building is found to sustain more damage than confined masonry buildings for the more intense damage states. Nonetheless, there is substantial more uncertainty in the prediction of the likelihood of damage and this can be attributed to the rather small sample size.

**Table 6. AIC values for the three models assuming probit link function fitted to the observed and simulated tsunami intensity measures of DB_Sunda2018.**

| Model | AIC | | | |
|---|---|---|---|---|
| | Observed flow depth | Simulated flow depth | Simulated flow velocity | Simulated hydrodynamic force |
| M3 | 129.9 | 138.5 | 224.2 | 194.9 |
| M4 | 137.7 | 148.4 | 227.7 | 210.3 |

| Model | | | | |
|---|---|---|---|---|
| **M5** | 131.6 | 139.8 | 225.3 | 196.1 |
| **M1** | 162.0 | 169.0 | 246.5 | 216.9 |

**Table 7. Likelihood ratio tests summary for all available observed and simulated tsunami intensity measures of DB_Sunda2018.**

| Model | *p*-value | | | |
|---|---|---|---|---|
| | **Observed flow depth** | **Simulated flow depth** | **Simulated flow velocity** | **Simulated hydrodynamic force** |
| **M3** | | | | |
| ——— | ~0.41 | ~0.72 | ~0.08 | ~0.05 |
| **M4** | | | | |
| **M3** | | | | |
| ——— | ~0.56 | ~0.39 | ~0.36 | ~0.35 |
| **M5** | | | | |
| **M3** | | | | |
| ——— | ~0.00 | ~0.00 | ~0.00 | ~0.00 |
| **M1** | | | | |

### 4.2. DB_Palu2018

We also fit GLM models to the data in DB_Palu2018 using the observed tsunami flow depth to express the tsunami intensity and then, to construct fragility curves and their 90 % confidence intervals for the three individual damage states (Fig. 11). The data seems to produce fragility curves with positive slopes for $d_{S1}$ and $d_{S2}$ and a negative slope

for $d_{S3}$. This latter observation is counter-intuitive as it is expected the likelihood of collapse to grow with the increase of the tsunami depth. This outcome could be attributed to the collected sample, which includes very few collapsed buildings observed at low flow depth levels.

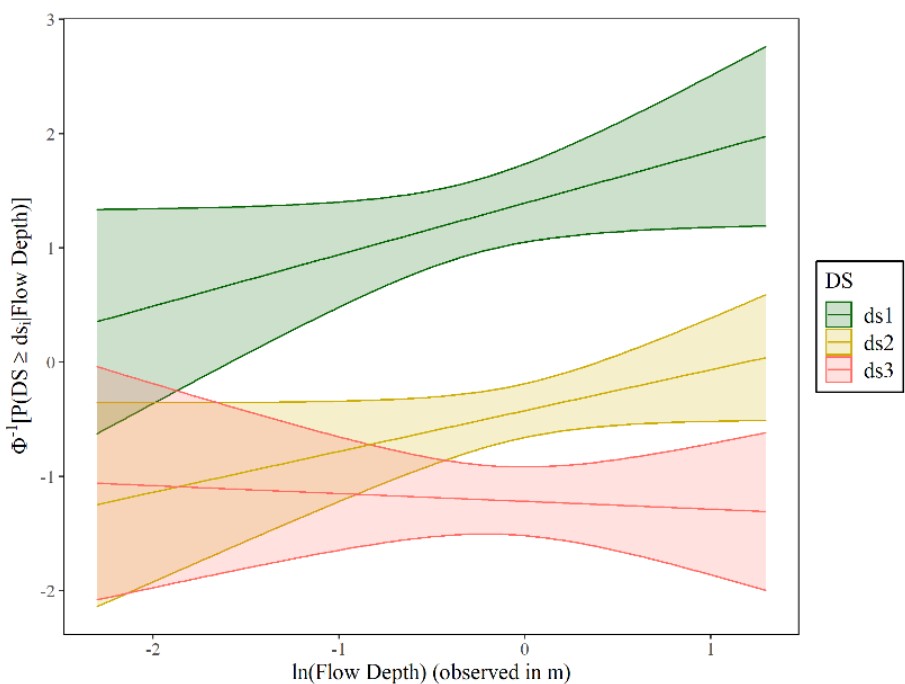

**Figure 11. Probit functions fitted for each individual damage state to DB_Palu2018 to assess whether the observed flow depth is an efficient descriptor of damage. The 90 % confidence interval are plotted.**

Based on the observations of the exploratory analysis, we use identical slopes for the fragility curves for all three damage states ($ds_1$-$ds_3$) to tackle the negative slope for $ds_3$ and three link functions. Therefore, model M1 is fitted to DB_Palu2018 assuming that the tsunami intensity is expressed in terms of simulated flow depth, flow velocity and hydrodynamic force. Table 8 depicts the AIC values for each model. We note that for all cases the flow depth fits the data the best. Table 8 also shows that the logit function fits the data best. The regression coefficients of the 2018 Sulawesi-Palu fragility curves for the logit function are depicted in Table E2.

**Table 8. AIC values for model M1 fitted to the simulated tsunami intensity measures of DB_Palu2018.**

| Link function | Model | AIC | | |
| --- | --- | --- | --- | --- |
| | | Simulated flow depth | Simulated flow velocity | Simulated hydrodynamic force |
| **probit** | M1 | 276.8 | 286.3 | 283.3 |
| **logit** | M1 | **276.2** | **286.3** | **283.1** |
| **cloglog** | M1 | 280.3 | 286.5 | 284.7 |

### 4.3. DB_Thailand2004

The exploratory analysis aims to identify trends in the shape of the fragility curves for each damage state. Thus, we fit GLM models to DB_Thailand2004 to construct fragility curves for the three individual damage states and we plot them with their 90 % confidence interval in Fig. 12. The data seems to produce fragility curves with positive slopes for all three damage states and also are parallel to each other, which suggests that the slope should be identical for all three curves.

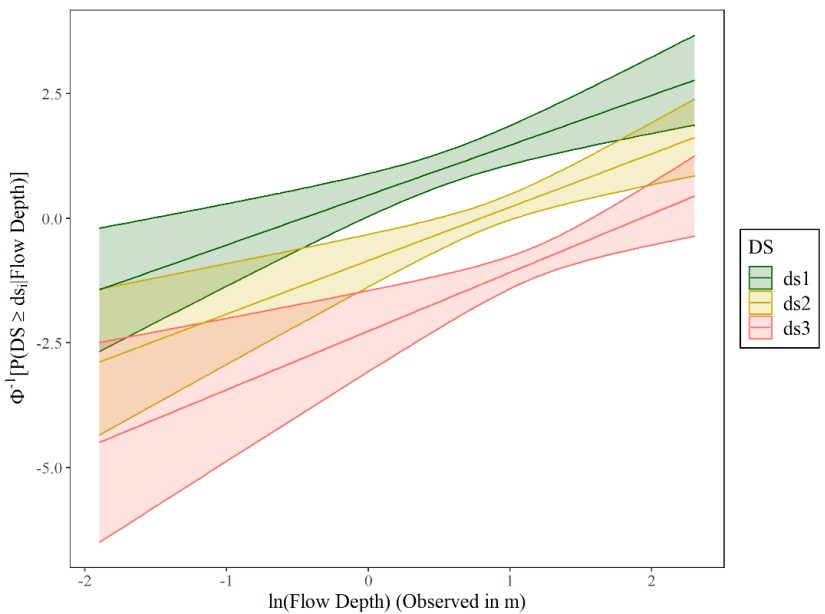

**Figure 12. Probit functions fitted for each individual damage state to DB_Thailand2004 to assess whether the observed flow depth is an efficient descriptor of damage. The 90 % confidence interval are plotted.**

Based on the observations of the exploratory analysis, we consider model M1 as the most suitable. To test its goodness of fit, model M2, which relaxes the assumption that the slope of all three curves is identical, is also fitted

to the data. In Table 9, the comparison of the AIC values for the two models also shows that M1 is the model which fits the data best for all three link functions considered in this study (i.e., probit, logit and cloglog). We also perform a likelihood ratio test to confirm that the improvement in the fit provided by the more complex M2 model over M1 is not statistically significant. The *p*-value is found to be equal to 0.76, 0.95 and 0.33 for the probit, logit and cloglog functions, respectively, which is significantly above the 0.05 threshold. This suggests that M2 does not provide a statistically better fit to the data, therefore the less complex M1 model fits the data best. The regression coefficients of the 2004 Indian Ocean (Khao Lak/ Phuket) fragility curves for the best fitted model M1 with logit link function can be found in Table E3.

**Table 9. AIC values for the two models fitted to the observed flow depth of DB_Thailand2004.**

| Model | AIC | | |
|---|---|---|---|
| | **Observed flow depth** | | |
| **Link function** | **probit** | **logit** | **cloglog** |
| **M1** | **264.3** | **262.4** | **263.5** |
| **M2** | 267.8 | 266.3 | 265.3 |

## 5. Results

### 5.1 Building fragility curves of the 2018 Sunda Strait tsunami

The fragility curves determine conditional damage probabilities according to the tsunami intensity measures of the 2018 Sunda Strait event for both confined masonry concrete (Fig. 13a-c) and timber (Fig. 14a-c) buildings of DB_Sunda2018. In Figure 14a,b, there is no data to predict the shape of the curves between 0-1 m flow depth and 0-1 m/s flow velocity. The curves as a function of the observed flow depth reveal a great similarity with the ones based on the simulated flow depth from TUNAMI two-layer model (Figs. 13a,14a). For instance, when the observed and simulated flow depths reach 3 m, the likelihood of minor to major damage (i.e., $\geq ds_1, ds_2$) for both timber and confined masonry buildings is approximately 99 % (Fig. 14a,b). By contrast, the likelihood of complete damage (i.e., $\geq ds_3$) is 70 % for timber buildings and only 10 % for confined masonry buildings. Consequently, the tsunami functions based on observation and simulation are highly similar, which illustrates the accuracy and the reliability of the tsunami inundation model. The curves show that confined masonry-type buildings have higher performance than timber structures. When the flow depth is greater than 5 m and 2.5 m, the probability of complete damage is around 99 % for confined masonry and timber buildings respectively. We also compare the completely damaged/washed away fragility curve for confined-masonry buildings to Syamsidik et al., 2020, who developed the curve as a function of observed flow depth for these buildings, as depicted in Fig. 13a. Fragility curves representing complete damage/washed away are similar up to 4.5-m flow depth. Each curve estimates a 15 % building damage probability at 3.5-m flow depth. However, a few data points are available beyond 5 m in the Sunda Strait area. Therefore, the damage probability uncertainty is greater upon this value, hence the difference between our $ds_3$-curve and the one produced by Syamsidik et al. (2020). The curves as functions of the maximum simulated flow velocity and the hydrodynamic force are displayed in Figs. 13b,14b and Figs. 13c,14c for confined masonry concrete and timber buildings, respectively.

- **The 2018 Sunda Strait curves for confined masonry concrete buildings**

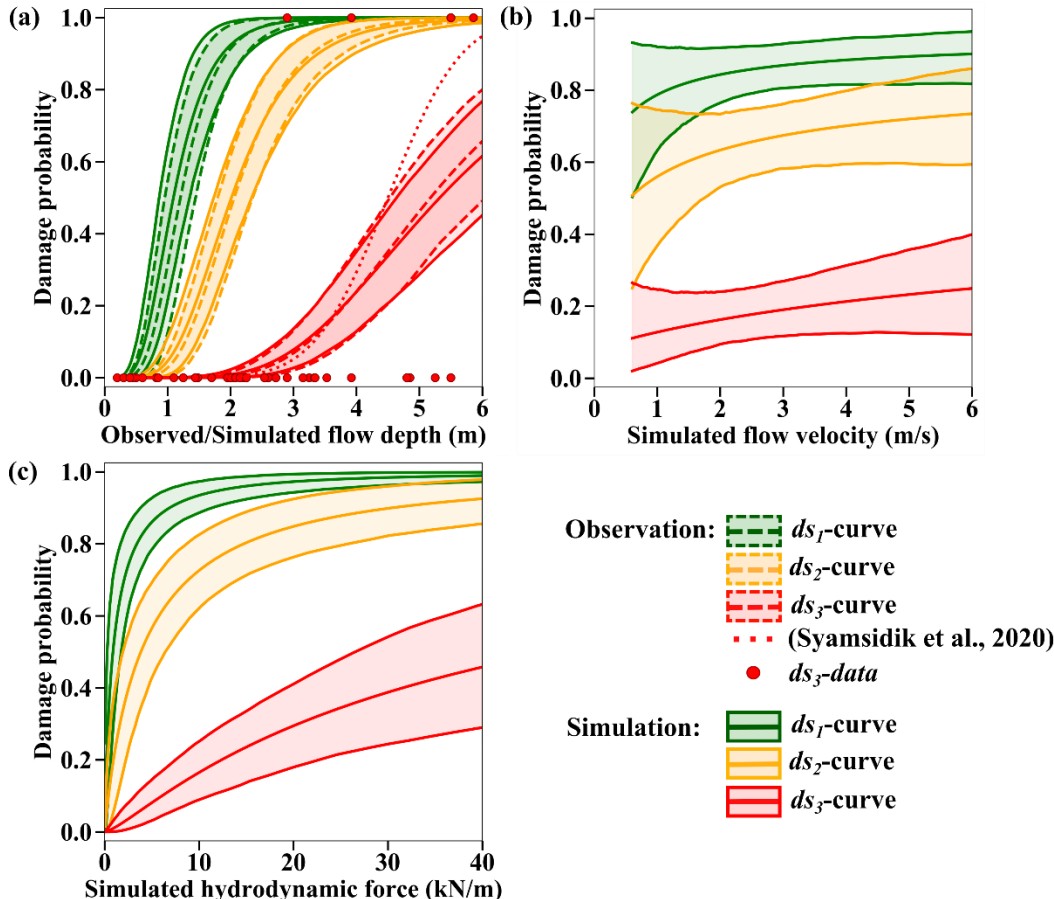

**Figure 13. Best-estimate fragility curves, with their 90 % confidence intervals, as functions of (a) the observed and the maximum simulated flow depths, (b) the maximum simulated flow velocity and (c) the simulated hydrodynamic force for confined masonry concrete buildings of DB_Sunda2018 sustaining minor/moderate damage ($ds_1$), major damage ($ds_2$) and complete damage/washed away ($ds_3$) in Sunda Strait area.**

- **The 2018 Sunda Strait curves for timber buildings**

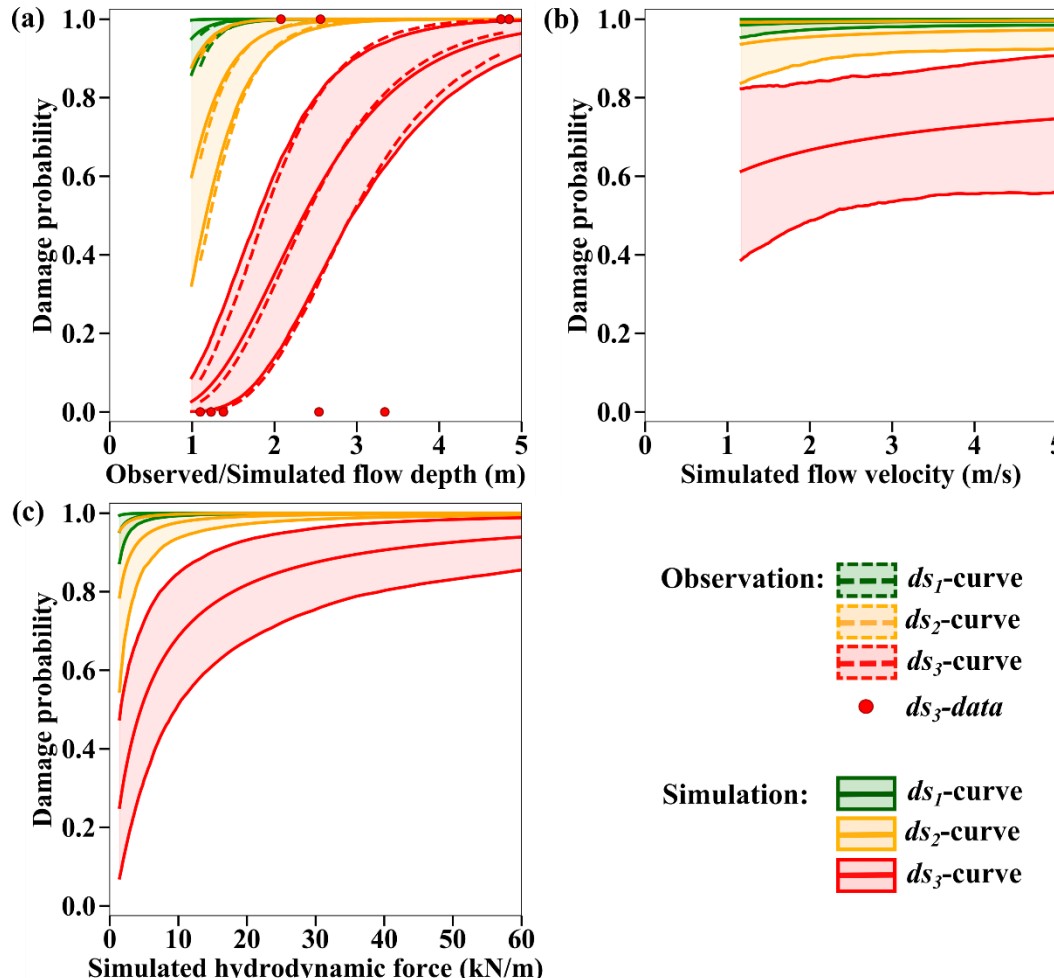

**Figure 14.** Best-estimate fragility curves, with their 90 % confidence intervals, as functions of (a) the observed and the maximum simulated flow depths, (b) the maximum simulated flow velocity and (c) the simulated hydrodynamic force for timber buildings of DB_Sunda2018 sustaining minor/moderate damage ($ds_1$), major damage ($ds_2$) and complete damage/washed away ($ds_3$) in Sunda Strait area.

**5.2 Building fragility curves of the 2018 Sulawesi-Palu tsunami**

The 2018 Sulawesi-Palu tsunami curves are developed for confined masonry buildings with unreinforced clay brick of DB_Palu2018. The computed and surveyed curves show a similar damage trend. When the observed and simulated flow depths reach 1.5 m, the building damage probabilities for partial damage repairable (i.e., $\geq ds_1$), partial damage unrepairable (i.e., $\geq ds_2$) and complete damage (i.e., $\geq ds_3$) are around 90 %, 40 % and 15 % respectively (Fig. 15a). The fragility curves based on the observed and simulated flow depths are relatively similar, especially for $ds_1$ and $ds_3$. The curves based on the flow velocity and the hydrodynamic force are displayed in Fig. 15b,c.

- **The 2018 Sulawesi-Palu curves for confined masonry buildings**

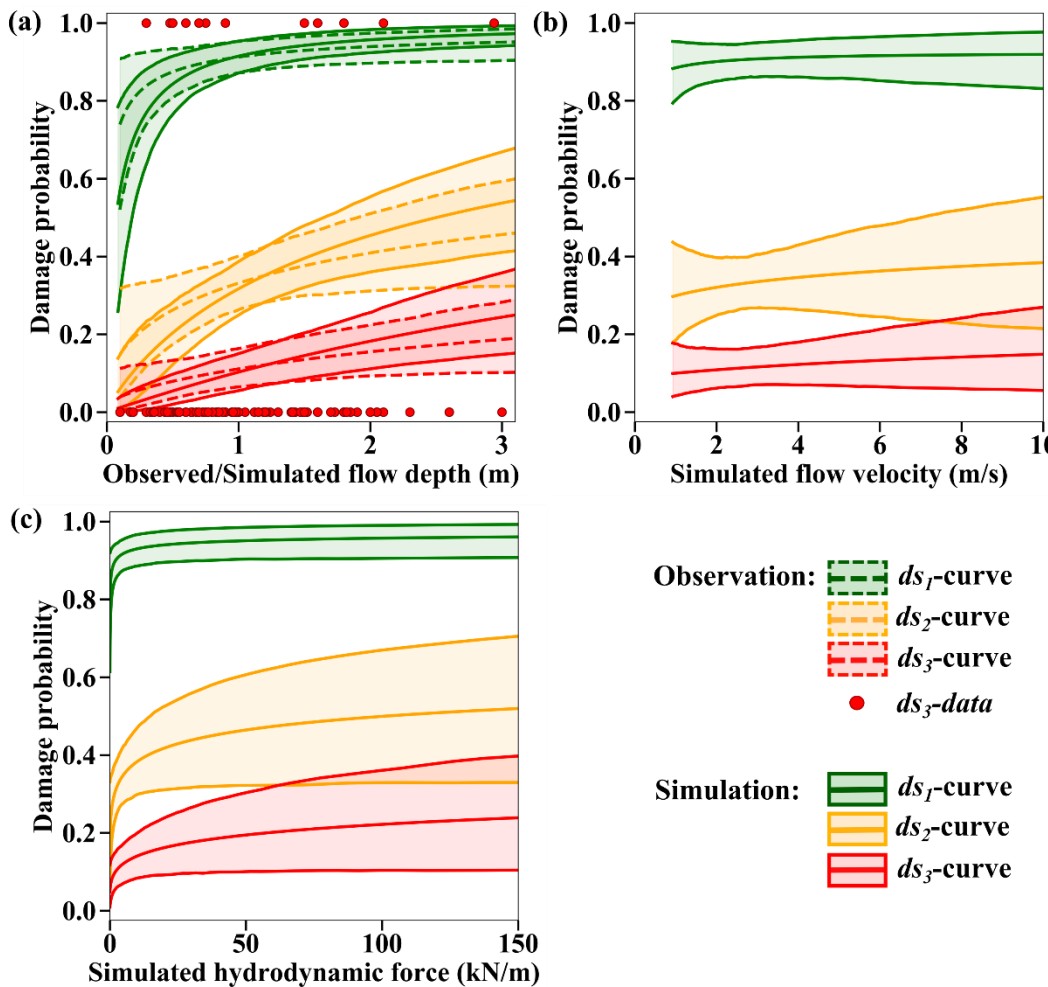

**Figure 15. Best-estimate fragility curves, with their 90 % confidence intervals, as functions of (a) the observed and the maximum simulated flow depths, (b) the maximum simulated flow velocity and (c) the simulated hydrodynamic force for confined masonry buildings with unreinforced clay brick of DB_Palu2018 sustaining partial damage repairable ($ds_1$), partial damage unrepairable ($ds_2$) and complete damage ($ds_3$) in Palu-City.**

### 5.3  Comparison between the 2018 and 2004 building fragility curves

In Figure 16, we compare (i) the Sunda Strait and Sulawesi-Palu $ds_3$-curves based on the simulated tsunami intensity measures for confined masonry-type buildings, (ii) the 2004 Indian Ocean (Khao Lak/Phuket, Thailand) $ds_3$-curve based on the observed flow depth for reinforced-concrete infilled frames buildings (Foytong and Ruangrassamee, 2007; Rossetto et al., 2007; Ruangrassamee et al., 2006) and (iii) the 2004 Indian Ocean (Banda Aceh, Indonesia) $ds_3$-curves produced by Koshimura et al. (2009a). The curves are based on a visual damage

interpretation of remaining roofs using the pre and post-tsunami satellite data (IKONOS) and are thus developed for mixed buildings (low-rise wooden, timber-framed and non-engineered reinforced-concrete constructions (Koshimura et al., 2009a; Saatcioglu et al., 2006)). For 1-m flow depth, the likelihood of complete damage is greater in Palu (10 %) than in Banda Aceh, Khao Lak/Phuket and Sunda Strait (Fig. 16a, Table 10). However, when the flow depth reaches 3 m, the damage probability is about 50 % in Banda Aceh, 25 % in Palu-city and less

than 20 % in Khao Lak/Phuket. We also note that the likelihood of completely damaged/washed away buildings is higher in Sunda Strait than in Khao Lak/Phuket above 4-m flow depth. However, the data points in Thailand are

mostly ranging from 0 to 5 m and the 90 % confidence interval upon this value is constantly increasing with the flow depth. Below 1 m/s, the flow velocity has a low impact on the damage probability in Banda Aceh (< 1 %). However, beyond this value, the probability of damage becomes very sensitive to the current velocity (Fig. 16b, Table 10). As an example, when the flow velocity attains 6 m/s, the curve estimates 99 % building damage probability in Banda Aceh. The hydrodynamic force also contributes to increase the probability of complete damage in Banda Aceh. For example, when the force reaches 25 kN/m, the damage probability is around 99 % in Banda Aceh (Fig. 16c, Table 10).


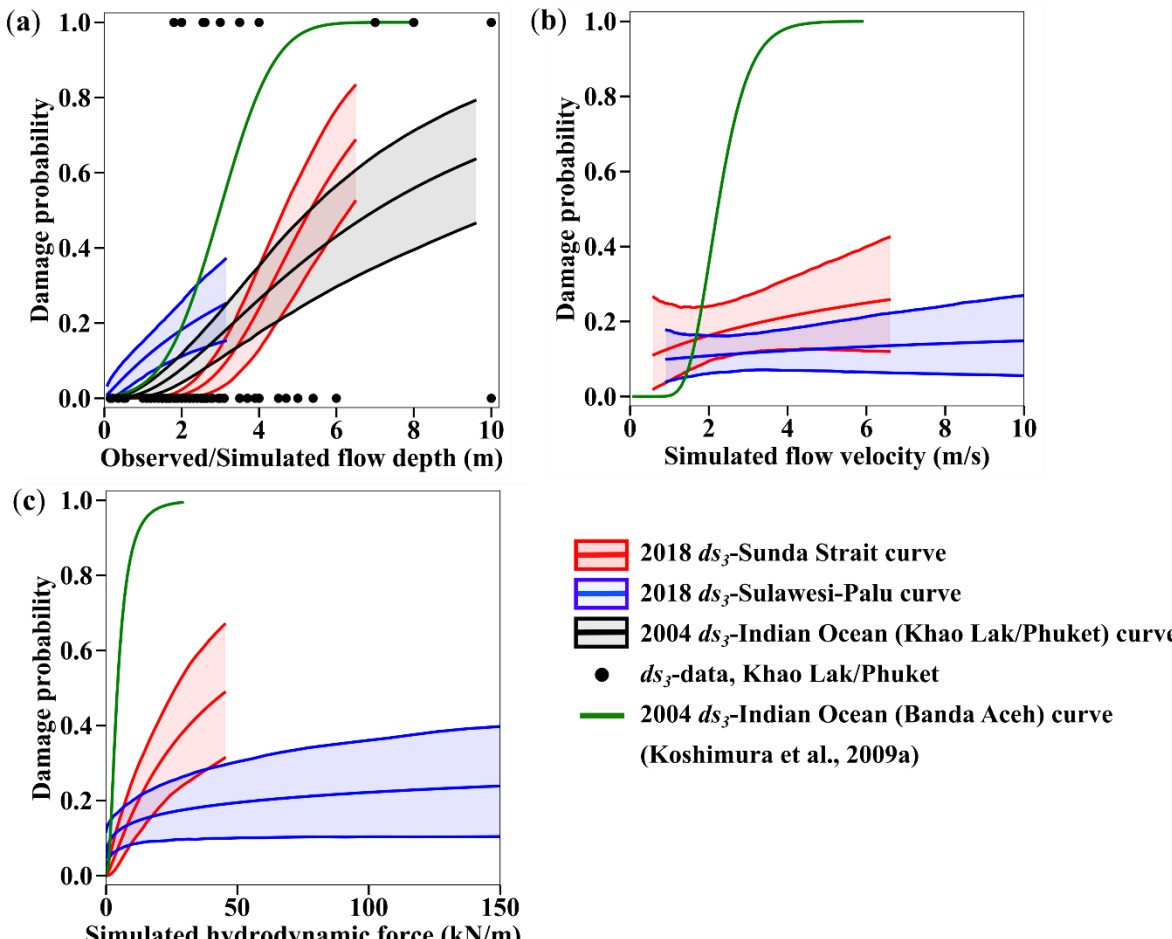

**Figure 16. Best-estimate fragility curves for the 2018 Sunda Strait tsunami, 2018 Sulawesi-Palu tsunami, 2004 IOT in Khao Lak/Phuket (Thailand) and Banda Aceh (Indonesia) as functions of (a) the observed/maximum simulated flow depth, (b) the maximum simulated flow velocity, and (c) the simulated hydrodynamic force. These fragility functions are developed only for completely damaged/washed away buildings with their 90 % confidence intervals.**


**Table 10. Damage probabilities of buildings reaching complete damage according to the intensity measures of the 2018 Sunda Strait, 2018 Sulawesi-Palu and 2004 Indian Ocean (Khao Lak/Phuket and Banda Aceh) tsunamis.**

| Tsunami intensity measure | | Building damage probability (%) | | | |
|---|---|---|---|---|---|
| | | Sunda Strait | Sulawesi-Palu | Khao Lak/Phuket | Banda Aceh |
| **Observed/Simulated flow** | 1 | < 1 | 10 | < 1 | 4 |
| **depth (m)** | 3 | 8 | 25 | 17 | 50 |

| | | | | | |
|---|---|---|---|---|---|
| | 6 | 62 | - | 43 | 99 |
| **Simulated flow velocity (m/s)** | 1 | 13 | 10 | - | < 1 |
| | 3 | 19 | 11 | - | 85 |
| | 6 | 25 | 13 | - | 99 |
| **Simulated hydrodynamic force (kN/m)** | 25 | 35 | 17 | - | 99 |
| | 50 | 48 | 19 | - | 99 |
| | 100 | - | 22 | - | - |

## 6. Discussion

### 6.1 Reliability of the building fragility curves

The reliability of the curves depends mainly on (i) the quality and the quantity of post-tsunami data and (ii) whether the tsunami intensity measures are efficient predictors of damage. With regard to the first factor, DB_Sunda2018, DB_Palu2018 and DB_Thailand2004 include relatively few data (Table 2). For each database, the relatively broad confidence intervals around the best-estimate fragility curves reflect the small sample size. Moreover, the complexity of each studied event also plays a role in how well the selected tsunami intensity measure can represent

the tsunami damage. In particular, in DB_Sunda2018 and DB_Thailand2004, only the tsunami load is responsible for the building damage. By contrast, in DB_Palu2018, buildings may have suffered prior damage due to ground shaking and liquefaction (Kijewski-Correa and Robertson, 2018; Sassa and Takagawa, 2019). Nonetheless, we are not able to establish precisely which of the surveyed buildings have suffered prior damage in the database and to what extent. The complexity of the 2018 Sulawesi-Palu event could introduce a bias in the tsunami fragility

assessment, and this has also been mentioned for other events such as the 2011 Great East Japan tsunami (Charvet et al., 2014). This bias could explain why we observed a negative slope for our $ds_3$-curves based on the observed flow depth combined with very few collapsed buildings, especially for very low intensity levels (Fig. 11). Despite the aforementioned reservations, the adopted statistical tests identified that the flow depth is consistently the best descriptor of the tsunami damage for both the DB_Sunda2018 and DB_Palu2018 data, while the flow velocity is

the worst. This finding is in line with similar observations made by Macabuag et al. (2016). De Risi et al. (2017) illustrated well the influence of the DEM resolution and the model sources on the efficiency of the flow velocity as a tsunami intensity measure. In Sunda Strait, the DEM resolution is relatively high (20 m) and it could explain why the flow velocity is not a good descriptor of damage. In Palu-City, we perform two-layer numerical modelling using the finest grid size of 1 m. However, the 2018 Palu tsunami is a complex event. The subaerial/submarine

landslides may not be the main cause of the tsunami, as shown by Ulrich et al. (2019), and it could have affected the flow velocity data. As the flow velocity of the Sunda Strait and Sulawesi-Palu tsunamis does not provide a good description of the damage, we cannot evaluate the impact of floating debris on Indonesian structures (Song et al., 2017). The hydrodynamic force of these events, computed from the flow velocity and the flow depth, does not provide a good description of the tsunami damage too.


### 6.2 Impact of the wave period, ground shaking and liquefaction events on the building damage probability

The curves comparison illustrates well the relationship between the 2004 Indian Ocean, the 2018 Sunda Strait and the 2018 Sulawesi-Palu tsunamis characteristics, summarized in Table 11, and the structural performance of buildings.


**Table 11. Characteristics of the 2004 Indian Ocean in Banda Aceh (Indonesia) and Khao Lak/ Phuket (Thailand), 2018 Sulawesi-Palu and the 2018 Sunda Strait tsunamis.**

| Tsunami event | Indian Ocean | Indian Ocean | Sulawesi-Palu | Sunda Strait |
|---|---|---|---|---|
| Database | (Koshimura et al., 2009a) | DB_Thailand2004 | DB_Palu2008 | DB_Sunda2018 |
| Location | Banda Aceh, Indonesia | Khao Lak/Phuket, Thailand | Palu-City Indonesia | Sunda Strait Indonesia |
| Tsunami source | earthquake | earthquake | landslides | landslide |
| Ground shaking* | + | - | + | - |
| Liquefaction* | + | - | + | - |
| Wave period | long (~40-45 min) | long (~40 min) | short (~3.5 min) | short (~7 min) |
| Construction type | mixed (e.g., reinforced concrete, timber…) | reinforced concrete | confined masonry | confined masonry, timber |

*+: recorded, -: not recorded.

*Impact of the wave period.*

The 2018 Sunda Strait tsunami and the 2004 IOT (Khao Lak/Phuket, Thailand) are characterized by dominant wave periods of about 7 min (Muhari et al., 2019) and 40 min (Karlsson et al., 2009; Puspito and Gunawan, 2005; Tsuji et al., 2006), respectively (Table 11). Damages from ground shaking or liquefaction episodes were not reported, so the tsunami is the main cause of building damage. We compare the Sunda Strait and the Indian Ocean

(Khao Lak/Phuket) curves based on the flow depth to investigate the impact of the tsunami wave period on buildings. In Figure 16a, the curves showed that the short wave period tsunami in the Sunda Strait is less damaging than the 2004 IOT below 5-m flow depth. For instance, for 3-m flow depth, the likelihood of complete damage is around 20 % in Khao Lak/Phuket against only 10 % in the Sunda Strait area (Table 10). On the other hand, above 5-m flow depth, the structures in Khao Lak/Phuket reveal a better performance than the ones in the Sunda Strait

area. As few data points are available beyond this value for completely damaged buildings, the Sunda Strait and the Indian Ocean (Khao Lak/Phuket) curves reliability is insufficient. Even though the long wave periods of the IOT seem to increase the likelihood of building damage, the sample size of collapsed buildings beyond 5-m flow depth is too short to validate this assumption.

*Impact of ground shaking and liquefaction events.*

The city of Banda Aceh and Khao Lak/Phuket area have been damaged by the 2004 IOT. Along Banda Aceh shores, the simulated tsunami wave period is ranging from 40 to 45 min (Prasetya et al., 2011; Puspito and Gunawan, 2005) and the one simulated in Khao Lak/Phuket is estimated to approximatively 40 min (Karlsson et al., 2009; Puspito and Gunawan, 2005; Tsuji et al., 2006). Although the tsunami wave periods are similar at both

locations, the 2004 Indian Ocean earthquake was strongly felt in the city of Banda Aceh, where it lasted about 10 min (Table 11). The earthquake intensity is estimated to VII to VIII on the Modified Mercalli Scale (Ghobarah et al., 2006; Saatcioglu et al., 2006). Despite that the ground acceleration was not recorded in the damage zones, seismic failure was distinguished from tsunami damage. For example, buildings with 3 to 5 stories were heavily damaged by the ground motion, which was amplified by the soft soil characteristics, compared to low rise

structures. In Figure 16a, the curves estimate about 50 % and 20 % of building damage probabilities for complete damage in Banda Aceh and Khao Lak/Phuket respectively, for 3-m flow depth (Table 10). Therefore, the building resilience is higher in Khao Lak/Phuket than in Banda Aceh. It comes from the fact that the Khao Lak/Phuket curve is developed for reinforced concrete buildings while the ones in Banda Aceh are produced for mixed buildings (Koshimura et al., 2009a). Another reason is that the 2004 Indian Ocean earthquake was not recorded in

Khao Lak/Phuket, so the ground motion did not damaged the buildings before the tsunami arrival. Furthermore, the likelihood of complete damage is very high for low inundation depth levels in Banda Aceh. This feature is usually observed for building suffering prior damage such as ground shaking and/or liquefactions episodes, as mentioned by Charvet et al. (2014) for the 2011 Great East Japan event.

The 2018 Sulawesi-Palu event is characterized by short wave periods of about 3.5 min according to Syamsidik et al. (2019a), like the 2018 Sunda Strait tsunami (Table 11). However, the curves based on the flow depth are remarkably different (Fig. 16a). For instance, for 3-m flow depth, the likelihood of complete damage is 25 % in Palu against 10 % in Sunda Strait, which means that buildings affected by the Sulawesi-Palu tsunami were more susceptible to complete damage. Most importantly, up to 2-m flow depth, the building damage probability is higher

in Palu than in Banda Aceh, affected by ground shaking and then, hit by a long wave period tsunami. As an example, for 1-m flow depth, the building damage probability of complete damage is about 10 % in Palu against less than 5 % in Banda Aceh (Table 10). The main cause of structural damage caused by the Sulawesi-Palu tsunami is still investigated. Mas et al. (2020) suggested that the tsunami hydrodynamic or debris impact might be the main cause of structural destruction in the waterfront area of Palu-Bay. Here, the flow velocity and the hydrodynamic

force are not good descriptors of damage, so we cannot support this assumption (Song et al., 2017). On the other hand, Palu-City sits on alluvial soil layers from Palu River and is thereby vulnerable to liquefaction disaster (Darma and Sulistyantara, 2020; Goda et al., 2019; Kijewski-Correa and Robertson, 2018). Even though the largest liquefaction areas were recorded outside the inundation zone (Watkinson and Hall, 2019), Sassa and Takagawa (2019) and Kijewski-Correa and Robertson (2018) observed land retreats along the coastal area of Palu-City (Fig.

17a,b). Most of the masonry-type buildings completely damaged are very close to these coastal retreats. Some of them were washed away by the tsunami. Therefore, these buildings do not have flow depth values and could not be used for the tsunami fragility assessment (Fig. 17b). Furthermore, in Palu, the earthquake intensity is estimated to VII to VIII on the Modified Mercalli Scale but ground shaking was not the main cause of structural destruction (Kijewski-Correa and Robertson, 2018; Supendi et al., 2019). The likelihood of complete damage is also relatively

high for low flow depth levels, so ground motion could have triggered liquefaction events and enhanced the building susceptibility to tsunami damage in the waterfront of Palu-City. This assumption cannot be verified through satellite images, it needs direct and close observations, which might be erased by the tsunami.

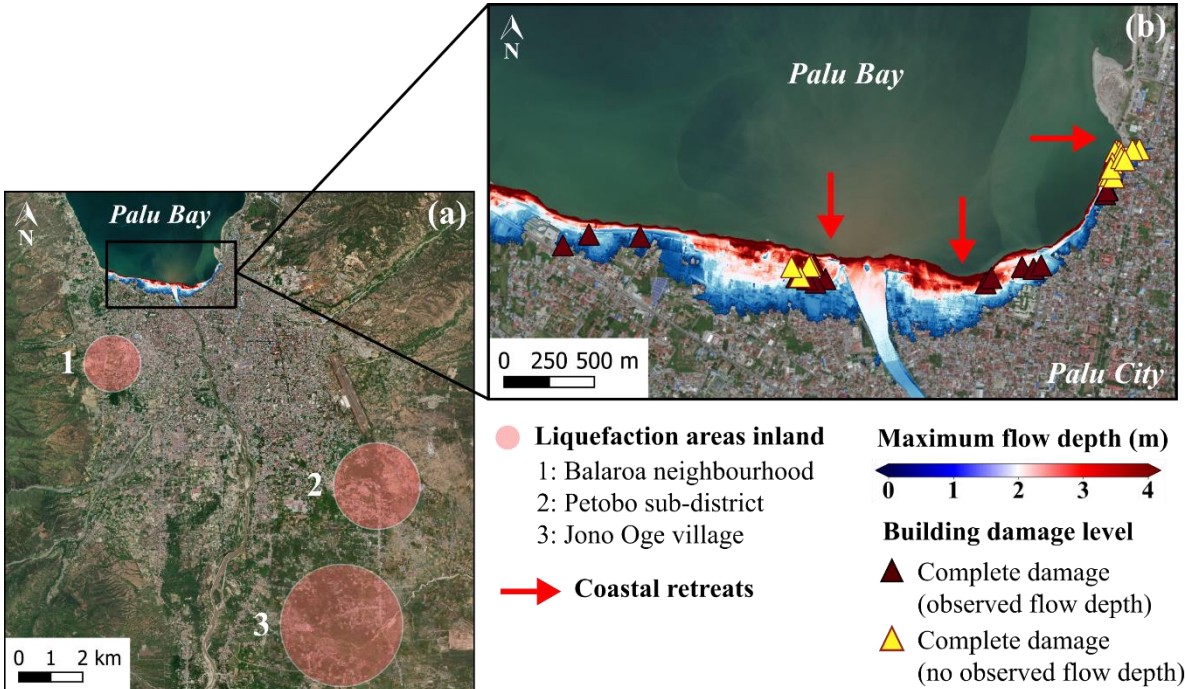

**Figure 17. (a) Liquefaction areas surveyed inland near Palu-City and (b) magnified view of the maximum simulated flow depth of the 2018 Sulawesi-Palu tsunami overlaying on the masonry-type buildings completely damaged ($ds_3$) and location of the coastal retreats surveyed in the waterfront of Palu-City (background ESRI).**

## 7. Conclusions

According to the GEM guidelines, building fragility curves of the 2018 Sunda Strait, 2018 Sulawesi-Palu and 2004 Indian Ocean (Khao Lak/Phuket, Thailand) tsunamis are empirically developed from post-tsunami databases respectively called DB_Sunda2018, DB_Palu2018 and DB_Thailand2004. To improve our understanding of the structural damage caused by the Sunda Strait and Sulawesi-Palu tsunamis, we reproduce their tsunami intensity measures (i.e., flow depth, flow velocity and hydrodynamic force) with TUNAMI two-layer model for the first time. The flow depth is constantly the best descriptor of tsunami damage for each event. The building fragility curves for complete damage reveal that: (i) the buildings affected by the Sunda Strait tsunami sustained less damage than the ones in Khao Lak/Phuket (IOT). For example, for 3-m flow depth, the building damage probability is around 20 % in Khao Lak/Phuket against 10 % in the Sunda Strait area, hit by a short wave period tsunami (landslide source). Considering the tsunami was the main cause of structural damage (i.e., damages related to ground shaking and/or liquefaction damages were not recorded), the longer wave period of the 2004 IOT may have increase the likelihood of complete damage, (ii) the building resilience is weaker in Banda Aceh than in Khao Lak/Phuket. For 3-m flow depth, the likelihood of complete damage is about 50 % in Banda Aceh and 20 % in Khao Lak/Phuket. Although both locations have been hit by the 2004 IOT, Banda Aceh was strongly affected by ground shaking before the tsunami arrival, and (iii) the buildings affected by the Sulawesi-Palu tsunami were more susceptible to be completely damaged than the ones affected by the IOT, in Banda Aceh (i.e., $\leq$ 2 m). As an example, for 1-m flow depth, the building damage probability of complete damage is about 10 % in Palu and 5 % in Banda Aceh. The Sulawesi-Palu tsunami is a complex event as it may not be the only cause of structural destruction. The 2018 Sulawesi earthquake caused minor damage to buildings and most importantly could have

triggered liquefaction events in the waterfront of Palu-City, where coastal retreats have been observed, increasing the building susceptibility to tsunami damage.

- **Appendix A. Two-layer modelling of a subaerial/submarine landslide.**

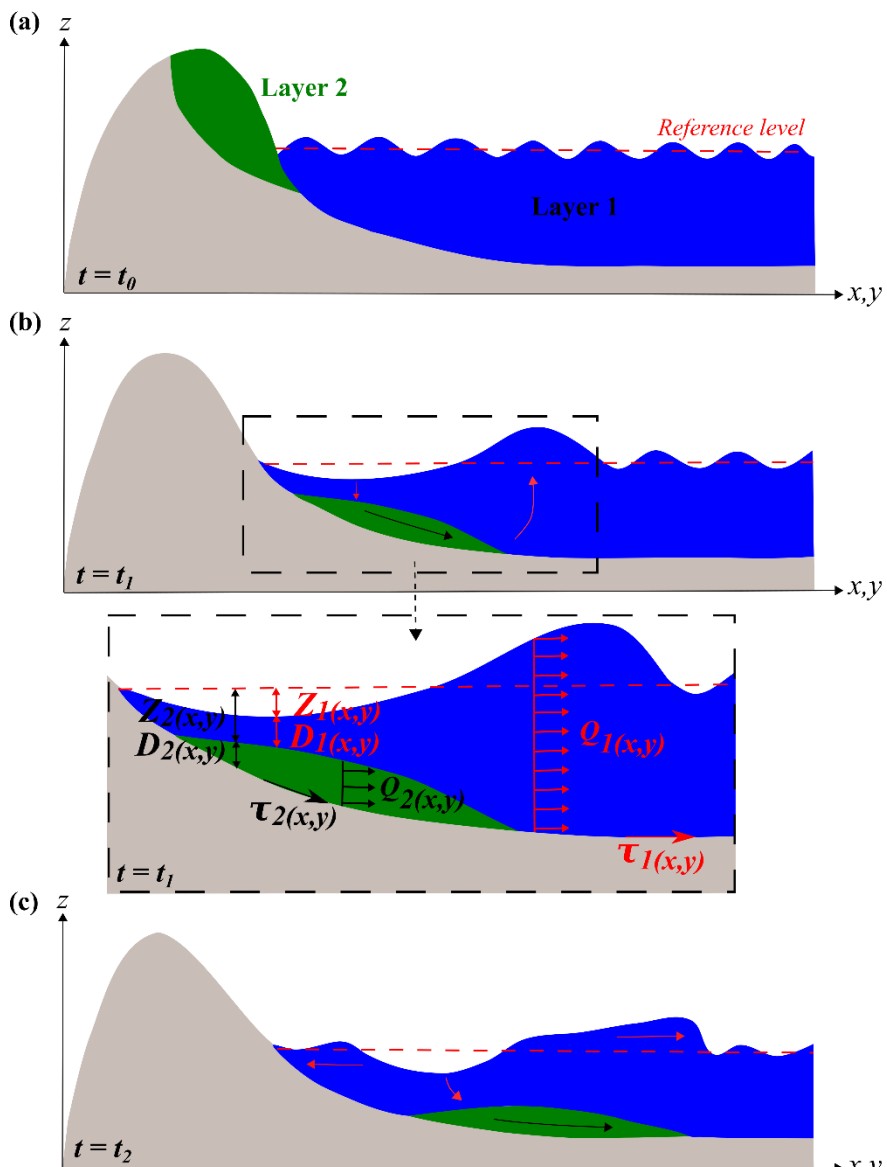

**Figure A1. Two-layer modelling of a subaerial/submarine landslide (from the original sketch of Pakoksung et al., 2019), (a) pre-failure, (b) generation of negative and positive waves due to the landslide and (c) landslide in progress and wave propagation.**

- **Appendix B. The 2018 Sunda Strait tsunami generation, propagation and inundation modelling.**

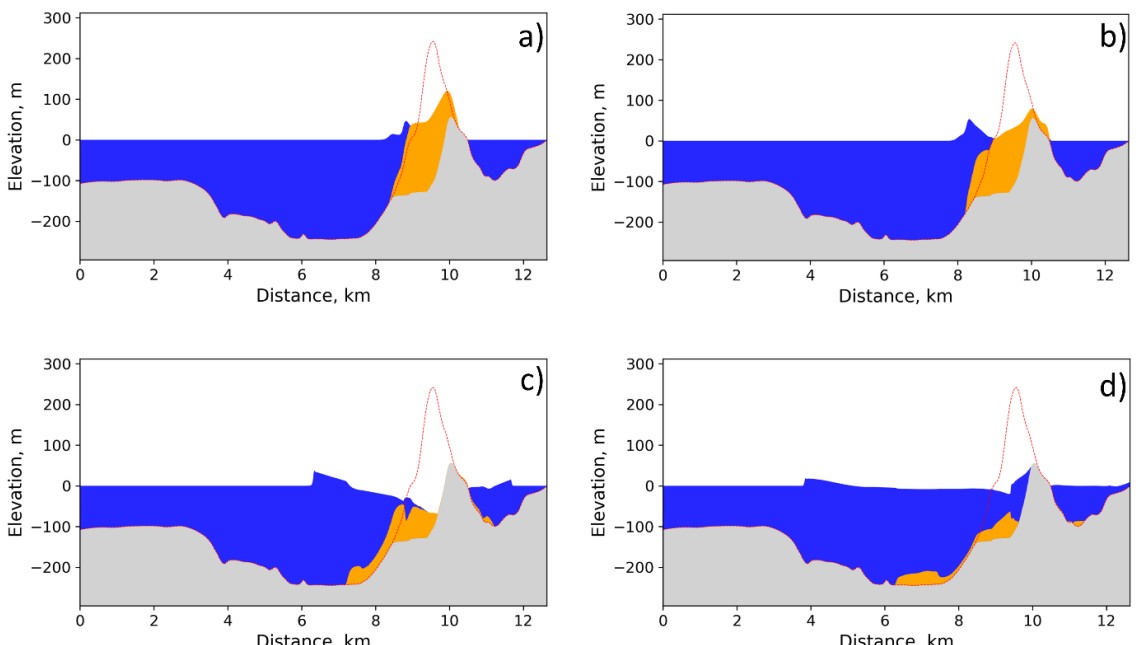

**Figure B1. Temporal evolution of the 2018 Sunda Strait tsunami wave (a) 10 s, (b) 20 s, (c) 60 s, and (d) 120 s after the volcano flank collapse. The red line is the topography/bathymetry before the landslide (Pakoksung et al., 2020).**

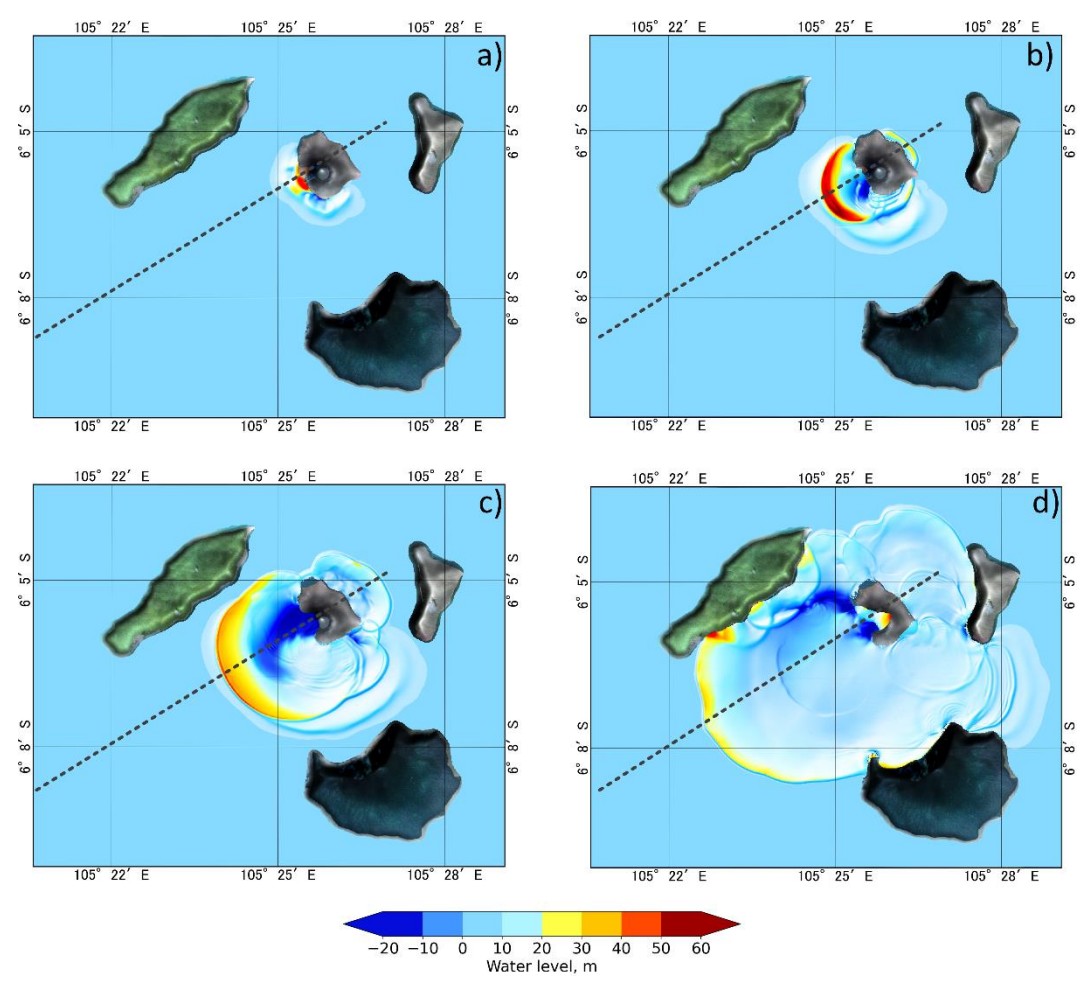

**Figure B2. Temporal evolution of the 2018 Sunda Strait tsunami wave (a) 10 s, (b) 20 s, (c) 60 s, and (d) 120 s after the volcano flank collapse (Pakoksung et al., 2020).**

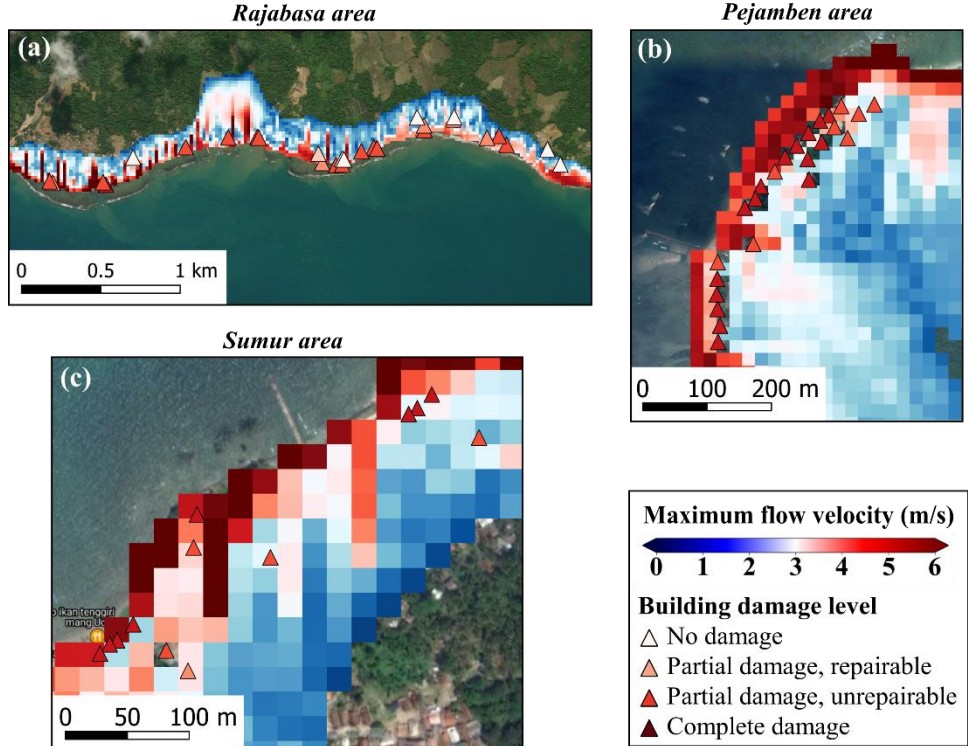

**Figure B3. (a,b,c)** Magnified views of the maximum simulated flow velocity of the 2018 Sunda Strait tsunami overlaying on the damaged building data in Rajabasa, Pejamben and Sumur areas (background ESRI and © Google Maps).

- **Appendix C. The 2018 Sulawesi-Palu tsunami generation, propagation and inundation modelling.**

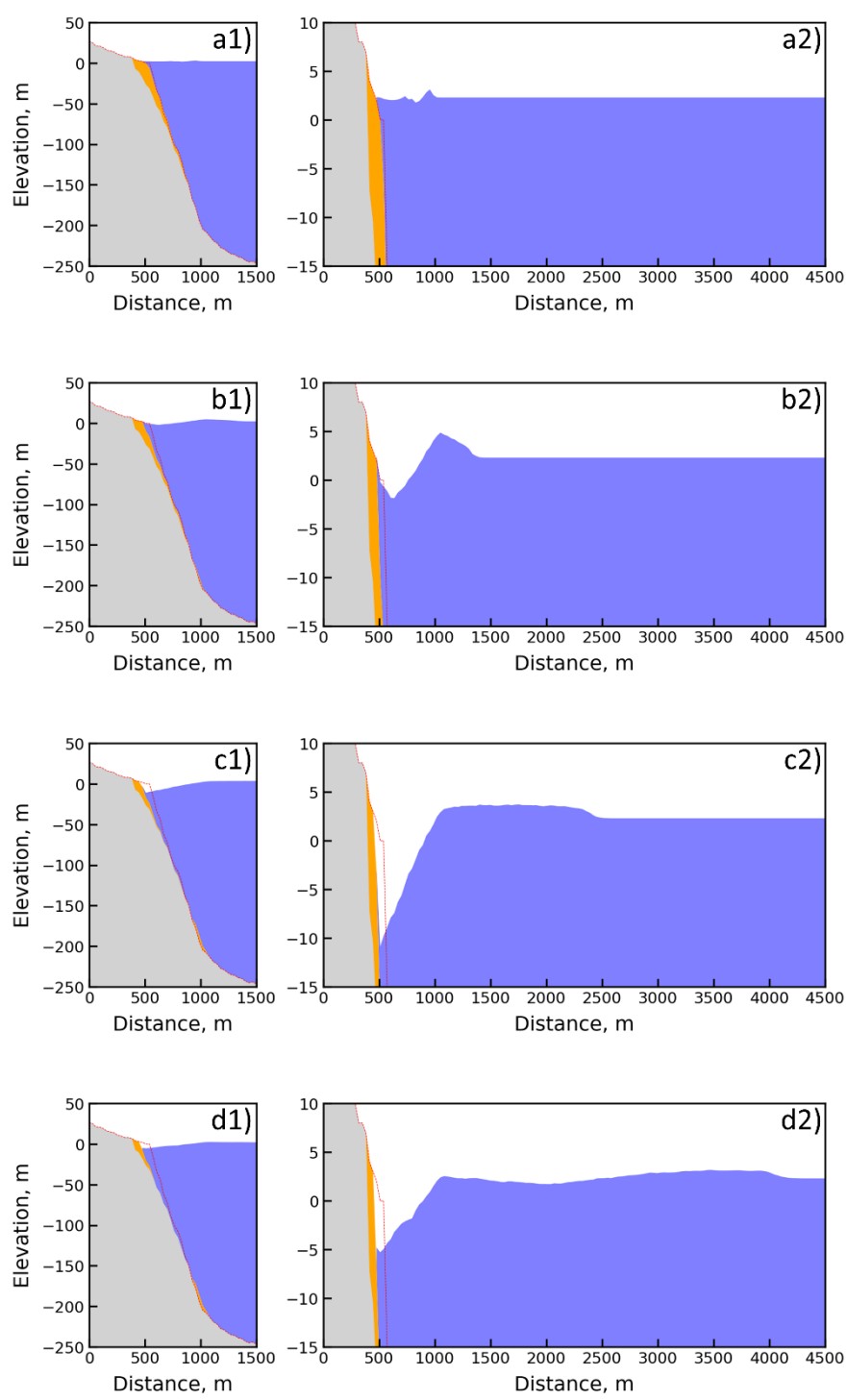

**Figure C1. Temporal evolution of the 2018 Sulawesi-Palu tsunami wave (a) 2 s, (b) 10 s, (c) 30 s, and (d) 60 s after S8 landslide. The red line is the topography/bathymetry before S8 landslide (Pakoksung et al., 2019).**

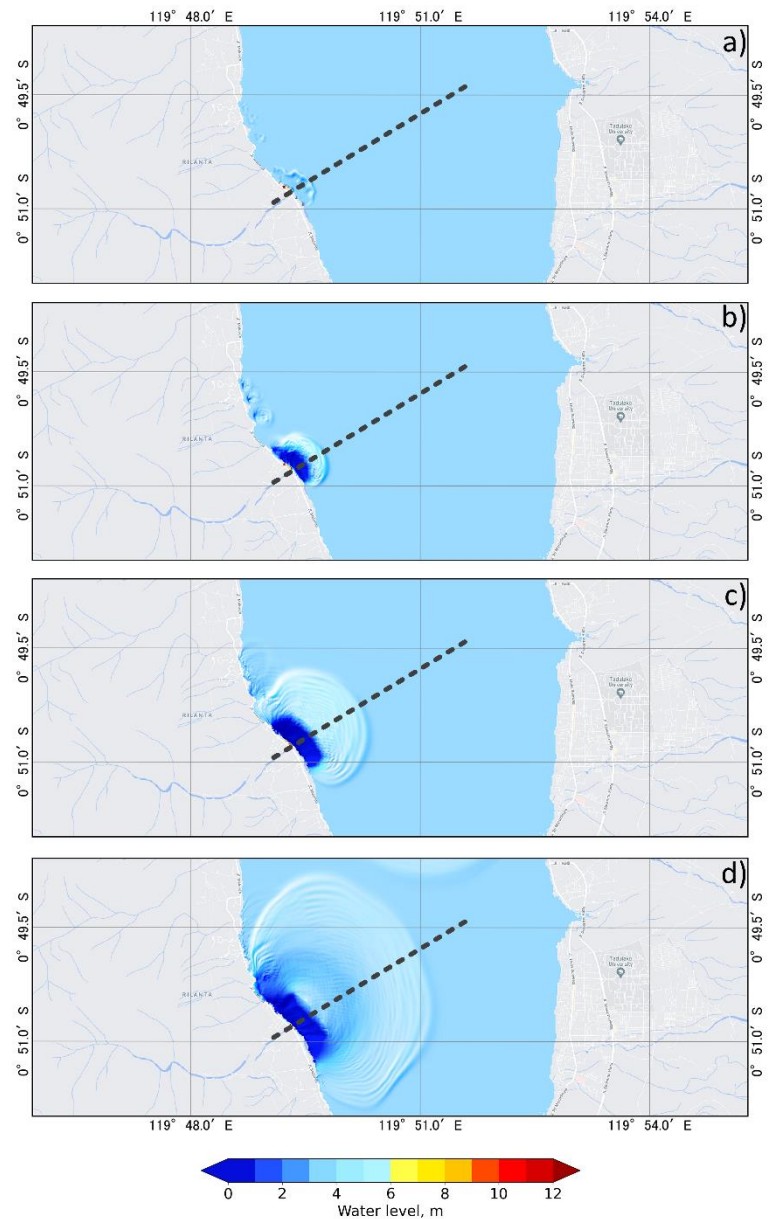

**Figure C2. Temporal evolution of the 2018 Sulawesi-Palu tsunami wave (a) 2 s, (b) 10 s, (c) 30 s, and (d) 60 s after S8 landslide (Pakoksung et al., 2019).**

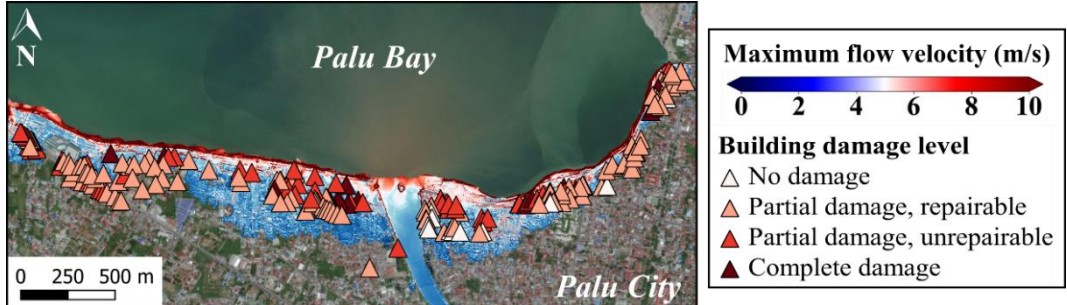

**Figure C3. Sulawesi-Palu final tsunami inundation model with the maximum simulated flow velocity overlaying on the damaged building data (background ESRI).**

- **Appendix D. Statistical model selection: comparison of AIC values for logit and cloglog link functions (DB_Sunda2018)**

**Table D1. AIC values for the three models assuming logit and cloglog link function fitted to the observed and simulated tsunami intensity measures of DB_Sunda2018.**

| Model | AIC | | | |
|-------|-----|-----|-----|-----|
| | **Observed flow depth** | **Simulated flow depth** | **Simulated flow velocity** | **Simulated hydrodynamic force** |
| **logit** | | | | |
| **M3** | 132.5 | 139.9 | 224.3 | 196.0 |
| **M4** | 146.4 | 153.8 | 229.0 | 220.3 |
| **M5** | 134.2 | 141.3 | 225.5 | 197.4 |
| **M1** | 163.6 | 169.5 | 247.0 | 217.9 |
| **cloglog** | | | | |
| **M3** | 134.8 | 139.9 | 224.3 | 200.9 |
| **M4** | 144.5 | 151.9 | 230.4 | 218.4 |
| **M5** | 136.1 | 140.9 | 225.9 | 202.6 |
| **M1** | 168.8 | 172.2 | 247.8 | 224.2 |

- **Appendix E. Regression coefficients for the building fragility curves of the 2018 Sunda Strait, Sulawesi-Palu and 2004 Indian Ocean (Khao Lak/Phuket) tsunamis.**

**Table E1. Regression coefficients for the 2018 Sunda Strait tsunami fragility curves based on DB_Sunda2018**

| Tsunami intensity measure | Regression coefficients (best-estimate, standard error) | | | | |
|---------------------------|------|------|------|------|------|
| | $\theta_{01}$ | $\theta_{02}$ | $\theta_{03}$ | $\theta_1$ | $\theta_{2(class=Timber)}$ |
| **Observed flow depth** | -0.29, 0.415 | -1.99, 0.402 | -4.52, 0.639 | 2.76, 0.408 | 2.08, 0.416 |
| **Simulated flow depth** | -0.26, 0.377 | -1.69, 0.355 | -4.03, 0.545 | 2.40, 0.346 | 1.96, 0.390 |
| **Simulated flow velocity** | 0.80, 0.300 | 0.14, 0.293 | -1.17, 0.307 | 0.27, 0.276 | 1.40, 0.296 |
| **Simulated hydrodynamic force** | -4.07, 1.016 | -4.95, 1.058 | -6.50, 1.116 | 0.61, 0.118 | 1.45, 0.311 |

**Table E2. Regression coefficients for the 2018 Sulawesi-Palu tsunami fragility curves based on DB_Palu2018**

| Tsunami intensity measure | Regression coefficients (best-estimate, standard error) | | | |
|---------------------------|------|------|------|------|
| | $\theta_{01}$ | $\theta_{02}$ | $\theta_{03}$ | $\theta_1$ |
| **Observed flow depth** | 2.33, 0.315 | -0.71, 0.193 | -2.09, 0.286 | 0.57, 0.272 |
| **Simulated flow depth** | 2.37, 0.319 | -0.79, 0.199 | -2.20, 0.293 | 0.91, 0.286 |
| **Simulated flow velocity** | 2.07, 0.428 | -0.87, 0.370 | -2.23, 0.428 | 0.18, 0.335 |
| **Simulated hydrodynamic force** | 0.35, 1.034 | -2.65, 1.061 | -4.03, 1.096 | 0.24, 0.127 |

**Table E3. Regression coefficients for the 2004 IOT in Khao Lak/Phuket (Thailand) based on DB_Thailand2004**

| Tsunami intensity measure | Regression coefficients (best-estimate, standard error) | | | |
|---------------------------|------|------|------|------|
| | $\theta_{01}$ | $\theta_{02}$ | $\theta_{03}$ | $\theta_1$ |
| **Observed flow depth** | 0.71, 0.377 | -1.59, 0.361 | -3.84 0.481 | 2.00, 0.342 |

*Code and data availability.* Post-tsunami field surveys data are available from references cited in the text. The bathymetric and topographic data for the Sunda Strait area were provided by BATNAS and DEMNAS, Indonesia, respectively (http://tides.big.go.id/DEMNAS/index.html). The Agency for Geo-spatial Information (BIG), Indonesia provided the bathymetric and topographic data for Palu-Bay. The tidal gauge records were supplied by the Coastal Disaster Mitigation Division, Ministry of Marine Affairs and Fisheries, Jakarta, Indonesia. Spatial data in this study are depicted through QGIS software.

*Author Contributions.* FI, AS, KP, EL, II and FB designed and coordinated this research. AS, KP and EL performed the tsunami simulations and participated to the calibration of the inundation models in Palu-Bay and in the Sunda Strait. SS and RP contributed to the tsunami data collection in Sunda Strait and Palu areas, respectively. II developed the fragility functions through advanced statistical analysis. All authors contributed to the drafting of the manuscript.

*Competing interest.* The authors declare that they have no conflict of interest.

*Acknowledgements.* We greatly acknowledge the two reviewers for their constructive comments and recommendations that helped to improve the quality of this manuscript. This research was funded and supported by the Japan Society for the Promotion of Science (JSPS) Grant-in-Aid for Young Scientists, the JSPS-NRCT Bilateral Research grant, the World Class Professor (WCP) Program 2018-2020 Promoted by Ministry of Education And Culture of Republic of Indonesia, the Pacific Consultants Co., Ltd., the Willis Research Network (WRN), the Tokio Marine & Nichido Fire Insurance Co., Ltd., the National Institute of Water and Atmospheric Research (Project: CARH2106), UKRI GCRF Urban Disaster Risk Hub, and GLADYS.

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
