# Peer review of "Characteristics of building fragility curves for seismic and non-seismic tsunamis: case studies of the 2018 Sunda Strait, 2018 Sulawesi-Palu and 2004 Indian Ocean tsunamis."

_Natural Hazards and Earth System Sciences, 2020_

## Referee Comment (RC1) · Anonymous Referee #1 · 29 Dec 2020

The paper systematically evaluates tsunami hazard and damage data from three Indonesian tsunami events to develop tsunami fragility functions and compares the developed tsunami fragility functions for different events as well as for different building typologies. The significance of this work lies in this systematic comparison based on rigorous statistical techniques and, as the authors claimed, this has not been done in the literature. This is a useful research contribution and thus, eventually, this work should be published. Having mentioned that, I found that several clarifications are necessary to appreciate the significance of this work. I suggest the following points to be

considered during the revision.

Major comments: In Section 3.1.1, the two-layer model is explained. Can the authors explain more on how physically the layer 1 and layer 2 interact? From the equations, the interacting aspects of the two layers are expressed in terms of the water level gradients $Z_1$ and $Z_2$. So the flux $Q_1$ in layer 1 is fluid, while the flux $Q_2$ in layer 2 is s.oil mass? A figure would be useful addition for this section. Equation (7): please specify the units. Figure 2 and Section 3.1.2: It is not clear how the computational cells that correspond to buildings (Figure 2d) can be inundated or not in tsunami simulation. In Section 3.2: could you comment on the vertical accuracies of the DEM/DSM used for the investigations? How were they derived? I would guess local LiDAR data? Throughout the investigations, were the tidal effects taken into account? For the 2018 Palu earthquake, the tidal levels have important contributions (e.g. Goda et al., 2019). In Section 3.2.3, how credible the landslide source model for the Palu event? For example, a detailed seismic source model can explain the majority portion of the observed tsunami in Palu Bay (e.g. Ulrich et al., 2019). How were the effects due to the coseismic deformation and tidal level considered (e.g. Goda et al., 2019)? In light of the missing elements in the tsunami source model, the landslide source model may be considered to be biased. I think this discussion is important for the NHESS journal audience. This is a comment: the scatter plot shown in Figure 9 is not well correlated (i.e. simulation vs observation), which may be due to mis-specified tsunami source. Also note that 'Figure 9' is misspelled. Page 13: Can other link functions other than probit be used? Figure 10 (and other figures as well): Can the data also be displayed? Can the authors clarify the confidence interval indicates the confidence interval of the regression line or the prediction interval of the prediction model? I think by including the data points in Figure 10, this becomes obvious. I think this clarification is important because the number of data is small. Section 5, Line 430: I do not understand the intention of showing the tsunami fragility models based on simulated intensity values? When the tsunami simulations are calibrated reasonably well with the observations, using the same damage data, the fitted fragility models are expected to be similar (as demonstrated in Figure

13). But I do not see the benefit of using the simulated tsunami intensity values unless the authors use the damage data where the observations are not available and thus the tsunami intensity values need to be estimated. But this work does not investigate this aspect. Altogether the simulated cases can be removed. Figure 13: as discussed by the authors, the fragility functions based on flow velocity and (probably) hydrodynamic force do not show realistic features and thus not really useful. It may be useful to show such results for one case but for other cases, they are not really useful, especially for flow velocity. My concern is that careless readers may attempt to use such models as black box models. Figure 14: why the data are only shown for x values greater than 1? Should they start with the theoretical constraints that zero fragility for zero hazard values? My concern is again that careless users may take such unrealistic models as they are. Figure 15: I understand that the results are based on statistical fitting but these curves do not look realistic. Are they reliable? I think the reliability of the curves should be a part of the discussion (beyond the statistical confidence level etc). Can one use these functions reliably? Figure 16: From my perspectives, the comparison of the curves based on flow velocity and hydrodynamic force is not robust. I would suggest focusing on the flow depth based models which show some realistic fragility features.

Goda K, Mori N, Yasuda T, Prasetyo A, Muhammad A and Tsujio D (2019) Cascading Geological Hazards and Risks of the 2018 Sulawesi Indonesia Earthquake and Sensitivity Analysis of Tsunami Inundation Simulations. Front. Earth Sci. 7:261. doi: 10.3389/feart.2019.00261 Ulrich, T., Vater, S., Madden, E. H., Behrens, J., van Dinther, Y., van Zelst, I., et al. (2019). Coupled, physics-based modeling reveals earthquake displacements are critical to the 2018 Palu, Sulawesi Tsunami. Pure Appl. Geophy. 1–41. doi: 10.1007/s00024-019-02290-5

Minor comments Page 1, Line 18: cumulative distribution functions -> delete cumulative distribution. Strictly speaking, the fragility function is not the cumulative distribution function and this expression is confusing. I would suggest deleting 'cumulative distribution'. There are a few places that have the same expression. Page 1, Line 28: 'liquefaction events . . .' The majority of the damage and loss during the Palu earthquake was due to slope failures (which involve liquefaction as physical failure mechanism). It is not clear (especially in the abstract), this 'liquefaction' refers to the slope failure cases (e.g. Petobo) or the flat coastal area along Palu Bay. Given the nature of this event, it would be better to rewrite this sentence to be more specific which area/incidences the authors are referring to. Page 1, Line 38: vertical -> vertical and horizontal. Page 1, Line 41: period -> periods. Page 2, Line 44: were -> was. Page 2, Line 49: few -> a few. Page 2, Line 50: delete finally. Page 2, Line 60: reported to -> reported at. Page 2, Line 60: what is 'largely exceeded'? The meaning is not clear. Page 2, Line 62: assumption -> hypothesis (I think hypothesis is more appropriate). Page 2, Line 68: The sentence 'Koshimura et al. . . .' reads strangely in a sense that the tsunami fragility concept existed before this work. I agree that the work by Koshimura et al. was very influential. Page 2, Line 69: delete 'cumulative distribution'. Page 3, Line 86: treated -> analyzed. Page 3, Line 93: exposed -> investigated. Page 5, first line: are -> is. Page 14, Line 284: appear -> appears. Page 19, Line 384: depicted -> listed or summarized. Page 20, Line 398: identical curves -> identical slopes?

---

## Referee Comment (RC2) · Anonymous Referee #2 · 15 Jan 2021

Lahcene et al. 2020 use field data and numerical modelling to present building fragility curves for three past tsunami events in Indonesia and Thailand. They use data from the 2004 Indian Ocean tsunami, the 2018 Palu tsunami and the 2018 Sunda Strait tsunami. Their goal is to demonstrate that the differences in the building fragility curves are due to hydrodynamic parameters of the waves, ground shaking and liquefaction. It is of great interest to understand how ground shaking or liquefaction weakens buildings posterior to a tsunami impact. However, the authors do not sufficiently demonstrate how building fragility curves may contribute to this understanding in the manuscript.

The authors use numerical modelling for some of the cases, but the application of models, their sources, and modelling framework are not specified sufficiently. The application of the advanced statistical methods used is not very clear. The authors should add a statistical methods section to explain why, what, and how they use the different statistical tools and cite corresponding references. The authors should rewrite the conclusions because they are not based on the study the authors performed. Finally, parts of the manuscript could be better structured and, in some sections, the excessive use of passive voice phrases reduces the readability. I suggest a careful review with a native speaker or using a grammar and spelling checker. Nevertheless, I believe the manuscript has some potential and could be eventually considered for publication if the authors consider the following comments.

Major comments:

1: The authors present the fragility curves for the three events in the four locations (Fig. 16 a). The curves demonstrate that for low flow depth values (less than 2 m flow dept) building fragility was largest in Palu 2018 followed by 2004 Banda Aceh, followed by 2004 Khao Lak, followed by Sunda Strait 2018. Above 2 m flow depth, the curves for 2004 Banda Aceh demonstrate the largest building fragility. The authors conclude that ground shaking and liquefaction contributed to the fragility curves for Palu 2018 and Banda Aceh 2004. Although it is possible that both ground shaking and liquefaction may contribute to the fragility curves, the authors do not demonstrate this. Hence their claim is pure speculation. To demonstrate that ground shaking that allows inferring the damage of buildings that have not been hit by the tsunami. Maybe they could use seismic intensities or peak ground acceleration to infer the damage of buildings that have not been hit by the tsunami inundation area to foster their hypothesis.

2: The authors write in the conclusions, page 30, line 573 f. : '..., it is clearly demonstrated that liquefaction events can increase building susceptibility.' and page 30, line NHESSD
574 f. : '..., the building were previously affected by severe liquefaction episodes.' This sentence is not a conclusion from their work. Most importantly is to mention that the largest liquefaction areas were located outside the tsunami inundation areas. The authors should consult Watkinson and Hall (2019) and Syifa et al. (2019). Even though Sassa and Takagawa (2019) conclude that they found evidence for extensive liquefaction in coastal areas, the authors should quantify how many of the database's observed buildings were affected by liquefaction. The authors could overlap the liquefaction areas with figure 8 to see how many of the buildings were affected. The current state of the manuscript does not allow to conclude that buildings were weakened by liquefaction. Moreover, Mas et al. (2020) write that tsunami hydrodynamic and debris impact forces may have been the principal causes of failure and collapse in Palu Bay's waterfront area.

3: The conclusion that the building fragility curves for Banda Aceh and Khao Lak are different because of the ground shaking in Banda Aceh are incomplete. Just because the locations were hit bit the same tsunami event, does not necessarily mean that the wave period was the same in both locations. The rupture at the Sunda Megathrust was longer than 1000 km, and slip rates along the fault were heterogeneous (Rhie et al. 2007, Koshimura et al. 2009). Consequently, waves with different periods and hydrodynamic features may have impacted Khao Lak and Banda Aceh. Applying numerical models for both sites could show the differences, but the authors do not present tsunami simulations for Banda Aceh and Khao Lak.

4: Regarding the numerical modelling in the manuscript, the authors should comment on why they use modelling for the 2018 Palu and Sunda Strait events but not for the 2004 Indian Ocean tsunami. I believe the authors could draw interesting conclusions if they would compare impacting wave shapes for the 2004 Indian Ocean tsunami in Khao Lak and Banda Aceh.

5: Further, the authors should clearly state the motivation for their numerical modelling efforts. It is not clear why they use models where many observations exist.

**NHESSD**
6: The Digital Elevation Model (DEM) for Palu has a resolution of 1 m. The DEM for the Sunda Strait event has a resolution of 20 m. If the authors want to compare the cases, they should use the same cell size for DEMs or explain why they believe that simulations are comparable when using 400 times bigger cell size. Apart from that, I believe the reader would be interested in the data that allows building a DEM with 1 m resolution for Palu. The authors should also name the references for the dataset used in all DEMs.

7: The authors use hypothetical landslide sources for the 2018 Palu event. Those are not in agreement with some other published studies (Ulrich et al. 2019, Gusman et al. 2019). In figure 6, the authors present those hypothetical landslides as principal tsunami source without explaining why they have this assumption. The authors must include a review of previously published sources and comment on the reasons for modifying the sources in their manuscript. In figure 9, they compare the observed with simulated flow depth and claim that their model is in good agreement with the observations. To my understanding, figure 9 in the manuscript demonstrates that the simulation does not match the observation. The authors must explain why they believe the model is of good quality. Further in the manuscript's conclusion, the authors write on page 30, line 574 f.: 'Although Palu-Bay was hit by a non-seismic tsunami....'. These are the authors' assumptions and therefore are not valid as a conclusion and need to be rewritten. Please also note that figure 9 is misspelt.

8: The authors should include a paragraph on proposed flank collapse sources from other studies on the 2018 Sunda Strait tsunami. Please include Williams et al. (2019), Grilli et al. (2019), Omira and Ramalho (2020), Dogan et al. (2021).

9: The authors write that they automatically corrected the flow depth traces for the 2018 Sunda Strait event. It is not clear how the authors do that. Are they using GPS field measurements or LIDAR data? If the authors use a method previously presented, then they must cite the corresponding reference. They observe a mean difference of flow depth values of 0.28 m for 94 traces. How far is this value representative for the 94
traces, and it is not clear if the authors use this value for correction? I suggest rewriting this section. The resolution of the DEM is 20 m. Do the authors believe this resolution is sufficient to obtain reasonable values of flow depth and flow velocity?

10: Consistency with abbreviations and variables: Sometimes the authors use for Indian Ocean tsunami IOT (e.g. page 1, line 23, line 31; page 3, line 91; etc.) sometimes they use IO (e.g. page 2, line 81; page 4, line 101; page 20, line 416). YouThey authors use GEM as an abbreviation on page 2, line 82 but only introduce the Global Earthquake Model (GEM) on page 13. The authors also use the abbreviation AIC without stating what the abbreviation stands for (page 18 ff.). In 4.1 the authors use the Greek letter pi as probability, and in 4.2 they use P. What is the difference?

Minor comments:

Page 2, line 51: Some other studies consider less volume and other particularities of the collapse. Please include them (Williams et al. 2019, Grilli et al. 2019, Omira and Ramalho 2020, Dogan et al. 2021).

Page 2, line 81: IO or IOT? Please be consistent.

Page 2, line 82: Please indicate what the abbreviation GEM stands for.

Page 3, line 91: I suggest rewriting the sentences since seismic and non-seismic curves is not clear.

Page 4, line 101: IO or IOT? Consistency!

Page 4, line 106: Why do the authors believe the databases are statistically representative since they explain later in section 4 that they use reduced samples of the databases DB\_Palu2018' and DB\_Sunda2018'. For example, from 463 observations in Palu they use 124 observation. I recommend restructuring the manuscript combining section 2 and 4 or put the final databases used in section 2. Is the number of timber buildings enough to be statistically significant? NHESSD
Page 5, line 142 f.: Please specify the appropriate kinematic and dynamic boundary conditions for the interfacing layers.

Page 6, Eq. 7 ff.: In equation 7 is d a constant? Please define theta.

Page 7, figure 2: It is not clear what the building occupation ratio is. Please introduce a definition. What do the polygons in 2 (d) represent? Please add a legend. Are those cells 100% covered? What about the rest of the layer? 0% occupation? Please clarify!

Page 7, line 180: Please avoid having two letters for the same variable.

Page 8, figure 3: It is unclear how the corrections are applied to the Digital Surface Model.

Page 9, figure 4: What do mean by profile realized? Are those measurements?

Page 10, figure 5: Do the triangles represent single buildings? It is probably better to choose a representative area on a scale with many surveyed buildings like a city or village instead of large parts of the coast. The figure now does not illustrate well the flow depth close to the buildings. What about the flow velocity plots?

Page 10, line 213: Is landslide S8 oriented towards the city? Isn't that the slide that was captured by the pilot in the departing plane? Isn't the slide direction perpendicular to the bay?

Page 10, line 215 f.: What is meant by landslide ratio of 1.2?

Page 10, line 217 f.: Why do you only overlay 175 traces?

Page 10, line 220 – 225: This section is not very clear.

Page 11, table 3: What are the sources for the volume of the landslides?

Page 12, figure 8: What is the source of the topography and bathymetry data for the DEM with 1 m resolution.

Page 12, figure 9: Figure is misspelt. The figure demonstrates that the model does not
well represent the observations. What is an S8 ratio of 1.2?

Page 12, line 237 – page 13, line 239: I suggest putting the number and type of buildings for all locations in a table.

Page 13, table 4: Please specify why you only use 124 out of 463 flow depth values for Palu.

Page 13, line 252: Please put 'Global Earthquake Model' the first time you use the abbreviation.

Page 13, section 4.1 & section 4.2: What is the difference? First, you identify the explanatory variable for building damage. Then in section 4.2 you include the damage states and the model selection. I believe you could make this section 4 much shorter by focusing on the relevant information. I suggest preparing a short and concise paragraph on the statistical methods used and then present the results for each site. I also miss a short introductory phrase to the Akaike Information Criterion (AIC) and the likelihood ratio tests you applied.

Page 14, 284 f.: There is something wrong in this sentence 'The intercept of the curves for the two material types appear be sustainably different.'

Page 15, 16 and 17, figures 10, 11 and 12: I recommend plotting the confidence intervals with lines only without shaded areas because in the overlapping areas you get different colours than depicted in the legend. In case some reader would like to print the manuscript in black and white, only it will be more illustrative. Furthermore, I suggest introducing a symbol for the variable flow depth. I believe it is better to delete the word material in the legend since it creates some ambiguity with the symbols used. Also, confined masonry is not a material; it is a construction or building type. In figures 10 and 11, you use the same symbol for timber and reinforced concrete, I suggest selecting a unique symbol for each construction type.

Page 15, line 301: Instead of material I would suggest using construction or building
type. Confined masonry or reinforced concrete are construction types, not materials.

Page 15, figure 10 and line 293, etc.: You use a couple of times 'in order to'. There is no need for using 'in order', a simple 'to' is enough.

Page 16, line 305 f.: Instead of 'GLM models are finally fitted to DB\_Thailand2004 in order to construct fragility curves and their 90 % confidence intervals for the three individual damage states, as depicted in Fig. 12.' I suggest to writing: 'We fit GLM model to DB\_Thailand2004 to construct fragility curves for the three damage states and plot them with their 90 % confidence interval in Fig. 12.'. Generally, I suggest avoiding passive voice use because of this increase the readability of a manuscript.

Page 17, Eq. 18: The indexing of the model equations is not precise. Please check the standard of the journal.

Page 18, lines 329 – 337: I suggest depicting the functions in exemplary plots, and possibly you could simplify the verbal description. Line 331: Eq. 8.2 does not exist.

Page 18, line 348: Please explain what is meant by AIC values.

Page 19, line 380: Please define hydrodynamic force and explain how you compute it.

Page 20, line 409 – 415: This is much text for a simple conclusion. Please simplify.

Page 20, line 416: IO. Please be consistent.

Page 21, line 434 f.: Here you write: 'The curves suggest that confined masonry-type buildings have higher performance than timber structures.' On page 19, line 388: 'A timber building is found to sustain more damage than a confined masonry one for all damage states.' Please clarify.

Page 22, figure 13: It is unclear which curve was produced by Syamsidik et al. (2020). The red dashed line is Syamsidik et al. (2020)? Please put the reference also in the legend of the figure to be exact. Can you explain why the curve of Syamsidik et al. (2020) is that different to yours? I do not understand why you put the data points in

NHESSD
the figures. Is there no better way to illustrate your data? It is hard to distinguish the data points some of the red points that superimpose orange ones are hard to see. In figure 13 (a) and (b) why are the points distributed differently? Moreover, I propose putting 'confined masonry buildings' in the figure. Maybe in the legend or in a figure title. Otherwise, the reader may mix up the figures of confined masonry and timber. I suggest plotting the observed flow depth's fragility curves with the ones of the simulated flow depth. Otherwise, it is hard to see any difference. Is there a difference?

Page 23, figure 14: Please consider the comments of figure 13 also for this figure 14. I suggest putting 'timber buildings' somewhere in the figure. Maybe in the legend or a figure title. Otherwise, the reader may mix up the figures of confined masonry and timber.

Page 24, line 458: Please review and correct the following sentences: 'The fragility curves based on observation and simulation are similar enough to consider the computed curves as functions of the hydrodynamic features of the tsunami reliable (Fig. 15c,d).'

Page 25, figure 15: Please consider the comments for figures 13 and 14 for this figure. I suggest plotting the fragility curves of observed and simulated flow depth in the same graph (hence combine (a) and (b)).

Page 25, table 11: Why do you use tsunami intensity measure values different from Table 10? It makes them less comparable.

Page 26, line 472: If Koshimura et al. (2009a) building fragility curves are for mixed buildings and your curves are for mixed buildings are they comparable? What are the percentages of each construction type?

Page 26, line 488: How do you explain why Banda Aceh buildings are destroyed at 6 m/s flow velocity whereas in Palu and Sunda Strait buildings sustain this flow velocity?

Page 27, figure 16: I suggest adding the reference to the legend of the figure.

**NHESSD**
Page 28, table 13: It is not proven that a non-seismic source triggered the Palu tsunami. You need to explain what the symbols + and – mean in the lines liquefaction and ground shaking. I suggest changing the line 'Construction material' to 'Construction type'.

Page 28, line 514 f.: Why do you think the Sunda Strait event buildings reveal a better performance than in Khao Lak? How many buildings were affected in Sunda Strait with flow depth values larger than 5m? How much area was inundated with flow depth values larger than 5 m?

Page 28, line 527 f.: Please be aware that the largest areas of liquefaction are located outside of the tsunami inundation area. Please see Watkinson & Hall (2019) and Syifa et al. (2019). Although Sassa and Takagawa (2019) identified some small liquefaction areas near the coast, you need to quantify how liquefaction processes effectively damaged many buildings in your database.

Page 28, line 529: It is not proven that only landslides generated the Palu-tsunami. I suggest changing the phrasing' non-seismic source'.

Page 29, lines 532 – 536: The problem here is that your estimates of the flow velocity are based on numerical modelling for the Sunda Strait with 20 m resolution and for the Palu event with 1 m resolution. This makes them hardly comparable.

Page 29, lines 536 – 540: This argument is not enough to conclude that liquefaction was the principal cause for structural destruction. Mas et al. (2020) write that tsunami hydrodynamic, and debris impact forces may have been the principal causes of failure and collapse in Palu Bay's waterfront area.

Page 29, line 540: It is pure speculation that liquefaction episodes are mostly responsible for the building damage. Prove it.

The rest of the manuscript is based on this hypothesis. Consequently, you should present some facts or rewrite the last part of the discussion and conclusion.

References:

NHESSD
Mas, E., Paulik, R., Pakoksung, K., Adriano, B., Moya, L., Suppasri, A., ... & Koshimura, S. (2020). Characteristics of Tsunami Fragility Functions Developed Using Different Sources of Damage Data from the 2018 Sulawesi Earthquake and Tsunami. Pure and Applied Geophysics, 177(6), 2437-2455.

Dogan, G. G., Annunziato, A., Hidayat, R., Husrin, S., Prasetya, G., Kongko, W., ... & Yalciner, A. C. (2021). Numerical Simulations of December 22, 2018 Anak Krakatau Tsunami and Examination of Possible Submarine Landslide Scenarios. Pure and Applied Geophysics, 1-20.

Grilli, S. T., Tappin, D. R., Carey, S., Watt, S. F., Ward, S. N., Grilli, A. R., ... & Muin, M. (2019). Modelling of the tsunami from the December 22, 2018 lateral collapse of Anak Krakatau volcano in the Sunda Straits, Indonesia. Scientific reports, 9(1), 1-13.

Omira, R., & Ramalho, I. (2020). Evidence-Calibrated Numerical Model of December 22, 2018, Anak Krakatau Flank Collapse and Tsunami. Pure and Applied Geophysics, 177(7), 3059-3071.

Watkinson, I. M., & Hall, R. (2019). Impact of communal irrigation on the 2018 Palu earthquake-triggered landslides. Nature Geoscience, 12(11), 940-945.

Sassa, S., & Takagawa, T. (2019). Liquefied gravity flow-induced tsunami: first evidence and comparison from the 2018 Indonesia Sulawesi earthquake and tsunami disasters. Landslides, 16(1), 195-200.

Syifa, M., Kadavi, P. R., & Lee, C. W. (2019). An artificial intelligence application for post-earthquake damage mapping in Palu, Central Sulawesi, Indonesia. Sensors, 19(3), 542.

Gusman, A. R., Supendi, P., Nugraha, A. D., Power, W., Latief, H., Sunendar, H., ... & Daryono, M. R. (2019). Source model for the tsunami inside Palu Bay following the 2018 Palu earthquake, Indonesia. Geophysical Research Letters, 46(15), 8721-8730.

Rhie, J., D. Dreger, R. Burgmann, and B. Romanowicz. (2007). Slip of the 2004
Sumatra–Andaman Earthquake from joint inversion of long-period global seismic waveforms and GPS static Offsets, Bull. Seismo. Soc. Am., 97(1A):S115–S127

Ulrich, T., Vater, S., Madden, E. H., Behrens, J., van Dinther, Y., Van Zelst, I., ... & Gabriel, A. A. (2019). Coupled, physics-based modeling reveals earthquake displacements are critical to the 2018 Palu, Sulawesi Tsunami. Pure and Applied Geophysics, 176(10), 4069-4109.

Williams, R., Rowley, P., & Garthwaite, M. C. (2019). Reconstructing the Anak Krakatau flank collapse that caused the December 2018 Indonesian tsunami. Geology, 47(10), 973-976.

---

## Author Comment (AC1) · 27 Feb 2021

Dear Referee 1,

We would like to thank you for the time spent on our manuscript. We are very pleased that you highly evaluated our work. We highly appreciate your constructive comments and suggestions. You also pointed out the clarifications required to improve the original manuscript. We modified the manuscript according to your recommendations. Please find our answers and corrections below (all changes are highlighted in red in the manuscript).

- **Major comments**

| Reviewer comments | Our answers | Corrected manuscript |
|---|---|---|
| In Section 3.1.1, the two-layer model is explained. Can the authors explain more on how physically the layer 1 and layer 2 interact? From the equations, the interacting aspects of the two layers are expressed in terms of the water level gradients Z1 and Z2. So the flux Q1 in layer 1 is fluid, while the flux Q2 in layer 2 is soil mass? A figure would be useful addition for this section. | The flux Q1 is water while the flux Q2 is granular material (soil). We added more explanations and Fig. A1 to better understand the meaning of each term. | Line 167: $\rho_1$ and $\rho_2$ are the densities of the seawater and the landslide. The fifth term of the momentum equations (Eqs. 2, 3, 5, 6) represents the interaction between the two layers. The tsunami model…

 Line 170: …, respectively (Fig. A1 - Appendix A)

 **Please, see Fig. A1 below (Appendix A).** |
| Equation (7): please specify the units. | We thank the reviewer for pointing this out. We added the units as well as more explanations. | Line 189: $n_o$ corresponds to the Manning's roughness coefficient ($n_o = 0.025$ s.m$^{-1/3}$), $C_D$ represents the drag coefficient ($C_D = 1.5$ (Federal Emergency Management Agency (FEMA), 2003)) and the constant $d$ signifies the horizontal scale of buildings (~15 m). $\theta$ is the building occupation ratio in percent (0-100 %) for each computational cell of 20 x 20 m$^2$ and 1 x 1 m$^2$ resolutions for Sunda Strait and Palu areas, respectively. $\theta$ is obtained by computing the building area over each pixel using GIS data. The computational cell corresponding to buildings can be inundated by the $n$ Manning coefficient through the term $D$, which represents the simulated flow depth (m). In the urban areas of Sunda Strait and Palu, the average occupation ratios are 24 % and 84 % respectively (Fig. 2b,d). In non-residential area, we set the Manning's roughness coefficients inland and on the seafloor to 0.03 and 0.025 respectively, which are typical values for vegetated and shallow water areas (Kotani, 1998). |
| Figure 2 and Section 3.1.2: It is not clear how the computational cells that correspond to buildings (Figure 2d) can be inundated or not in tsunami simulation. | The computational cell corresponding to buildings can be inundated by the $n$ Manning coefficient through the last term of Eq (7): $D$, which represents the simulated flow depth (m). | Line 193: …using GIS data. The computational cell corresponding to buildings can be inundated by the $n$ Manning coefficient through the term $D$, which represents the simulated flow depth (m). In the urban… |
| In Section 3.2: could you comment on the vertical accuracies of the DEM/DSM used for the | Corrected. | Line 180: BATNAS and DEMNAS, Indonesia, provided the bathymetric and topographic data with 180 and 8 m- |

| | | |
|---|---|---|
| investigations? How were they derived? I would guess local LiDAR data? | | resolutions, respectively. The data was established from SAR images (http://tides.big.go.id/DEMNAS/index.html). Both datasets were resampled to three computational domains with a grid size of 20-m resolution (Fig. 2a,b). In Palu-City, the bathymetric and topographic data with 1-m resolution were obtained through Lidar images and supplied by the Agency for Geo-spatial Information (BIG), Indonesia (Fig. 2b,c). For tsunami inundation…

Line 211: To correct the Digital Surface Model (DSM), we removed the vegetation, buildings and infrastructures elevations based on the linear smoothing method and used the resulting Digital Elevation Model (1$^{st}$ DEM) as topography in the tsunami inundation model (Fig. 3). The vertical accuracy of the DSM/DEM is about 4 m. The 2018 Sunda Strait… |
| Throughout the investigations, were the tidal effects taken into account? For the 2018 Palu earthquake, the tidal levels have important contributions (e.g. Goda et al., 2019). In Section 3.2.3, how credible the landslide source model for the Palu event? For example, a detailed seismic source model can explain the majority portion of the observed tsunami in Palu Bay (e.g. Ulrich et al., 2019). How were the effects due to the coseismic deformation and tidal level considered (e.g. Goda et al., 2019)? In light of the missing elements in the tsunami source model, the landslide source model may be considered to be biased. I think this discussion is important for the NHESS journal audience This is a comment: the scatter plot shown in Figure 9 is not well correlated (i.e. simulation vs observation), which may be due to mis-specified tsunami source. | We agreed with the reviewer. The tsunami inundation model is now part of the discussion (Section 6.1). Contrary to Sunda, we took into account tidal effects in Palu. As mentioned by Pakoksung et al. (2019), TUNAMI-N2 does not reproduce the effect of seismic deformation. So, we considered that the 2018 Palu tsunami was triggered by subaerial/submarine landslides only (TUNAMI two-layer model). Furthermore, some observed and simulated flow depths are very different in Palu. To tackle this issue, we decided to set a confidence interval of 1 m to develop accurate curves. The observed and simulated curves based on the flow depth are relatively similar, so it shows the consistency of the 1-m confidence interval. | Line 233: We increased the mean sea level (MSL) by 2.3 m to reproduce the high tide during the 2018 Palu tsunami. As shown by Pakoksung et al. (2019), the observed waveform at Pantoloan tidal gauge does not fit the simulated one with the Finite Fault Model of TUNAMI-N2. Although recent studies show that seismic seafloor deformation may be the primary cause of the tsunami (Gusman et al., 2019; Ulrich et al., 2019), in this study, the main assumption is that the 2018 Sulawesi-Palu was triggered by subaerial/submarine landslides. According to Heidarzadeh et al. (2018), a large landslide to the north or the south of Pantoloan tidal gauge is responsible for the significant height wave recorded. Arikawa et al. (2018) also identified several sites of potential subsidence in the northern part of Palu-Bay. Based on these previous studies, we assume two large landslides: L1 and L2. Small landslides (S1-S12) also occurred in the bay; their location stands on observations from satellite imagery, field surveys and video footage (Arikawa et al., 2018; Carvajal et al., 2019) (Fig. 6). The trial and error method aims to achieve the volume of the landslides (Table 3). In Figure 7, the submarine landslides model reproduces well the tsunami observations at Pantoloan. The calibration … |

| | | **Please, see the added Section 6.1.** |
|---|---|---|
| Also note that 'Figure 9' is misspelled. | Thank you very much. We corrected it. | Figure 9. Comparison between observed and simulated flow depths at damaged building for a S8 ratio of 1.2; a confidence interval is set at 1-m flow depth. |
| Page 13: Can other link functions other than probit be used? | We thank the reviewer for this request. We decided to include the sensitivity analysis of the statistical model based on the link function. Following the GEM guidelines, we considered overall three functions: the probit, logit and cloglog. Overall, the choice of link function does not change the discussion. It was found that the probit function fits the Sunda data best. The logit function fits the Palu and Thailand data best. The change in the link function does not notably change the shape of the fragility curves. | **Please, see the revised Sections 4.1 and 4.2, and the updates in Appendix C and Appendix D.** |
| Figure 10 (and other figures as well): Can the data also be displayed? Can the authors clarify the confidence interval indicates the confidence interval of the regression line or the prediction interval of the prediction model? I think by including the data points in Figure 10, this becomes obvious. I think this clarification is important because the number of data is small. | We thank the reviewer for this comment. Figures 10-12, which we believe the reviewer refers to, are part of the exploratory analysis. The sole aim is to show trends in the data which will be useful to construct the statistical model in the following section. All we need to see is whether the intercept and/or the slope of a best estimate curve changes for different variables. For this reason, we use the inverse of the cumulative standard normal distribution in the y axis and the natural logarithm of the tsunami intensity in the x axis. To present the data points will mean to estimate the inverse standard normal cumulative distribution function of the probability that a given building will experience a given damage state or above. For our case, we have building-by-building damage data therefore this probability (e.g., $P(DS \geq ds_1 |\text{Flow depth})$) is either 0 or 1 for which the inverse of the standard normal cumulative distribution function is not defined. For this reason, we did not present the data points. Instead, we updated the text to avoid confusion. | Line 319: … curves. The confidence in the exact shape of the mean curves is estimated and presented in terms of the 90 % confidence intervals around the best-estimate curves. |
| Section 5, Line 430: I do not understand the intention of showing the tsunami fragility models based on simulated intensity values? When the tsunami simulations are calibrated | We understand the point of view of the reviewer. We highlighted the benefits of the simulated fragility curves in the abstract and the introduction. The main reason is | **Please, see the revised abstract and the last paragraph of the introduction.** |

| | | |
|---|---|---|
| reasonably well with the observations, using the same damage data, the fitted fragility models are expected to be similar (as demonstrated in Figure 13). But I do not see the benefit of using the simulated tsunami intensity values unless the authors use the damage data where the observations are not available and thus the tsunami intensity values need to be estimated. But this work does not investigate this aspect. Altogether the simulated cases can be removed. | that 2018 Sunda Strait and Sulawesi-Palu tsunamis are uncommon events still poorly understood compared to the 2004 IOT. The flow depth is the only tsunami intensity measure recorded during the field surveys. So, to improve our understanding of the structural damage caused by the Sunda Strait and Sulawesi-Palu tsunamis and to discuss the impact of wave period, ground shaking and liquefaction events, we reproduce their tsunami intensity measures (i.e., flow depth, flow velocity and hydrodynamic force). Moreover, this is the first attempt to develop fragility curves as functions of the flow depth, the flow velocity and the hydrodynamic force for the 2018 Sunda Strait and 2018 Palu tsunamis based on TUNAMI two-layer model. | |
| Figure 13: as discussed by the authors, the fragility functions based on flow velocity and (probably) hydrodynamic force do not show realistic features and thus not really useful. It may be useful to show such results for one case but for other cases, they are not really useful, especially for flow velocity. My concern is that careless readers may attempt to use such models as black box models. | We thank and agreed with the reviewer. In the discussion, we discussed whether the tsunami intensity measures are efficient predictors of damage (Section 6.1). The flow velocity and the hydrodynamic force (please, see the drag force formula) are not providing a good description of the tsunami damage, compared to the flow depth. This is a valid contribution to the field. Therefore, the $2^{nd}$ part of the discussion (Section 6.2) is based on the curves function of the flow depth only. Careful readers should rather use fragility curves based on observation as they are of higher quality. | Line 172: … during the tsunami inundation. The hydrodynamic force acting on buildings and infrastructure is defined as the drag force per unit width of the structure (Koshimura et al., 2009). $$F = \frac{1}{2} C_D \rho u^2 D$$ $C_D$ represents the drag coefficient ($C_D = 1.0$), $\rho$ is the water density ($\rho = 1000$ kg/m$^3$), $u$ stands for the current velocity (m/s), and $D$ is the inundation depth (m). **Please, see the added Section 6.1 and the revised Section 6.2.** |
| Figure 14: why the data are only shown for x values greater than 1? Should they start with the theoretical constraints that zero fragility for zero hazard values? My concern is again that careless users may take such unrealistic models as they are. | We thank the reviewer for pointing this out. It does not mean that there is no potential damage between 0-1 m flow depths or 0-1 m/s flow velocity. The reason is that we do not have data to predict the shape of the curves. | Line 469: …DB_Sunda2018'. In Fig. 14a,b, there is no data to predict the shape of the curves between 0-1 m and 0-1 m/s. The curves…. |
| Figure 15: I understand that the results are based on statistical fitting but these curves do not look realistic. Are they reliable? I think the reliability of the curves should be a part of the discussion (beyond the statistical confidence level etc). Can one use these functions reliably? Figure 16: From my perspectives, the | We agreed with the reviewer. The reliability of the curves is discussed in Section 6.1. Compared to the flow depth, the flow velocity and the hydrodynamic force are not good predictors of damage. For this reason, we are not discussing the building damage probability based on these tsunami intensity measures | **Please, see the added Section 6.1.** |

| comparison of the curves based on flow velocity and hydrodynamic force is not robust. I would suggest focusing on the flow depth based models which show some realistic fragility features. | for the 2018 Sunda Strait and Palu tsunami. | |
|---|---|---|

- **Minor comments**

| Reviewer comments | Our answers | Corrected manuscript |
|---|---|---|
| Page 1, Line 18: cumulative distribution functions -> delete cumulative distribution. Strictly speaking, the fragility function is not the cumulative distribution function and this expression is confusing. I would suggest deleting 'cumulative distribution'. There are a few places that have the same expression. | We are very sorry for this confusing expression and we corrected it. | These  functions express the likelihood of a structure reaching or exceeding a damage state in response to a tsunami hazard intensity measure. |
| Page 1, Line 28: 'liquefaction events: : :' The majority of the damage and loss during the Palu earthquake was due to slope failures (which involve liquefaction as physical failure mechanism). It is not clear (especially in the abstract), this 'liquefaction' refers to the slope failure cases (e.g. Petobo) or the flat coastal area along Palu Bay. Given the nature of this event, it would be better to rewrite this sentence to be more specific which area/incidences the authors are referring to. | We are very sorry and cleared this part. Here, we mentioned liquefaction events related to ground failures in the waterfront of Palu-City. We also made the distinction with the slope failure cases observed inland in Section 6.2 (e.g., Petobo, Jono and Balaroa). | Abstract: Similar to the Banda Aceh case, the Sulawesi-Palu tsunami load may not be the only cause of structural destruction. The buildings susceptibility to tsunami damage in the waterfront of Palu-City could have been enhanced by liquefaction events triggered by the 2018 Sulawesi earthquake.

**Please, see the revised Section 6.2.**

Conclusion: The Sulawesi-Palu tsunami is a complex event as it may not be the only cause of structural destruction. The 2018 Sulawesi earthquake caused minor damage to buildings and most importantly could have triggered liquefaction events in the waterfront of Palu-City (e.g., coastal retreats) increasing the building susceptibility to tsunami damage. |
| Page 1, Line 38: vertical -> vertical and horizontal. | Corrected | ...causing horizontal and vertical movement of the ocean floor… |
| Page 1, Line 41: period -> periods. | Corrected | …longer wave periods attacking the coast … |
| Page 2, Line 44: were -> was. | Corrected | …strong ground shaking was reported … |
| Page 2, Line 49: few -> a few. | Corrected | After a few months … |
| Page 2, Line 50: delete finally. | Corrected | …the Anak Krakatau Volcano  erupted … |
| Page 2, Line 60: reported to -> reported at. | Corrected | …the wave height reported at the Pantoloan tidal gauge… |
| Page 2, Line 60: what is 'largely exceeded'? The meaning is not clear. | Corrected | … The fault mechanism did not suggest that the tsunami would be so destructive. The wave reached rapidly Palu (~8 min), implying that its source was inside or near the bay |

| | | (Muhari et al., 2018; Omira et al., 2019). Its short wave… |
|---|---|---|
| Page 2, Line 62: assumption -> hypothesis (I think hypothesis is more appropriate). | Corrected | … the main hypothesis is that … |
| Page 2, Line 68: The sentence 'Koshimura et al. : : :' reads strangely in a sense that the tsunami fragility concept existed before this work. I agree that the work by Koshimura et al. was very influential. | Corrected | The term "tsunami fragility" is a new measure to estimate structural damage and casualties caused by a tsunami, as mentioned by Koshimura et al., 2009b. |
| Page 2, Line 69: delete 'cumulative distribution'. | Corrected | Tsunami fragility curves are cumulative distribution functions expressing |
| Page 3, Line 86: treated -> analyzed. | Corrected | … are analysed separately … |
| Page 3, Line 93: exposed -> investigated. | Corrected | … are investigated. |
| Page 5, first line: are -> is. | Corrected | … is ignored … |
| Page 14, Line 284: appear -> appears. | Corrected | … as the two curves appears to … |
| Page 19, Line 384: depicted -> listed or summarized. | Corrected | … are listed in … |
| Page 20, Line 398: identical curves -> identical slopes? | Corrected | … identical slopes … |

[Figure]

**Figure A1. Two-layer modelling of a subaerial/submarine landslide (from the original sketch of Pakoksung et al., 2019), (a) pre-failure, (b) generation of negative and positive waves due to the landslide and (c) landslide in progress and wave propagation.**

**References:**

Arikawa, T., Muhari, A., Okumura, Y., Dohi, Y., Afriyanto, B., Sujatmiko, K. A. and Imamura, F.: Coastal subsidence induced several tsunamis during the 2018 Sulawesi earthquake, Journal of Disaster Research, 13, 1–3, doi:10.20965/jdr.2018.sc20181204, 2018.

Carvajal, M., Araya- Cornejo, C., Sepúlveda, I., Melnick, D. and Haase, J. S.: Nearly instantaneous tsunamis following the Mw 7.5 2018 Palu earthquake, Geophysical Research Letters, 46(10), 5117–5126, doi:10.1029/2019GL082578, 2019.

Federal Emergency Management Agency (FEMA): Coastal construction manual, FEMA 55, Third Edition (FEMA 55), 296p., 2003.

Gusman, A. R., Supendi, P., Nugraha, A. D., Power, W., Latief, H., Sunendar, H., Widiyantoro, S., Wiyono, S. H., Hakim, A. and Muhari, A.: Source model for the tsunami inside palu bay following the 2018 palu earthquake, Indonesia, Geophysical Research Letters, 46(15), 8721–8730, doi:10.1029/2019gl082717, 2019.

Heidarzadeh, M., Muhari, A. and Wijanarto, A. B.: Insights on the source of the 28 September 2018 Sulawesi tsunami,

Indonesia based on spectral analyses and numerical simulations, Pure and Applied Geophysics, 176(1), 25–43, doi:10.1007/s00024-018-2065-9, 2018.

Koshimura, S., Namegaya, Y. and Yanagisawa, H.: Tsunami fragility - A new measure to identify tsunami damage, Journal of Disaster Research, 4(6), 479–488, doi:10.20965/jdr.2009.p0479, 2009.

Kotani, M.: Tsunami run-up simulation and damage estimation using GIS, Pacific Coast Engineering, Japan Society of Civil Engineers (JSCE), 45, 356–360, 1998.

Muhari, A., Imamura, F., Arikawa, T., Hakim, A. R. and Afriyanto, B.: Solving the puzzle of the September 2018 Palu, Indonesia, tsunami mystery: clues from the tsunami waveform and the initial field survey data, Journal of Disaster Research, 13(Scientific Communication), sc20181108, doi:10.20965/jdr.2018.sc20181108, 2018.

Omira, R., Dogan, G. G., Hidayat, R., Husrin, S., Prasetya, G., Annunziato, A., Proietti, C., Probst, P., Paparo, M. A., Wronna, M., Zaytsev, A., Pronin, P., Giniyatullin, A., Putra, P. S., Hartanto, D., Ginanjar, G., Kongko, W., Pelinovsky, E. and Yalciner, A. C.: The September 28th, 2018, tsunami in Palu-Sulawesi, Indonesia: a post-event field survey, Pure and Applied Geophysics, 176(4), 1379–1395, doi:10.1007/s00024-019-02145-z, 2019.

Pakoksung, K., Suppasri, A., Imamura, F., Athanasius, C., Omang, A. and Muhari, A.: Simulation of the submarine landslide tsunami on 28 September 2018 in Palu Bay, Sulawesi Island, Indonesia, using a two-layer model, Pure and Applied Geophysics, 176(8), 3323–3350, doi:10.1007/s00024-019-02235-y, 2019.

Ulrich, T., Vater, S., Madden, E. H., Behrens, J., van Dinther, Y., van Zelst, I., Fielding, E. J., Liang, C. and Gabriel, A. A.: Coupled, physics-based modeling reveals earthquake displacements are critical to the 2018 Palu, Sulawesi tsunami, Pure and Applied Geophysics, 176(10), 4069–4109, doi:10.1007/s00024-019-02290-5, 2019.

---

## Author Comment (AC2) · 27 Feb 2021

Dear Referee 2,

We would like to thank you for the time spent on our manuscript. We are very pleased that you highly evaluated our work. We highly appreciate your constructive comments and suggestions. You also pointed out the clarifications required to improve the original manuscript. We modified the manuscript according to your recommendations. Please find our answers and corrections below (all changes are highlighted in red).

- **Major comments**

| Reviewer comments | Our answers | Corrected manuscript |
|---|---|---|
| The authors present the fragility curves for the three events in the four locations (Fig. 16a). The curves demonstrate that for low flow depth values (less than 2 m flow depth) building fragility was largest in Palu 2018 followed by 2004 Banda Aceh, followed by 2004 Khao Lak, followed by Sunda Strait 2018. Above 2 m flow depth, the curves for 2004 Banda Aceh demonstrate the largest building fragility. The authors conclude that ground shaking and liquefaction contributed to the fragility curves for Palu 2018 and Banda Aceh 2004. Although it is possible that both ground shaking and liquefaction may contribute to the fragility curves, the authors do not demonstrate this. Hence their claim is pure speculation. To demonstrate that ground shaking played a significant role, the authors should present a measure of ground shaking that allows inferring the damage of buildings that have not been hit by the tsunami. Maybe they could use seismic intensities or peak ground acceleration to infer the damage of buildings that have not been hit by the tsunami. Alternatively, they could comment on which extent buildings were damaged outside the tsunami inundation area to foster their hypothesis. | We thank the reviewer for this comment. In Banda Aceh, the ground acceleration of the 2004 Indian Ocean earthquake was not recorded in the damage zone. The earthquake intensity is estimated to VII to VIII on the Modified Mercalli Scale. Ghobarah et al. (2005) also mentioned that buildings in Banda Aceh was strongly affected by the ground shaking (which lasted ~10 min) and tsunami damage was distinguished from seismic damage (e.g., substantial damage to infrastructure with 3-5 stories compared to low rise structures). For the 2018 Sulawesi-Palu tsunami, the main cause of structural damage is still investigated. The earthquake intensity is estimated to VII to VIII on the Modified Mercalli Scale (Supendi et al., 2019) but Kijewski-Correa and Robertson (2018) mentioned that the ground motion only slightly damaged the buildings in Palu-City. | **Please, see the revised Section 6.2.** |
| The authors write in the conclusions, page 30, line 573 f. : ': : :, it is clearly demonstrated that liquefaction events can increase building susceptibility.' and page 30, line 574 f. : ': : :, the building were previously affected by severe liquefaction episodes.' This sentence is not a conclusion from their work. Most importantly is to mention that the largest liquefaction areas were located outside the tsunami inundation areas. The | We thank and agreed with the reviewer. In this study, the term liquefaction refers to the ground failures in the waterfront of Palu-City. Even though the largest liquefaction areas were recorded outside the tsunami inundation zone (Watkinson and Hall, 2019), Sassa and Takagawa (2019) and Kijewski-Correa and Robertson (2018) observed land retreats along coastal areas of Palu-City, which is highly | **Please, see the revised Section 6.2 and Fig. 17 below.** |

| | | |
|---|---|---|
| authors should consult Watkinson and Hall (2019) and Syifa et al. (2019). Even though Sassa and Takagawa (2019) conclude that they found evidence for extensive liquefaction in coastal areas, the authors should quantify how many of the database's observed buildings were affected by liquefaction. The authors could overlap the liquefaction areas with figure 8 to see how many of the buildings were affected. The current state of the manuscript does not allow to conclude that buildings were weakened by liquefaction. Moreover, Mas et al. (2020) write that tsunami hydrodynamic and debris impact forces may have been the principal causes of failure and collapse in Palu Bay's waterfront area. | vulnerable to liquefaction disaster (Darma and Sulistyantara, 2020; Kijewski-Correa and Robertson, 2018). In Palu post-tsunami database (DB_Palu2018), many masonry-type buildings do not have a flow depth value because they have been washed away. In Figure 17, many masonry-type buildings completely damaged are very close to these coastal retreats. Moreover, the likelihood of complete damage is very high for low inundation depth levels. This feature is usually observed for building suffering prior damage (e.g., ground shaking and/or liquefactions episodes), as mentioned by Charvet et al. (2014), for the 2011 Great East Japan event. Here, the likelihood of complete damage is higher in Palu than in Banda Aceh under 2-m flow depth. So, even if ground shaking is not the main cause of destruction, it may have triggered liquefactions in the waterfront of Palu-City and enhanced the building susceptibility to tsunami damage. This assumption cannot be verified through satellite images, it needs direct and close observations, which might have been erased by the tsunami. On the other hand, Mas et al. (2020) suggested that the tsunami hydrodynamic or debris impact might be the main cause of structural destruction in the waterfront area of Palu-Bay. As the flow velocity and the hydrodynamic force are not good descriptors of tsunami damage, we cannot support this assumption. | |
| The conclusion that the building fragility curves for Banda Aceh and Khao Lak are different because of the ground shaking in Banda Aceh are incomplete. Just because the locations were hit bit the same tsunami event, does not necessarily mean that the wave period was the same in both locations. The rupture at the Sunda Megathrust was longer than 1000 km, and slip rates along the fault were heterogeneous (Rhie et al. 2007, Koshimura et al. 2009). Consequently, waves with different periods and hydrodynamic features may have impacted Khao Lak and Banda Aceh. Applying numerical models for both sites could show the | We agreed with the reviewers. We compared the tsunami waveform at both locations and computed the wave period. Along Banda Aceh shores, the simulated tsunami wave period is ranging from 40 to 45 min (Prasetya et al., 2011; Puspito and Gunawan, 2005) and the one simulated in Khao Lak/Phuket is estimated to approximatively 40 min (Karlsson et al., 2009; Puspito and Gunawan, 2005; Tsuji et al., 2006). Therefore, the tsunami periods are very similar at both locations, and are unlikely responsible for the difference between Banda Aceh and Khao Lak/Phuket curves. In Figure 16a, the building resilience is higher | **Please, see the revised Section 6.2.**

Line 577: The city of Banda Aceh and Khao Lak/Phuket area have been damaged by the 2004 IOT. Along Banda Aceh shores, the simulated tsunami wave period is ranging from 40 to 45 min (Prasetya et al., 2011; Puspito and Gunawan, 2005) and the one simulated in Khao Lak/Phuket is estimated to approximatively 40 min (Karlsson et al., 2009; Puspito and Gunawan, 2005; Tsuji et al., 2006). |

| | | |
|---|---|---|
| differences, but the authors do not present tsunami simulations for Banda Aceh and Khao Lak. | in Khao Lak/ Phuket than in Banda Aceh. It comes from the fact that the Khao Lak/Phuket curve is developed for reinforced concrete buildings while the ones in Banda Aceh are produced for mixed buildings (Koshimura et al., 2009a). Another reason is that seismic damages due to the 2004 Indian Ocean earthquake were not recorded in Khao Lak/Phuket before the tsunami arrival. Furthermore, in Banda Aceh, the likelihood of complete damage is very high for low inundation depth levels; this feature is usually observed for building suffering prior damage (e.g., ground shaking and/or liquefactions episodes) (Charvet et al., 2014). | |
| Regarding the numerical modelling in the manuscript, the authors should comment on why they use modelling for the 2018 Palu and Sunda Strait events but not for the 2004 Indian Ocean tsunami. I believe the authors could draw interesting conclusions if they would compare impacting wave shapes for the 2004 Indian Ocean tsunami in Khao Lak and Banda Aceh. | We thank the reviewer for this comment. We highlighted the benefits of the simulated fragility curves in the abstract and the introduction. The main reason is that 2018 Sunda Strait and Sulawesi-Palu tsunamis are uncommon events still poorly understood compared to the 2004 IOT. The flow depth is the only tsunami intensity measure recorded during the field surveys. So, to improve our understanding of the structural damage caused by the Sunda Strait and Sulawesi-Palu tsunamis and to discuss the impact of wave period, ground shaking and liquefaction events, we reproduced their tsunami intensity measures. This is also the first attempt to develop fragility curves as functions of the flow depth, the flow velocity and the hydrodynamic force of the Sunda Strait and Palu tsunamis based on TUNAMI two-layer model. | **Please, see the revised abstract and the last paragraph of the introduction.** |
| Further, the authors should clearly state the motivation for their numerical modelling efforts. It is not clear why they use models where many observations exist. | Please, see the explanations above. | **Please, see the revised abstract and the last paragraph of the introduction.** |
| The Digital Elevation Model (DEM) for Palu has a resolution of 1 m. The DEM for the Sunda Strait event has a resolution of 20 m. If the authors want to compare the cases, they should use the same cell size for DEMs or explain why they believe that simulations are comparable | We thank the reviewer for pointing this out. The resolution of the DEM is now part of the discussion (Section 6.1). In Sunda Strait and Palu, we performed simulations with bathymetry and topography of 20x20 m² and 1x1 m² respectively. For both events, the curves based on the | Line 180: BATNAS and DEMNAS, Indonesia, provided the bathymetric and topographic data with 180 and 8 m-resolutions, respectively. The data was established from SAR images (http://tides.big.go.id/DEMNAS/index.html). Both datasets were resampled to three computational |

| | | |
|---|---|---|
| when using 400 times bigger cell size. Apart from that, I believe the reader would be interested in the data that allows building a DEM with 1 m resolution for Palu. The authors should also name the references for the dataset used in all DEMs. | observed and simulated flow depths are similar, so we are confident to compare Sunda Strait and Palu curves based on the flow depth. However, De Risi et al. (2017) illustrated well the influence of the DEM resolution on the efficiency of the flow velocity as a tsunami intensity measure. In Sunda Strait, the DEM resolution is relatively high (20 m), this is one of the limitations in this study and it could explain why the flow velocity is not a good descriptor of tsunami damage. Therefore, we cannot compare the Sunda Strait curve based on the flow velocity with the one for Palu, where we perform two-layer numerical modelling using the finest grid size of 1 m. We also added the references for the DEMs. | domains with a grid size of 20-m resolution (Fig. 2a,b). In Palu-City, the bathymetric and topographic data with 1-m resolution were obtained through Lidar images and supplied by the Agency for Geo-spatial Information (BIG), Indonesia (Fig. 2b,c). For tsunami inundation… **Please, see the added Section 6.1.** |
| The authors use hypothetical landslide sources for the 2018 Palu event. Those are not in agreement with some other published studies (Ulrich et al. 2019, Gusman et al. 2019). In figure 6, the authors present those hypothetical landslides as principal tsunami source without explaining why they have this assumption. The authors must include a review of previously published sources and comment on the reasons for modifying the sources in their manuscript. In figure 9, they compare the observed with simulated flow depth and claim that their model is in good agreement with the observations. To my understanding, figure 9 in the manuscript demonstrates that the simulation does not match the observation. The authors must explain why they believe the model is of good quality. Further in the manuscript's conclusion, the authors write on page 30, line 574 f.: 'Although Palu-Bay was hit by a non-seismic tsunami: : :.'. These are the authors' assumptions and therefore are not valid as a conclusion and need to be rewritten. Please also note that figure 9 is misspelt. | We agreed with the reviewer. We clarified the choice of the tsunami inundation model, which is also part of the discussion (Section 6.1). The conclusion has been rewritten too. In Figure 9, some observed and simulated flow depths are different in Palu-City. To tackle this issue, we decided to develop Palu curves based on the flow depth data included in the 1-m confidence interval only. The observed and simulated curves based on the flow depth are relatively similar, especially for $ds_1$ and $ds_3$, so we believe that the curves based on the simulated flow depth are accurate enough. | Line 233: We increased the mean sea level (MSL) by 2.3 m to reproduce the high tide during the 2018 Palu tsunami. As shown by Pakoksung et al. (2019), the observed waveform at Pantoloan tidal gauge does not fit the simulated one with the Finite Fault Model of TUNAMI-N2. Although recent studies show that seismic seafloor deformation may be the primary cause of the tsunami (Gusman et al., 2019; Ulrich et al., 2019), in this study, the main assumption is that the 2018 Sulawesi-Palu was triggered by subaerial/submarine landslides. According to Heidarzadeh et al. (2018), a large landslide to the north or the south of Pantoloan tidal gauge is responsible for the significant height wave recorded. Arikawa et al. (2018) also identified several sites of potential subsidence in the northern part of Palu-Bay. Based on these previous studies, we assume two large landslides: L1 and L2. Small landslides (S1-S12) also occurred in the bay; their location stands on observations from satellite imagery, field surveys and video footage (Arikawa et al., 2018; Carvajal et al., 2019) (Fig. 6). The trial and error method aims to achieve the volume of the landslides (Table 3). In Figure |

| | | |
|---|---|---|
| | | 7, the submarine landslides model reproduces well the tsunami observations at Pantoloan. The calibration … |
| | | Figure 9. Comparison between observed and simulated flow depths at damaged building for a S8 ratio of 1.2; a confidence interval is set at 1-m flow depth. |
| | | **Please, see the added Section 6.1 and the revised conclusion.** |
| The authors should include a paragraph on proposed flank collapse sources from other studies on the 2018 Sunda Strait tsunami. Please include Williams et al. (2019), Grilli et al. (2019), Omira and Ramalho (2020), Dogan et al. (2021). | We thank the reviewer and included the references. | Line 56: The tsunami generation process is unclear. The subaerial/submarine landslide volume is still investigated and ranges between 0.10 and 0.30 km$^3$ according to recent studies (Dogan et al., 2021; Grilli et al., 2019; Omira and Ramalho, 2020; Paris et al., 2020; Williams et al., 2019). Almost… |
| The authors write that they automatically corrected the flow depth traces for the 2018 Sunda Strait event. It is not clear how the authors do that. Are they using GPS field measurements or LIDAR data? If the authors use a method previously presented, then they must cite the corresponding reference. They observe a mean difference of flow depth values of 0.28 m for 94 traces. How far is this value representative for the 94 traces? It is not clear if the authors use this value for correction? I suggest rewriting this section. The resolution of the DEM is 20 m. Do the authors believe this resolution is sufficient to obtain reasonable values of flow depth and flow velocity? | We agreed with the reviewer and rewrote the section. The resolution of the DEM is part of the discussion (please, see the explanations above). The Digital Surface Model (DSM) was established from SAR images. We removed the vegetation, buildings and infrastructures elevations based on the linear smoothing method. However, in some areas, the simulated flow depths are underestimated compared to the observed ones (mean difference of 0.28 m +/- 1 m). Consequently, we corrected the DEM once again by removing 0.28 m at buildings using QGIS. Based on the 2$^{nd}$ DEM, which is more reliable at buildings, we achieved "good agreement" for the 2018 Sunda Strait tsunami model. | Line 180: BATNAS and DEMNAS, Indonesia, provided the bathymetric and topographic data with 180 and 8 m-resolutions, respectively. The data was established from SAR images (http://tides.big.go.id/DEMNAS/index.html). Both datasets were resampled to three computational domains with a grid size of 20-m resolution (Fig. 2a,b). In Palu-City…

 Line 211: To correct the Digital Surface Model (DSM), we removed the vegetation, buildings and infrastructures elevations based on the linear smoothing method and used the resulting Digital Elevation Model (1$^{st}$ DEM) as topography in the tsunami inundation model (Fig. 3). The vertical accuracy of the DSM/DEM is about 4 m. The 2018 Sunda Strait…

 Line 221: …difference of 0.28 m $\pm$ 1 m. Using QGIS software, we smoothed the 1$^{st}$ DEM to remove these mean difference in elevation at buildings where the flow depth is underestimated. The resulting DEM (2$^{nd}$ DEM) provides a topography more reliable (Fig. 3). Three cross-sections along the Sunda Strait coasts show the different corrections |

| | | applied to the DSM (Fig. 4). *K* and *κ*… |
| --- | --- | --- |
| | | Update of Figure 4 caption. (a) Cross-sections along Sunda Strait coasts. One cross section is realized in the computational areas (b,e) 1, (c,f) 2 and (d,g) 3 to illustrate the topographic corrections applied to the Digital Surface Model (DSM) using QGIS (background ESRI and © Google Maps). |
| | | **Please, see the added Section 6.1.** |
| Consistency with abbreviations and variables. Sometimes the authors use for Indian Ocean tsunami IOT (e.g. page 1, line 23, line 31; page 3, line 91; etc.) sometimes they use IO (e.g. page 2, line 81; page 4, line 101; page 20, line 416). The authors use GEM as an abbreviation on page 2, line 82 but only introduce the Global Earthquake Model (GEM) on page 13. The authors also use the abbreviation AIC without stating what the abbreviation stands for (page 18 ff.). In 4.1 the authors use the Greek letter pi as probability, and in 4.2 they use P. What is the difference? | We are sorry and corrected it. | Please, see the changes in the manuscript. |

- **Minor comments**

| Reviewer comments | Our answers | Corrected manuscript |
| --- | --- | --- |
| Page 2, line 51: Some other studies consider less volume and other particularities of the collapse. Please include them (Williams et al. 2019, Grilli et al. 2019, Omira and Ramalho 2020, Dogan et al. 2021). | Corrected | Please, see the changes in "Major comments" table. |
| Page 2, line 81: IO or IOT? Please be consistent. | Corrected | Please, see the changes in "Major comments" table. |
| Page 2, line 82: Please indicate what the abbreviation GEM stands for. | Corrected | Please, see the changes in the manuscript. |
| Page 3, line 91: I suggest rewriting the sentences since seismic and non-seismic curves is not clear. | We agreed with the reviewer and rewrote this sentence. | Line 96: Then, we compared the fragility curves of the Sunda Strait, Sulawesi-Palu and Indian Ocean (Khao Lak/Phuket) tsunamis with the curves of the 2004 IOT, in Banda Aceh, Indonesia, produced by Koshimura et al. (2009a). |
| Page 4, line 101: IO or IOT? Consistency! | Corrected | Please, see the changes in the manuscript. |
| Page 4, line 106: Why do the authors believe the databases are statistically representative since they explain later in section 4 that they use reduced | We thank the reviewer for this. We acknowledge the confusion that might cause to the reader. We decided to change the section and to | **Please, see the revised Sections 2 and 4.** |

| | | |
|---|---|---|
| samples of the databases DB_Palu2018' and DB_Sunda2018'. For example, from 463 observations in Palu they use 124 observation. I recommend restructuring the manuscript combining section 2 and 4 or put the final databases used in section 2. Is the number of timber buildings enough to be statistically significant? | reflect the numbers which we will actually use to section 4 and to provide the reader explanations of why this course of actions is taken, where to find further explanations and what is the expected impact on the shape of the fragility curves in section 4. We decided against merging the two sections as we need to present the databases first since they are being used in section 3 to validate the simulations. | |
| Page 5, line 142 f.: Please specify the appropriate kinematic and dynamic boundary conditions for the interfacing layers. | The flux Q1 is water while the flux Q2 is granular material (soil). We added more explanations and a figure to better understand the meaning of each term. | Line 167: $\rho_1$ and $\rho_2$ are the densities of the seawater and the landslide. The fifth term of the momentum equations (Eqs. 2, 3, 5, 6) represents the interaction between the two layers. The tsunami model…

 Line 170: …, respectively (Fig. A1 - Appendix A)

 **Please, see Fig. A1 below (Appendix A).** |
| Page 6, Eq. 7 ff.: In equation 7 is d a constant? Please define theta. | We are sorry for this oversight. We corrected it. $d$ is a constant. | Line 189: $n_o$ corresponds to the Manning's roughness coefficient ($n_o$ = 0.025 $s.m^{-1/3}$), $C_D$ represents the drag coefficient ($C_D$ = 1.5 (Federal Emergency Management Agency (FEMA), 2003)) and the constant $d$ signifies the horizontal scale of buildings (~15 m). $\theta$ is… |
| Page 7, figure 2: It is not clear what the building occupation ratio is. Please introduce a definition. What do the polygons in 2(d) represent? Please add a legend. Are those cells 100% covered? What about the rest of the layer? 0% occupation? Please clarify! | We are sorry and corrected it. Through GIS, we delimited each building, then we computed the building occupation ratio over a pixel (20x20 m² in Sunda Strait and 1x1 m² in Palu-City). We defined the building occupation ratio as the building area per pixel. In non-residential areas (building occupation ratio = 0 %), we set the Manning's roughness coefficients inland and on the seafloor to 0.03 and 0.025 respectively, which are typical values for vegetated and shallow water areas (Kotani, 1998). In Figure 2d, the polygon corresponds to 1x1 m² pixels with an occupation ratio of 100 % (dark red pixels). | Line 192: $\theta$ is obtained by computing the building area over each pixel using GIS data. The computational cell corresponding to buildings can be inundated by the $n$ Manning coefficient through the term $D$, which represents the simulated flow depth (m). In the urban areas of Sunda Strait and Palu, the average occupation ratios are 24 % and 84 % respectively (Fig. 2b,d). In non-residential area, we set the Manning's roughness coefficients inland and on the seafloor to 0.03 and 0.025 respectively, which are typical values for vegetated and shallow water areas (Kotani, 1998).

 **Please, see the revised Fig. 2 below.** |
| Page 7, line 180: Please avoid having two letters for the same variable. | Corrected | Line 200: …$K$  and $\kappa$  proposed… |
| Page 8, figure 3: It is unclear how the corrections are applied to the Digital Surface Model. | Corrected | Please, see the changes in "Major comments" table. |

| | | |
|---|---|---|
| Page 9, figure 4: What do mean by profile realized? Are those measurements? | We changed the term "profile" by "cross-sections". Using QGIS (Terrain Profile tool), we did three cross-sections showing the DSM, the 1st DEM and the 2nd DEM to illustrate the different topographic corrections. They are not field measurements. | Line 221: … Using QGIS software, the 1st DEM is smoothed to remove the elevation difference at buildings where the flow depth is underestimated. The resulting DEM (2nd DEM) provides a topography more reliable at buildings (Fig. 3). Three cross-sections along the Sunda Strait coasts show the different corrections applied to the DSM (Fig. 4). *K* and *κ*...

 Update of Figure 4 caption. (a) Cross sections along Sunda Strait coasts. Using QGIS, one cross section is realized in the computational areas … |
| Page 10, figure 5: Do the triangles represent single buildings? It is probably better to choose a representative area on a scale with many surveyed buildings like a city or village instead of large parts of the coast. The figure now does not illustrate well the flow depth close to the buildings. What about the flow velocity plots? | We agreed with the reviewer and added the flow velocity plots in Appendix B. Each triangle corresponds to a single building. | Update of Figure 4 caption: …to illustrate the topographic corrections applied to the Digital Surface Model (DSM) at buildings (a triangle corresponds to a building), using QGIS (background ESRI and © Google Maps)

 **Please, see the revised Figs. 5 and B1 (Appendix B) below.** |
| Page 10, line 213: Is landslide S8 oriented towards the city? Isn't that the slide that was captured by the pilot in the departing plane? Isn't the slide direction perpendicular to the bay? | Yes | Line 248: …and (iii) it is close and ideally oriented to Palu-City; the slide direction, captured by an aircraft pilot, is perpendicular to the bay (Carvajal et al., 2019). The density… |
| Page 10, line 215 f.: What is meant by landslide ratio of 1.2? | Corrected | Line 252: For a landslide ratio of 1.2 (i.e., S8 volume is multiplied by 1.2), the tsunami model… |
| Page 10, line 217 f.: Why do you only overlay 175 traces? | In the west part of our computational zone, the simulated envelope is shorter than the surveyed one. For this reason, we overlaid 175 buildings while the surveyed tsunami envelope covers 220 buildings. | Line 253: … (a = 1.027). The simulated tsunami inundation zone overlays 175 traces out of 371 because (i) 151 buildings with flow depth traces are not included in our computational area (Fig. 2c) and (ii) 45 buildings are outside the simulated envelope, which is shorter than the surveyed one (Fig. 8). The geometric… |
| Page 10, line 220 – 225: This section is not very clear. | We are sorry and simplified it. | Line 257: …RMSE = 0.92 m). Therefore, to develop accurate and reliable curves, we set a 1-m confidence interval including 124 flow depth traces at buildings out of 175 (Fig. 9). In section 4.2, the Sulawesi-Palu tsunami fragility assessment is based on these 124 buildings (DB_Palu2018). *K* and *κ*... |
| Page 11, table 3: What are the sources for the volume of the landslides? | Corrected | Please, see the changes in "Major comments" table. |

| | | |
|---|---|---|
| Page 12, figure 8: What is the source of the topography and bathymetry data for the DEM with 1 m resolution? | Corrected | Please, see the changes in "Major comments" table. |
| Page 12, figure 9: Figure is misspelt. The figure demonstrates that the model does not well represent the observations. What is an S8 ratio of 1.2? | Thank you very much. We corrected it. | Please, see the changes in "Major comments" table.

Line 252: For a landslide ratio of 1.2 (i.e., S8 volume is multiplied by 1.2), the tsunami model… |
| Page 12, line 237 – page 13, line 239: I suggest putting the number and type of buildings for all locations in a table. | We thank the reviewer for his recommendations. | **Please, see the revised Tables 2 and 4.** |
| Page 13, table 4: Please specify why you only use 124 out of 463 flow depth values for Palu. | After the 2018 Sulawesi-Palu tsunami, Paulik et al. (2019) created a database including 463 flow depth traces at buildings. In our computational area (Palu-City), there are only 220 buildings with observed flow depth values and our simulated tsunami envelope covers 175 buildings. According to Fig. 9, the standard deviation and the Root Mean Square Error (RMSE) are high ($\kappa = 2.18$, RMSE = 0.92 m), so we decide to set a 1-m confidence interval, including 124 flow depth traces at buildings, to develop accurate curves. | Line 253: … (a = 1.027). The simulated tsunami inundation zone overlays 175 traces out of 371 because (i) 151 buildings with flow depth traces are not included in our computational area (Fig. 2c) and (ii) 45 buildings are outside the simulated envelope, which is shorter than the surveyed one (Fig. 8). The geometric… |
| Page 13, line 252: Please put 'Global Earthquake Model' the first time you use the abbreviation. | Corrected | Please, see the changes in "Major comments" table. |
| Page 13, section 4.1 & section 4.2: What is the difference? First, you identify the explanatory variable for building damage. Then in section 4.2 you include the damage states and the model selection. I believe you could make this section 4 much shorter by focusing on the relevant information. I suggest preparing a short and concise paragraph on the statistical methods used and then present the results for each site. I also miss a short introductory phrase to the Akaike Information Criterion (AIC) and the likelihood ratio tests you applied. | We thank the reviewer for their feedback. We changed the flow of section 4 in order to be more concise. The fragility assessment for each database includes two steps, in the first step, a simple model is fitted to subsets of the database to identify trends in the data. Then a more complex model is built which is fitted to the whole database based on the observations of the exploratory analysis. We also added a very brief description of the two goodness-of-fit tests we adopted and added references where the reader can find more information or examples regarding these tests. | **Please, see the revised Section 4.** |
| Page 14, 284 f.: There is something wrong in this sentence 'The intercept of the curves for the two material types appear be sustainably different.' | Many thanks for this. It was meant to read substantially. | Line 381: The curves for the two construction types appear to be substantially different. |
| Page 15, 16 and 17, figures 10, 11 and 12: I recommend plotting the confidence intervals with lines only | We thank the reviewer for this. We decided to change the legend. Indeed the term construction type is more | **Please, see the revised Figs. 10-16 below.** |

| | | |
|---|---|---|
| without shaded areas because in the overlapping areas you get different colours than depicted in the legend. In case some reader would like to print the manuscript in black and white, only it will be more illustrative. Furthermore, I suggest introducing a symbol for the variable flow depth. I believe it is better to delete the word material in the legend since it creates some ambiguity with the symbols used. Also, confined masonry is not a material; it is a construction or building type. In figures 10 and 11, you use the same symbol for timber and reinforced concrete, I suggest selecting a unique symbol for each construction type. | appropriate. We also decided to add lines to highlight the 90% confidence intervals. However, we also decided to keep the fill to also indicate the confidence intervals as this way we found it easier to read the figure in print or online. | |
| Page 15, line 301: Instead of material I would suggest using construction or building type. Confined masonry or reinforced concrete are construction types, not materials. | We changed the term. It now reads construction type. | **Please, see the changes in the manuscript.** |
| Page 15, figure 10 and line 293, etc.: You use a couple of times 'in order to'. There is no need for using 'in order', a simple 'to' is enough. | Corrected | **Please, see the changes in the manuscript.** |
| Page 16, line 305 f.: Instead of 'GLM models are finally fitted to DB_Thailand2004 in order to construct fragility curves and their 90 % confidence intervals for the three individual damage states, as depicted in Fig. 12.' I suggest to writing: 'We fit GLM model to DB_Thailand2004 to construct fragility curves for the three damage states and plot them with their 90 % confidence interval in Fig. 12.'. Generally, I suggest avoiding passive voice use because of this increase the readability of a manuscript. | Thank you for the correction. We rewrote as much as possible the passive voice to the active one. | **Please, see the changes in the manuscript.** |
| Page 17, Eq. 18: The indexing of the model equations is not precise. Please check the standard of the journal. | Corrected | $$\eta_{ij} = \begin{cases} \theta_0 + \theta_1 \tilde{x}_j & (19.1) \\ \theta_0 + \theta_{1i} \tilde{x}_j & (19.2) \\ \theta_0 + \theta_1 \tilde{x}_j + \theta_2 class & (19.3) \\ \theta_0 + \theta_{1i} \tilde{x}_j + \theta_2 class & (19.4) \\ \theta_0 + \theta_1 \tilde{x}_j + \theta_2 class + \theta_3 \tilde{x}_j class & (19.5) \end{cases}$$ |
| Page 18, lines 329 – 337: I suggest depicting the functions in exemplary plots, and possibly you could simplify the verbal description. Line 331: Eq. 8.2 does not exist. | Corrected | Line 348: By contrast, Eq.(19.2) allows… |
| Page 18, line 348: Please explain what is meant by AIC values. | Corrected | Line 360: Firstly, we compare the Akaike Information Criterion (AIC) values (Akaike, 1974), which estimates the prediction error of the examined models. |

| Page 19, line 380: Please define hydrodynamic force and explain how you compute it. | We are sorry and corrected it. | Line 172: The hydrodynamic force acting on buildings and infrastructure is defined as the drag force per unit width of the structure (Koshimura et al., 2009b).

$$F = \frac{1}{2} C_D \rho u^2 D$$

$C_D$ represents the drag coefficient ($C_D = 1.0$), $\rho$ is the water density ($\rho = 1000$ kg/m$^3$), $u$ stands for the current velocity (m/s), and $D$ is the inundation depth (m). |
|---|---|---|
| Page 20, line 409 – 415: This is much text for a simple conclusion. Please simplify. | Corrected | Line 452: Based on the observations of the exploratory analysis, we consider model M1 as the most suitable. To test its goodness of fit, model M2, which relaxes the assumption that the slope of all three curves is identical, is also fitted to the data. The comparison of the AIC values for the two models also shows in Table 9 that the M1 is the model which fits the data best for all three link functions considered in this study (i.e., probit, logit and cloglog). We also perform a likelihood ratio test to confirm that the improvement in the fit provided by the more complex M2 model over M1 is not statistically significant. The *p*-value is found to be equal to 0.76, 0.95 and 0.33 for the probit, logit and cloglog functions, respectively, which is significantly above the 0.05 threshold. This suggests that M2 does not provide a statistically better fit to the data, therefore the less complex M1 model fits the data best. The regression coefficients of the 2004 Indian Ocean (Thailand) fragility curves for the best fitted model M1 with logit link function can be found in Table D3 (Appendix D). |
| Page 20, line 416: IO. Please be consistent. | Corrected | Please, see the changes in "Major comments" table. |
| Page 21, line 434 f.: Here you write: 'The curves suggest that confined masonry-type buildings have higher performance than timber structures.' On page 19, line 388: 'A timber building is found to sustain more damage than a confined masonry one for all damage states.' Please clarify. | We are sorry and rephrase the second sentence. | Line 416: A timber building is found to sustain more damage than confined masonry buildings for the more intense damage states. |
| Page 22, figure 13: It is unclear which curve was produced by Syamsidik et al. (2020). The red dashed line is | Thank you very much for your suggestions. We only displays the *ds₃*-data as they are useful for the | **Please, see the revised Figs. 13-16 below.** |

| | | |
|---|---|---|
| Syamsidik et al. (2020)? Please put the reference also in the legend of the figure to be exact. Can you explain why the curve of Syamsidik et al. (2020) is that different to yours? I do not understand why you put the data points in the figures. Is there no better way to illustrate your data? It is hard to distinguish the data points some of the red points that superimpose orange ones are hard to see. In figure 13 (a) and (b) why are the points distributed differently? Moreover, I propose putting 'confined masonry buildings' in the figure. Maybe in the legend or in a figure title. Otherwise, the reader may mix up the figures of confined masonry and timber. I suggest plotting the observed flow depth's fragility curves with the ones of the simulated flow depth. Otherwise, it is hard to see any difference. Is there a difference? | discussion. We also added the references to the figures. We mainly attribute the difference between our $ds_3$-curve and the one produced by Syamsidik et al. (2020) to the fact that a few data are available beyond 4.5-5 m, so the uncertainty increases and the confidence interval widens. | |
| Page 23, figure 14: Please consider the comments of figure 13 also for this figure 14. I suggest putting 'timber buildings' somewhere in the figure. Maybe in the legend or a figure title. Otherwise, the reader may mix up the figures of confined masonry and timber. | Corrected. | **Please, see the revised Figs. 13-15 below.** |
| Page 24, line 458: Please review and correct the following sentences: 'The fragility curves based on observation and simulation are similar enough to consider the computed curves as functions of the hydrodynamic features of the tsunami reliable (Fig. 15c,d). | Corrected | Line 474: Consequently, the tsunami functions based on observation and simulation are highly similar, which illustrates the accuracy and the reliability of the tsunami inundation model. We also display the curves as functions of the maximum simulated flow velocity in Figs. 13b,14b, and the hydrodynamic force in Figs. 13c,14c.

 Line 495: The fragility curves based on the observed and simulated flow depths are relatively similar, especially for $ds_1$ and $ds_3$. The curves based on the flow velocity and the hydrodynamic force are also displayed in Fig. 15b,c. |
| Page 25, figure 15: Please consider the comments for figures 13 and 14 for this figure. I suggest plotting the fragility curves of observed and simulated flow depth in the same graph (hence combine (a) and (b)). | Corrected | **Please, see the revised Figs. 13-15 below.** |
| Page 25, table 11: Why do you use tsunami intensity measure values | As we combine the curves based on the simulated and observed flow depths, we deleted Tables 10 and 11. | **Please, see the changes in the manuscript.** |

| | | |
|---|---|---|
| different from Table 10? It makes them less comparable. | | |
| Page 26, line 472: If Koshimura et al. (2009a) building fragility curves are for mixed buildings and your curves are for mixed buildings are they comparable? What are the percentages of each construction type? | Even if the curves are for mixed buildings, it is difficult to compare them as the percentage of construction type may be different at both locations. Moreover, Banda Aceh curve are based on visual damage interpretation of remaining roofs using the pre and post-tsunami satellite data (IKONOS). We cannot determine the percentage of each construction type. According to Koshimura et al. (2009a) and Saatcioglu et al. (2006), there are low-rise wooden, timber-framed and non-engineered reinforced-concrete constructions in Banda Aceh. | / |
| Page 26, line 488: How do you explain why Banda Aceh buildings are destroyed at 6 m/s flow velocity whereas in Palu and Sunda Strait buildings sustain this flow velocity? | The flow velocity and the force are not good descriptors of tsunami damage. We discussed the building damage probability based on the flow depth only. | **Please, see the revised Section 5.3 and the added Section 6.1.** |
| Page 27, figure 16: I suggest adding the reference to the legend of the figure. | Corrected | **Please, see the revised Fig. 16 below.** |
| Page 28, table 13: It is not proven that a non-seismic source triggered the Palu tsunami. You need to explain what the symbols + and − mean in the lines liquefaction and ground shaking. I suggest changing the line 'Construction material' to 'Construction type'. | We agreed with the reviewer. Corrected. | **Please, see the revised Table 13.** |
| Page 28, line 514 f.: Why do you think the Sunda Strait event buildings reveal a better performance than in Khao Lak? How many buildings were affected in Sunda Strait with flow depth values larger than 5m? How much area was inundated with flow depth values larger than 5 m? | We made a mistake and corrected it. Thank you for pointing this out. 9 buildings collapsed with flow depth values larger than 5 m against 7 structures in Khao Lak/Phuket. | Line 571: As few data points for completely damaged buildings are available beyond this value, the Sunda Strait and the Indian Ocean (Thailand) curves reliability is insufficient. Even though the long wave periods of the IOT (Thailand) seem to increase the likelihood of building damage, the sample size of collapsed buildings beyond 5-m flow depth is too short to confirm this assumption. |
| Page 28, line 527 f.: Please be aware that the largest areas of liquefaction are located outside of the tsunami inundation area. Please see Watkinson & Hall (2019) and Syifa et al. (2019). Although Sassa and Takagawa (2019) identified some small liquefaction areas near the coast, you need to quantify how liquefaction processes effectively damaged many buildings in your database. | Corrected. | Please, see the changes in "Major comments" table. |

| | | |
|---|---|---|
| Page 28, line 529: It is not proven that only landslides generated the Palu-tsunami. I suggest changing the phrasing' non-seismic source'. | We agreed and deleted the sentence. Palu tsunami inundation model is now part of the discussion. | **Please, see the added Section 6.1.** |
| Page 29, lines 532 – 536: The problem here is that your estimates of the flow velocity are based on numerical modelling for the Sunda Strait with 20 m resolution and for the Palu event with 1 m resolution. This makes them hardly comparable. | We agreed with the reviewer. We discussed the impact of the DEM resolution on the flow velocity in Section 6.1. The flow velocity and the hydrodynamic force are not good descriptors of damage, we are not using them to discuss the tsunami impact on building damage probability. | Please, see the changes in "Major comments" table. |
| Page 29, lines 536 – 540: This argument is not enough to conclude that liquefaction was the principal cause for structural destruction. Mas et al. (2020) write that tsunami hydrodynamic, and debris impact forces may have been the principal causes of failure and collapse in Palu Bay's waterfront area. | Corrected | Please, see the changes in "Major comments" table. |
| Page 29, line 540: It is pure speculation that liquefaction episodes are mostly responsible for the building damage. Prove it. The rest of the manuscript is based on this hypothesis. Consequently, you should present some facts or rewrite the last part of the discussion and conclusion. | Corrected | Please, see the changes in "Major comments" table. |

[Figure]

**Figure 2. (a,c) Computational areas (1-3) in the Sunda Strait and Palu-City, magnified view of the building occupation ratio (b) in the Sunda Strait (20-m resolution) and (d) 
[revised manuscript text omitted]

---

## Author Response (AR2)

**Dear Editor,**

Once again, we would like to thank you for your decision and your time. We made the corrections required by the reviewer. All changes are highlighted in blue in the manuscript.

**Dear Referee 3,**

We would like to thank you for the time spent on our manuscript. We highly appreciate your constructive comments and suggestions. You also pointed out the clarifications required to improve the original manuscript. We modified the manuscript according to your recommendations. Please find our answers and corrections below (all changes are highlighted in blue in the manuscript).

- **Specific comments**

| Reviewer comments | Our answers | Corrected manuscript |
|---|---|---|
| P1,L23, change "of both tsunamis" to "of these two tsunamis" | Corrected | Line 23: … the hydrodynamic force of these two tsunamis for the first time. |
| P1,L25, change "for each event" to "for both events" | Corrected | Line 25: …tsunami damage for both events. |
| P1,L41-42: the sentence "These tsunamis are likely to cause greater damage due to surrounding areas affected by prior damage due to ground shaking and/or liquefaction" requires rephrasing, please try: "These tsunamis are likely to cause greater destruction as they can follow prior damaging earthquake ground shaking and/or liquefaction" | We thank the reviewer for the suggestion and corrected it. | Line 41-42: These tsunamis are likely to cause greater destruction as they can follow prior damaging earthquake ground shaking and/or liquefaction (Sumer et al., 2007; Sutikno, 2016). |
| P2, L43: the sentence "The tsunamis also tend to have longer wave periods attacking the coast" is unclear as all tsunamis cause long waves. | Corrected | Line 43: Earthquake-generated tsunamis also tend to have longer wave periods attacking the coast than non-seismic ones. |
| P2, L46: add "a" before "strong". | Corrected | Line 46: … a strong ground shaking… |
| P2,L47: In the sentence "This megathrust earthquake was the second largest ever recorded (wave period ranging from 20 to 50 min) (Løvholt et al., 2006)" the authors recall that the Indian Ocean events was the 2nd largest recorded earthquake while they refer to the tsunami periods to support that. This sentence needs rephrasing because tsunami metrics cannot be used in such a way to infer the earthquake intensity. | We agreed with the reviewer and deleted the information related to the wave period. | Line 47: This megathrust earthquake was the second largest ever recorded (wave period ranging from 20 to 50 min) (Løvholt et al., 2006) and caused the deadliest tsunami in the world. |
| P2,L48: add "of" before "Asian" | Corrected | Line 48: …a dozen of Asian and African… |
| P2,L51: replace "one" by "tsunami waves" | Corrected | Line 51: …can also initiate tsunami waves… |
| P2,L53: change "a short" to "a relatively short" | Corrected | Line 53: …a relatively short wave period tsunami… |
| P2,L59: change "loss property" to "loss to property" and cite Omira et al. 2019 paper on the field Palu post-tsunami field survey. | Corrected | Line 59: …and considerable loss to property (Association of Southeast Asian Nations (ASEAN)-Coordinating Centre for Humanitarian Assistance on disaster, 2018; Omira et al., 2019). |
| P2,L60-61: the sentence "The Sulawesi earthquake ($M_w$= 7.5) occurred along the Palu-Koro strike- | We thank the reviewer for the clarification and corrected it. | Line 60-61: The Sulawesi earthquake ($M_w$= 7.5) occurred near the Palu-Koro strike-slip fault, 50 km northwest of |

| | | |
|---|---|---|
| slip fault, 50 km northwest of Palu-Bay" needs revision as the earthquake was initiated outside the Palu-Koro fault and only partially ruptured it (see Socquet et al. 2019 papers among others). | | Palu-Bay (Fig. 1d) (Socquet et al., 2019). |
| P2,L65-67: Also the sentence "So far, the main hypothesis is that the horizontal displacement of the fault triggered a massive submarine landslide inside Palu-Bay, responsible for the main tsunami." needs substantial revision as there are, in my opinion, more plausible hypotheses of the tsunami generation (coastal landslides, horizontal coseismic deformation, combination of coastal landslides and coseismic deformation …) than "a massive submarine landslide" that should be easily identifiable from the post-event bathymetric survey (Frederik et al., 2019) | We thank the reviewer for pointing this out and rephrased it. | Line 65: Some studies suggested that submarine landslides are responsible for the main tsunami. Moreover, a dozen of coastal landslides were reported during field surveys and likely contributed to amplify tsunami waves (Arikawa et al., 2018; Heidarzadeh et al., 2019; Muhari et al., 2018; Omira et al., 2019; Pakoksung et al., 2019). However, according to Ulrich et al. (2019), those subaerial/submarine landslides may not be the only tsunami source as the Sulawesi earthquake rupture may have also induced a large portion of the tsunami waves. |
| :P2,L68: the reference to Heidarzadeh et al. 2018 is not adequate here. | Corrected | Line 68: (Arikawa et al., 2018; Heidarzadeh et al., 2019; Muhari et al., 2018; Omira et al., 2019; Pakoksung et al., 2019). |
| P2,L71: omit "new" before "measure", "recently developed/proposed" could fit better? | Corrected | Line 71: …is a measure recently proposed to estimate… |
| P3,L93: replace "…..poorly understood" by "…., remaining less understood". | Corrected | Line 93: …uncommon events remaining less understood… |
| P3,L97: change "with the curves of the 2004 IOT" to "to those derived for the 2004 IOT" | Corrected | Line 97: …tsunamis to those derived for the 2004 IOT… |
| P4,L108: start a new sentence after "events": These databases …. | Corrected | Line 108: …by these events. These databases… |
| P5,Section 2.1 and Table 1: It is not clear to me how two different damage states "Minor" and "Moderate" can be gathered in one unique "Ds1". According to Suppasri et al.'s 2019 classification the "Minor damage" corresponds to no significant structural or non-structural damage with possibility of building use after minor floor and wall clean up, while the "moderate damage" refers to non-structural damage with use after moderate repairs, which indicates the large difference between the two damage states. The authors must provide plausible justification of such a merging. | We agreed with the reviewer that there is a difference between minor and moderate damage states. In this study, we simply gather them to find the best harmonization with the two other damage scales used for each tsunami event. According to Suppasri et al. (2020), "minor damage" represents damages found on windows and doors, no damage on wall and on structural component, and "moderate damage" represents one side wall damages, no damage on column and beam. Considering minor damage equivalent to no damage is not consistent so we believe that minor and moderate damage states are equivalent to "partial damage repairable" mentioned by Paulik et al. (2019) and "damage to secondary members" proposed by Ruangrassamee et al. (2006). | / |

| | | |
|---|---|---|
| P7, Equation 7: the authors presented the hydrodynamic (drag) force and stated that the terms u stands for the maximum current velocity, and D for the maximum inundation depth. However in the drag force the term to be considered is the maximum of the combination $u^2*D$ representing the momentum flux per unit mass, which is different from $(u(max))^2*D(max)$ used by the authors (See Yeh 2007-Design Tsunami Forces for Onshore Structures). This difference can lead to incorrect results and the authors must provide explanation on the way they used the tsunami hydrodynamic force. | We are very sorry for this mistake and corrected it. | Line 176-177: …u stands for the current velocity (m/s), and D is the inundation depth (m). |
| P13, Table 3: It is unclear how the volumes of the landslides are estimated. | We are sorry. We added more explanations. As we do not have the soil property, we used the trial and error method. Based on this method and the topography/bathymetry data provided by BIG after the tsunami, we identified the best landslides parameters (we recreated the landslides slopes with Eqs. 4, 5 and 6). | Line 240: …From the trial and error method and the topographic/bathymetric data provided by the Agency for Geo-spatial Information (BIG), Indonesia, we determined the soil property and achieved the volume of the landslides (Table 3). In Figure 7… |
| Results: For both Anak-Krakatau and Palu tsunamis, the authors are asked to present not only the inundation maps but also the results of tsunami generation depicting snapshots of landslide downslope dynamics and generated waves. | We added the results of tsunami generation depicting snapshots of landslide downslope dynamics and generated waves, as suggested by the reviewer. | Please, see the added Figs. B1-B2 and C1-C2 and the changes in Sections 3.2.2 and 3.2.3. |
| Figure 1: Add the geographical coordinates for all the maps. No need for numbers 1, 2 and 3 in Fig.a instead replace by (b), (c) and (d), respectively. Change the colour of "Sunda Trench" to be easily readable. The orange rectangle in Fig. c doesn't show "Anak Krakatau" but the 4 Islands formed after the 1883 Krakatau eruption. Add a reference to the 2018 Palu earthquake epicentral location. | We thank the reviewer for the suggestions and corrected it. | Please, see the revised Fig. 1. |
| Figure 2: The same as for Figure 1: Add the geographical coordinates for all the maps. No need for numbers 1, 2 and 3 as they refer to Figs. (b), (c) and (d), respectively. | Numbers 1-3 do not refer to Figs. b-d. We are sorry for the confusion. It refers to the 3 computational grids used for the simulation in Sunda Strait area. | Please, see the revised Fig. 2. |
| Figure 5: Add the geographical coordinates for all the maps. | Corrected | Please, see the revised Fig. 5. |
| Figure 6: The same as for Figure 5 | Corrected | Please, see the revised Fig. 6. |